# Enhancing Single-Precision with Quasi Double-Precision: Achieving Double-Precision Accuracy in the Model for Prediction Across Scales-Atmosphere (MPAS-A) version 8.2.1

Jiayi Lai[1], Lanning Wang[1,2], Qizhong Wu[1,2], Yizhou Yang[3], and Fang Wang[4]

[1]College of Global Change and Earth System Science, Faculty of Geographical Science, Beijing Normal University, Beijing 100875, China

[2]Joint Center for Earth System Modeling and High Performance Computing, Beijing Normal University, Beijing 100875, China

[3]National Supercomputing Center, Wuxi 214026, China

[4]CMA Earth System Modeling and Prediction Centre (CEMC), Beijing 100081, China

**Correspondence:** Lanning Wang (wangln@bnu.edu.cn) and Yizhou Yang (yang.yizhou@outlook.com)

**Abstract.** The limitations of high-performance computing (HPC) significantly constrain the development of numerical models. Traditional numerical models often employ double precision to ensure result accuracy, but this comes at a high computational cost. While using lower precision can substantially reduce computational expenses, it may introduce round-off errors that can affect accuracy under certain conditions. The Quasi double-precision algorithm (QDP algorithm) compensates for these round-off errors by maintaining corrections, thus improving result accuracy. To investigate the effectiveness of this algorithm in enhancing the accuracy of numerical model results, this paper applies to the single-precision version of the Model for Prediction Across Scales-Atmosphere (MPAS-A), and its performance is evaluated across two idealized and two real-data cases. The results show that the application of QDP algorithm reduces the surface pressure bias by 68%, 75%, 97%, and 96% in the respective cases. Compared to double-precision experiments, the runtime is reduced by 28.6%, 28.5%, 21.1%, and 5.7%. This study demonstrates that QDP algorithm provides effective and cost-efficient computational capabilities for numerical models.

## 1 Introduction

Since the advent of modern computers in the 1950s, numerical simulation-based weather and climate modeling has emerged as one of the most effective methods for exploring weather and climate systems, providing a new platform for numerical model research (Bauer et al., 2015). However, in order to achieve more accurate and precise simulation results, numerical weather and climate models are evolving towards higher resolutions and more complex physical parameterization schemes (Bauer et al., 2015). With the integration of increasingly complex modules to meet diverse requirements, numerical weather and climate models have developed rapidly, and the next generation of these models will feature unprecedented resolution and complexity (Hatfield et al., 2019). In light of these circumstances, the demand for more powerful HPC systems and more efficient computational methods has become particularly urgent. As noted by Bauer et al.(2015), the computational tasks of

future numerical model prediction (NMP) systems are expected to be 100 to 1000 times greater than those of 2015's systems. To bridge the gap between hardware advancements and application performance, the design of code and the selection of algorithms must focus on the optimization of floating-point operations and memory usage (Hatfield et al., 2019).

Mixed precision is a promising research direction in optimizing computational resources within numerical models. By reducing the bit-width required for number representation and thereby lowering the precision of floating-point numbers, mixed precision methods enable storage and computations to be performed with fewer bits. This approach reduces the computational and communication costs in numerical simulations such as climate modeling. Lower precision numerical representations is a feasible approach to reducing computational costs in complex numerical models (Dawson et al., 2017). Low precision computations, defined as operations utilizing fewer than 64 bits of significance, remarkably decrease resource requirements but may introduce round-off errors. To address this challenges, the study of mixed precision techniques has emerged.

In recent years, notable advancements have been made in the application of mixed-precision computing in numerical weather and climate models. Váňa et al. (2016) investigated the implementation of mixed-precision computing in the Integrated Forecast System (IFS) prediction model. They employed double precision in certain regions while utilizing lower precision in others. This approach significantly enhanced computational efficiency by an average of 40% while maintaining acceptable error margins. Dawson et al. (2018) expanded the scope of mixed-precision methods, demonstrating their applicability to simple thermal diffusion models, while key state variables are stored and updated with higher precision. For more complex real-world land surface schemes, they showed that using lower precision for the majority of computations while ensuring high-precision processing of state variables could still meet the requisite accuracy standards. Concurrently, Nakano et al. (2018) conducted an in-depth study on the dynamical core of the global compressible non-hydrostatic model, particularly in the baroclinic wave tests by Jablonowski and Williamson. Nakano et al. (2018) opted to use double precision for grid geometry calculations and single precision for other components. The results indicated that this strategy not only successfully simulated the growth of baroclinic waves with minimal error also reduced runtime by 46%. This study further corroborated the efficacy of mixed-precision computing in dynamical core calculations. Hatfield et al. (2019) applied mixed-precision computing to the Legendre transform in the IFS, successfully implementing half-precision computations. Remarkably, this modification reduced the computational cost to 25% of that in the double-precision reference test, significantly lowering computational overhead. This achievement underscored the substantial potential of mixed-precision computing in large-scale numerical prediction models. Oriol Tintó et al. (2019) applied mixed-precision methods to the Nucleus for European Modelling of the Ocean (NEMO). They discovered that 95.8% of the 962 variables could be computed using single precision. Additionally, in the Regional Ocean Modeling System (ROMS), all 1146 variables could be computed using single precision, with 80.7% of them even using half precision. This finding suggests that mixed-precision methods have extensive applicability in ocean modeling. Cotronei at al. (2020) converted the radiation component of the atmospheric model ECHAM to a single-precision algorithm, resulting in an approximately 40% acceleration in radiation calculations. This result indicates that applying single-precision computing in atmospheric models can significantly enhance computational efficiency while preserving computational accuracy to a reasonable extent. Paxton at al. (2022) further investigated the feasibility of reduced-precision computing. He conducted tests in the Lorenz system, shallow water approximation over a ridge, and the simplified parameterized coarse-resolution spectral global atmospheric

model (SPEEDY). The findings revealed that single precision (23 bits) sufficed for most computational needs, and in numerous cases, half precision (10 bits) could also achieve the desired results. This provides an important reference for adopting lower-precision computing in various models in the future. This year, Hugo et al. (2024) further substantiated the effectiveness of mixed-precision methods in the regional weather and climate model COSMO. The study found that the differences between double-precision and single-precision simulations were minimal, typically detectable only in the initial few hours or days of the simulation. However, single-precision simulations reduced computational costs by approximately 30%. In the same year, Chen et al. (2024) applied the principle of limited iterative development to identify equations that were insensitive to precision in weather and climate modeling tests, modifying them from double precision to single precision. This optimization resulted in a reduction of the runtime of the model's hydrostatic solver, non-hydrostatic solver, and tracer transport solver by 24%, 27%, and 44%, respectively, thereby substantially enhancing computational efficiency. In summary, mixed-precision computing exhibits broad application prospects and potential advantages in numerical weather and climate modeling. By flexibly applying varying precision computing methods while ensuring predictive accuracy, it is feasible to significantly enhance computational efficiency and reduce computational costs.

When utilizing mixed-precision computation, low-precision calculations inevitably introduce rounding errors, particularly when adding numbers with significantly different magnitudes. In such scenarios, the limited precision can cause the larger number to effectively "swallow" the smaller number, thereby compromising the accuracy of the result. For instance, consider the variables $A = 0.7315 \times 10^3$ (a large number) and $B = 0.4506 \times 10^{-5}$ (a small number). If the precision of the result is reduced to 4 significant digits, the outcome will be $0.7315 \times 10^3$, with the large number effectively overshadowing the small one. This phenomenon is especially pertinent in numerical modeling, where the introduction of biases into fundamental fields often necessitates the addition of large and small numbers, inherently causing round-off errors. These errors can accumulate over successive computations, leading to a degradation in model accuracy or even complete failure. Therefore, this issue can not be overlooked.

There have some methods to address the round-off errors. In an early study, Gill (1951) proposed a fourth-order, four-step explicit Runge-Kutta method aimed at correcting round-off errors during computation. This method constructs auxiliary variables at each step to compensate for the rounding errors generated, thereby further refining the results to achieve higher precision. However, this method is not applicable to other forms of numerical solutions. In addition to this, compensated summation methods can enhance the accuracy of summation by utilizing the floating-point precision supported by lower-level hardware (Higham, 1996). These methods rely on recursive summation and incorporate correction terms to reduce round-off errors. Subsequently, Møller (1965) and Kahan (1965) respectively proposed the QDP algorithm and the Kahan algorithm. The primary idea behind both methods is to make slight adjustments to the total sum to avoid the precision loss caused by adding a small, precise value to a much larger one in floating-point addition. The QDP algorithm has been validated in solving ordinary differential equations using the fourth-order Runge-Kutta method (Møller, 1965), where the error after precision reduction is essentially minimized to zero.

Currently, methods for compensating round-off errors are primarily employed in the step-by-step integration of ordinary differential equations ( Thompson et al., 1970; Tomonori et al., 1995; Dmitruk et al., 2023). However, their validation in

numerical models remains uncertain. Considering the broader applicability of the QDP method, which can be utilized for recursive summation in any format, and its superior performance in HPC environments compared to the Kahan algorithm (Kahan, 1965), this study aims to implement the QDP algorithm in MPAS-A model. The application of the QDP algorithm to a realistic numerical model, as presented in this study, represents a novel contribution to the field, with no prior research exploring this specific implementation.

Most works involving numerical models that reduce numerical precision adopt a mixed-precision scheme, where some variables use single precision while others remain in double precision to ensure integration stability, as demonstrated in the work of Chen et al. (2024). Currently, there are very few studies that almost entirely employ low precision (32-bit) in numerical models, only applied in IFS by Váňa et al. (2016). However, they only utilize single precision without considering error compensation for it. In this study, all variables in the numerical model were implemented using single precision, and QDP algorithm was applied to key variables. By using QDP algorithm, we can maintain integration stability comparable to that applying double precision scheme while significantly reducing memory requirements by lowering the numerical precision of all variables and improved the accuracy comparable to that applying the single precision. This approach not only reduces communication pressure but also allows for substantial increases in computational speed through vectorization optimization. The structure of this paper is as follows: Section 2 introduces the QDP algorithm, the MPAS model, application of QDP algorithm in MPAS-A, and the experimental design and configuration. Section 3 provides case study in MPAS. Section 4 presents conclusions and discussion of the experiments.

For clarity, the following abbreviations and glossary as shown in Table 2 are used throughout the all:

**Table 1.** Key abbreviations and glossary used in this study.

| Abbreviation | Definition/Description |
| --- | --- |
| MPAS-A | The atmospheric component of The Model for Prediction Across Scales (MPAS) |
| DBL | Double Precision simulation, used as the benchmark |
| SGL | Single Precision simulation |
| QDP | Single Precision simulation with the QDP applied (In this study, we apply the Quasi Double Precision method only to single-precision simulations. For convenience, we refer to the simulations using the QDP method in single-precision as QDP.) |
| QDP algorithm | Quasi Double-Precision, a algorithm to compensate round-off errors in low-precision computations (To distinguish between the Quasi Double Precision algorithm and single-precision simulations utilizing the Quasi Double Precision algorithm, we refer to the QDP algorithm as the "QDP algorithm" and the simulations as QDP) |

$$
\begin{aligned}
&1:\quad u := initial\ u;\\
&2:\quad c := 0;\\
&3:L:\ v := (< evalution\ of\ v >) + c;\\
&4:\quad s := u + v;\\
&5:\quad c := \big(v - (s - u)\big) + \Big(u - \big(s - (s - u)\big)\Big)\\
&6:\quad u := s\\
&7:\quad go\ to\ L:
\end{aligned}
$$

**Figure 1.** Iterative Process of QDP algorithm in Step-by-Step Integration.

$$
\begin{aligned}
&1:\quad c := 0;\\
&2:L:if\ u > v\ then;\\
&3:\qquad u := initial\ u;\\
&4:\qquad v := (< evalution\ of\ v >) + c;\\
&5:\qquad s := u + v;\\
&6:\qquad c := \big(v - (s - u)\big) + \Big(u - \big(s - (s - u)\big)\Big)\\
&7:\quad else\\
&8:\qquad v := initial\ v;\\
&9:\qquad u := (< evalution\ of\ u >) + c;\\
&10:\qquad s := u + v;\\
&11:\qquad c := \big(u - (s - v)\big) + \Big(v - \big(s - (s - v)\big)\Big)\\
&12:go\ to\ L:
\end{aligned}
$$

**Figure 2.** The QDP algorithm with magnitude preconditioning for Identifying large and small numbers.

## 2 Methodology, model and experiments

### 2.1 Quasi double-precision algorithm

The QDP algorithm, proposed by Møller et al. (1965), aims to address the precision loss that occurs when adding small values to large values in floating-point arithmetic. This precision loss typically arises from coarse truncation operations. The QDP algorithm reduces round-off errors by keeping corrections. Primarily applied in the step-by-step integration of ordinary differential equations, the algorithm significantly corrects rounding errors in sum, particularly in computers where truncation operations are not followed by proper rounding.

A brief introduction to the algorithm is as follows, with a detailed derivation available follows Møller et al. (1965). Define the floating-point numbers $u$, $v$, $s$, and $c$, where in each step of the time integration, $s = u + v$. By introducing a correction variable $c$ before computing sum ($s$) of $u$ and $v$ in each step, the final $s$ is adjusted to reduce round-off errors. This algorithm is illustrated in Fig. 1. It illustrate the iterative process (lines 3–7) and its role in error compensation, which enhances accuracy in time integration.

The process can be viewed as $v$ being continuously accumulated onto $u$; however, in numerical models' computations, it is impossible to ensure that $u$ is always greater than $v$. To enhance the precision of the correction process, a precondition of magnitude comparison is added to the algorithm, as shown in Fig. 2.

It is important to note that the applicability of the QDP algorithm has been thoroughly analyzed (Møller et al., 1965) , cases of inapplicability are exceedingly rare. Considering the numerous summation operations involved in numerical models, even if a few inapplicable instances occur, their impact on the overall result is negligible. Therefore, in practical applications, these infrequent cases are typically not considered.

## 2.2 MPAS-A

MPAS-A is a compressible, non-hydrostatic atmospheric numerical model developed by NCAR. It employs an unstructured centroidal Voronoi grid (mesh or tessellation) and a staggered C-grid for state variables as the basis for horizontal discretization in the fluid flow solver. MPAS-A consists of two main components: the model, which includes atmospheric dynamics and physics, and the initialization component, which generates initial conditions for the atmosphere and land surface, updates for sea surface temperature and sea ice, and lateral boundary conditions. Both components (model and initialization) are integral constructs within the MPAS software framework and utilize the same drivers and software infrastructure.

The MPAS-A solves the fully compressible, nonhydrostatic equations of motion (Skamarock et al., 2011). The spatial discretization uses a horizontal (spherical) centroidal Voronoi mesh with a terrain-following geometric-height vertical coordinate and C-grid staggering for momentum. The temporal discretization uses the explicit time-split Runge–Kutta technique from Wicker and Skamarock (2002) and Klemp et al. (2007).

The algorithm applied here primarily addresses the round-off error compensation between large and small numbers in addition. Currently, it is only applicable to the time integration process and has not been implemented in the spatial discretization process. Therefore, this section will provide a detailed introduction to the time integration scheme. For the spatial discretization scheme, please refer to Skamarock et al. (2011), and it will not be introduced upon here.

The formulation of the scheme can be considered in on dimension as Eq. (1)

$$\frac{\partial \phi}{\partial t} = \text{RHS}_\phi \tag{1}$$

The variable $\varnothing$ represents any prognostic variable in the prognostic equations, while RHS represents the right-hand side of the prognostic equations (i.e., the spatial discretization equation). In MPAS- A, a forward-in-time finite difference is used, and it can be written as Eq. (2):

$$\frac{\phi_i^{n+1} - \phi_i^n}{\Delta t} = \text{RHS}_\phi. \tag{2}$$

Where superscript represent the time step, and subscript represent the position of grid zone.

The two-order Runge-Kutta time scheme is used in MPAS-A as described in Gear et al. (1971) as Eq. (3), (4) and (5):

$$\phi^* = \phi^t + \frac{\Delta t}{2} \cdot \text{RHS}(\phi^t), \tag{3}$$

$$\phi^{**} = \phi^t + \frac{\Delta t}{2} \cdot \text{RHS}(\phi^*), \tag{4}$$

$$\phi^{t+\Delta t} = \phi^t + \Delta t \cdot \text{RHS}(\phi^{**}). \tag{5}$$

In this study, the version 8.2.1 of MPAS-A was used for the following reasons:

1. This research primarily focuses on the accumulation of variables in time integration, specifically the accumulation of time integration variables within the dynamical core. Version 8.2.1 supports the option to close physical processes during model construction, preventing the influence of physical processes on the results of the dynamical core. Therefore, this version was chosen. It should be noted that all cases in this study have closed physical processes.

2. This version supports single-precision operations, reducing the repetitive work of code modification. It is not the only version that supports single precision, but the latest version currently released.

## 2.3 Application of QDP algorithm in MPAS-A

According to Eq. (3), (4) and (5), it can be observed that in the time integration scheme, each step involves the process of adding tends on the basic field $\phi^t$. In numerical models, the basic field is generally much larger than the tends, which aligns with the principles of numerical computation regarding the addition of large and small numbers, as well as the time integration process. It is important to note that the QDP algorithm currently only addresses time integration and has not been validated during the spatial discretization process. The spatial discretization primarily involves subtraction, specifically the subtraction of a small number from a large number or the subtraction of two close values. Whether this algorithm is applicable in spatial discretization remains uncertain, therefore, we will not apply it in this context.

Based on the application principles of the algorithm, which involve the processes of adding large and small numbers as well as the time integration process, we have established a strategy for applying the QDP algorithm within the MPAS-A. Specific improvements are provided based on Eq. (6), (7), (8) and (9):

$$\frac{\partial V_H}{\partial t} = -\frac{\rho_d}{\rho_m}\left[\nabla_\zeta\left(\frac{p}{\zeta_z}\right) - \frac{\partial z_H p}{\partial \zeta}\right] - \eta\mathbf{k}\times\mathbf{V}_H - \nu_H\nabla_\zeta\cdot\mathbf{V} - \frac{\partial\Omega\nu_H}{\partial\zeta} - \rho_d\nabla_\zeta K - eW\cos\alpha_r - \frac{\nu_H W}{r_e} + \mathbf{F}_{\mathbf{V}_H}, \tag{6}$$

$$\frac{\partial W}{\partial t} = -\frac{\rho_d}{\rho_m}\left[\frac{\partial p}{\partial\zeta} + g\tilde{\rho}_m\right] - (\nabla\cdot\mathbf{v}W)_\zeta + \frac{uU + vV}{r_e} + e(U\cos\alpha_r - V\sin\alpha_r) + F_W, \tag{7}$$

$$\frac{\partial\Theta_m}{\partial t} = -(\nabla\cdot\mathbf{V}\theta_m)_\zeta + F_{\Theta_m}, \tag{8}$$

$$\frac{\partial\tilde{\rho}_d}{\partial t} = -(\nabla\cdot\mathbf{V})_\zeta. \tag{9}$$

The meaning of each variable in the equations exactly follows Skamarock et al. (2012), so that we don't repeating explanation. For a numerical model, the most crucial variables are the prognostic variables. Therefore, In the MPAS-A model we applied the QDP algorithm to the time integration process of these prognostic variables, including horizontal momentum ($V_H$), dry air density ($\tilde{\rho}_d$), potential temperature ($\Theta_m$) and vertical velocity (W) , that is, the calculation process on the left side of Eq. (6), (7), (8) and (9) (Only the predictive equations for the dynamic core are presented here, without the scalar transport). This study focuses on dynamic core, involving the gravity wave and acoustic wave, so we turned off the scalar transport in all cases. In order to be understood well, we provide the pseudo-code in the supplement.

## 2.4  Experimental design and configuration

This study aims to investigate whether the QDP algorithm can effectively compensate for the round-off errors that caused by reduced numerical precision. Setting DBL as the benchmark experiment. Two control experiments are also established: the first control experiment uses SGL, and the second control experiment uses QDP. By comparing the spatial root-mean-square error (Spatial RMSE) and spatial Mean Absolute Error (MAE) between these two control experiments and the benchmark experiment, this study evaluates the effectiveness of the QDP algorithm in reducing round-off errors.

To assess the application effect of the QDP algorithm, We selected these two ideal cases and the real-data case because they are the only complete datasets available for download on the MPAS website:

1. **Jablonowski and Williamson baroclinic wave:** A deterministic initial-value test case (Jablonowski and Williamson, 2006) for dry dynamical cores of atmospheric general-circulation models, is presented that assesses the evolution of an idealized baroclinic wave in the northern hemisphere. The primary objective is to assess the model's efficacy in replicating the typical dynamics of moist atmospheric conditions across various precision settings.

2. **Super-cell:** A reduced-radius sphere (Klemp et al. 2015) can be used to assess the behavior of nonhydrostatic processes in global atmospheric dynamical cores, as long as the simulated cases demonstrate good agreement with the corresponding flows in Cartesian geometry, for which analytical solutions are available.

3. **Real data:** with initial conditions generated using GFS data at 2014-09-10_00) using two different resolutions (total domain size of 120 km × 120 km and 240 km × 240 km).

To prevent the influence of other factors, the basic parameters of all cases are kept consistent, including the Number of acoustic steps per full RK step, config dynamics split steps, and config number of sub steps (integer), among others.

For clarity, the following abbreviations about test cases as shown in Table 2 are used throughout the all:

**Table 2.** Key abbreviations and glossary about test cases used in this study.

| Abbreviation | Definition/Description |
| --- | --- |
| JW wave | Jablonowski & Williamson baroclinic wave |
| SC | Super-cell |
| RD-120 | Real data with total domain size of $120 \times 120$ km |
| RD-240 | Real data with total domain size of $240 \times 240$ km |

## 3  Results and analysis

In this section, we introduce Spatial RMSE and MAE, and show results across four cases, include two ideal scenarios: Jablonowski and Williamson baroclinic wave and super-cell, as well as two real case ( with initial conditions generated using

GFS data) using two different resolutions. By using the Spatial RMSE and MAE for quantitative comparison, the differences between the benchmark and control experiments are used to evaluate the effectiveness of the QDP algorithm in reducing round-off error.

## 3.1 Spatial RMSE and MAE

To quantify the difference between the SGL, QDP, and DBL, (used as the benchmark), we calculate the Spatial RMSE. First, for each grid point, the temporal averages of the variables (e.g., surface pressure, 500hPa height) are computed across the entire simulation period for each experiment (SGL, QDP, and DBL). Then, the spatial RMSE is calculated as the root-mean-square difference between the temporally averaged fields of the control experiment (SGL or QDP) and the benchmark DBL, following Eq. (10):

$$\text{Spatial RMSE} = \sqrt{\frac{1}{N} \sum_{i=1}^{N} (M_i - C_i)^2}. \tag{10}$$

Where, $N$ is the total number of grid points, $M_i$ is the temporally averaged value at grid point i for the benchmark double-precision experiment, $C_i$ is the temporally averaged value at grid point i for the control experiment (SGL or QDP).

In addition to the spatial RMSE, we also calculate the spatial MAE to assess the magnitude of the difference between the control experiments (SGL and QDP) and the benchmark DBL, irrespective of the direction of the difference. Like the spatial

RMSE calculation, we first compute the temporal average for each grid point across the entire simulation period for each experiment. The MAE is then calculated as the average absolute difference between the temporally averaged fields of the control experiment and the benchmark experiment, following Eq. (11):

$$\text{Spatial MAE} = \frac{1}{N} \sum_{i=1}^{N} |M_i - C_i|. \tag{11}$$

where N is the total number of grid points, $M_i$ represents the temporally averaged value at grid point i for the benchmark

DBL, and $C_i$ represents the temporally averaged value at grid point i for the control experiment (either SGL or QDP).

Spatial RMSE is primarily used to measure the difference between predicted and actual values and is more sensitive to large errors. Spatial MAE calculates the average absolute prediction error, and is less sensitive to outliers than Spatial RMSE, making it more suitable for conventional error measurements. Therefore, the combination of Spatial RMSE and MAE provides a more comprehensive evaluation. When comparing the performance of different experiments, Spatial RMSE may be used to

230 quantify differences in extreme values (such as temperature fluctuations, ocean current speeds, etc.), while Spatial MAE is used to assess the accuracy of the model's overall trend. Combining both provides a better reflection of the algorithm's performance advantages.

As shown in Table 3 (spatial RMSE) and Table 4 (Spatial MAE), the addition of the QDP algorithm consistently improves accuracy (compared to SGL) across all cases. For specific analysis, please refer to the following contents( The results of spatial

RMSE and MAE are consistent, so to avoid duplication, only the results of spatial RMSE are analyzed in the following text).

**Table 3. The spatial RMSE** values of surface pressure compared to DBL for cases, unit: Pa. Note: JW wave = Jablonowski & Williamson baroclinic wave;SC = Super-cell; RD-120/240 = Real data with total domain size of 120/240 km.

| Case name | SGL | QDP (Proposed) |
|---|---|---|
| JW wave | $3.42 \times 10^{-2}$ | $1.09 \times 10^{-2}$ |
| SC | $8.80 \times 10^{-4}$ | $2.27 \times 10^{-4}$ |
| RD-120 | $6.33 \times 10^{-2}$ | $2.25 \times 10^{-3}$ |
| RD-240 | $6.68 \times 10^{-2}$ | $2.25 \times 10^{-3}$ |

**Table 4. The Spatial MAE** values of surface pressure compared to DBL for cases, unit: Pa. Note: JW wave = Jablonowski & Williamson baroclinic wave;SC = Super-cell; RD-120/240 = Real data with total domain size of 120/240 km.

| Case name | SGL | QDP (Proposed) |
|---|---|---|
| JW wave | $1.29 \times 10^{-2}$ | $3.81 \times 10^{-2}$ |
| SC | $8.79 \times 10^{-4}$ | $2.26 \times 10^{-4}$ |
| RD-120 | $5.38 \times 10^{-2}$ | $1.95 \times 10^{-3}$ |
| RD-240 | $5.52 \times 10^{-2}$ | $1.94 \times 10^{-3}$ |

### 3.2 Jablonowski and Williamson baroclinic wave

This case is a deterministic initial-value test case for dry dynamical cores of atmospheric general-circulation models(Jablonowski and Williamson, 2006), assesses the evolution of an idealized baroclinic wave in the northern hemisphere. The initial zonal state is quasi-realistic and entirely defined by analytical expressions, which are steady-state solutions of the adiabatic, invis-
240 cid primitive equations in a pressure-based vertical coordinate system (Jablonowski and Williamson, 2006). The experimental configuration is consistent with the test case presented by Jablonowski and Williamson (2006), with a time step of 450 seconds, 26 vertical levels, total domain size of 120 km × 120 km, and an integration period of 15 days.

The bias begins to appear at the tenth day. Starting from the tenth day, the bias of total energy and total mass caused by SGL can be reduced by using QDP algorithm (Figs. 3a and 3b). Unlike SGL, where the bias increases rapidly after more than 10
245 days, QDP has a very small bias compared to DBL. Therefore, it can be considered that QDP can be used to replace DBL in medium range weather forecast.

It can be found that SGL can increase the round-off error in all regions (Fig. 4a), especially in high-latitude regions, such as Southern Ocean westerly belt, its high wind speed increase error caused by SGL, but instability caused by high wind speeds is more important. Surprisingly, the bias can be reduced significantly in QDP (4b), it means that QDP can improve stability
compared to SGL. It should be emphasized that, this does not mean that the higher the wind speed, the better the improvement effect. Instead, the improvement effect is more pronounced in areas with larger errors. The spatial RMSE of surface pressure between DBL and SGL is $3.42 \times 10^{-2}$ Pa, as well as $1.09 \times 10^{-2}$ Pa between DBL and QDP, the error reduced by 68%.

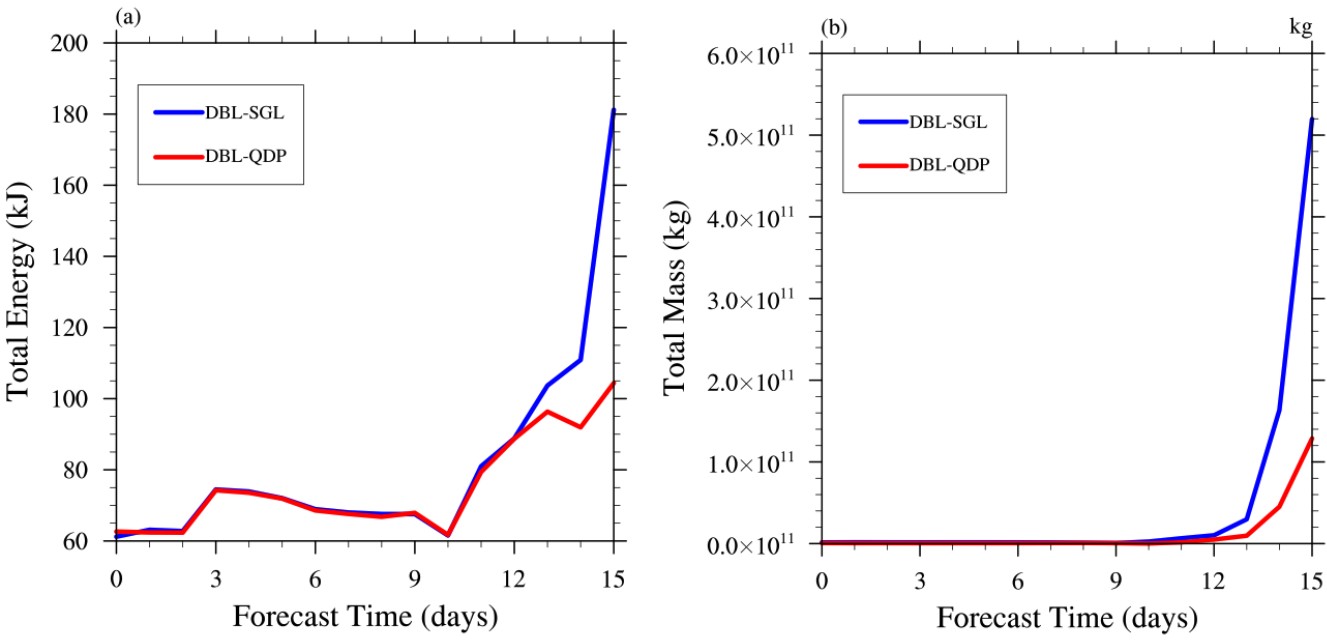

**Figure 3.** Time evolution of differences in (a) total energy and (b) total mass between DBL, SGL, and QDP (proposed) in the JW wave. Both figures highlight QDP's superior error compensation, especially over extended integration periods.

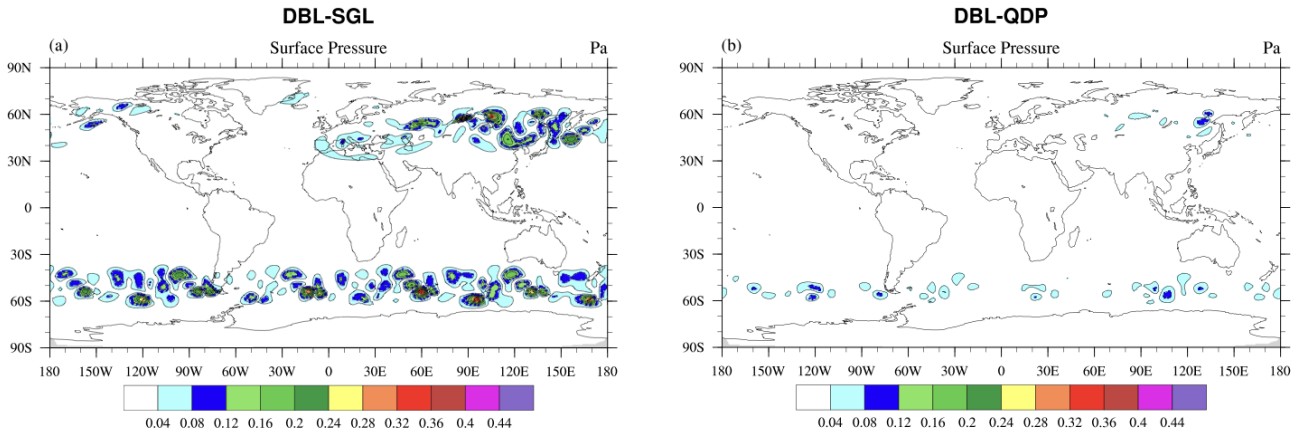

**Figure 4.** Spatial distributions of 1–15 day averaged surface pressure differences (Pa) between DBL and (a) SGL, (b) QDP in the JW wave case. QDP reduces errors more effectively, particularly in mid- to high-latitude regions.

The sources of unpredictability, as noted by Bauer et al. (2015), include instabilities that inject chaotic 'noise' at small scales and the upscale propagation of their energy. For the cases examined, both SGL and QDP begin to exhibit errors after 10 days of

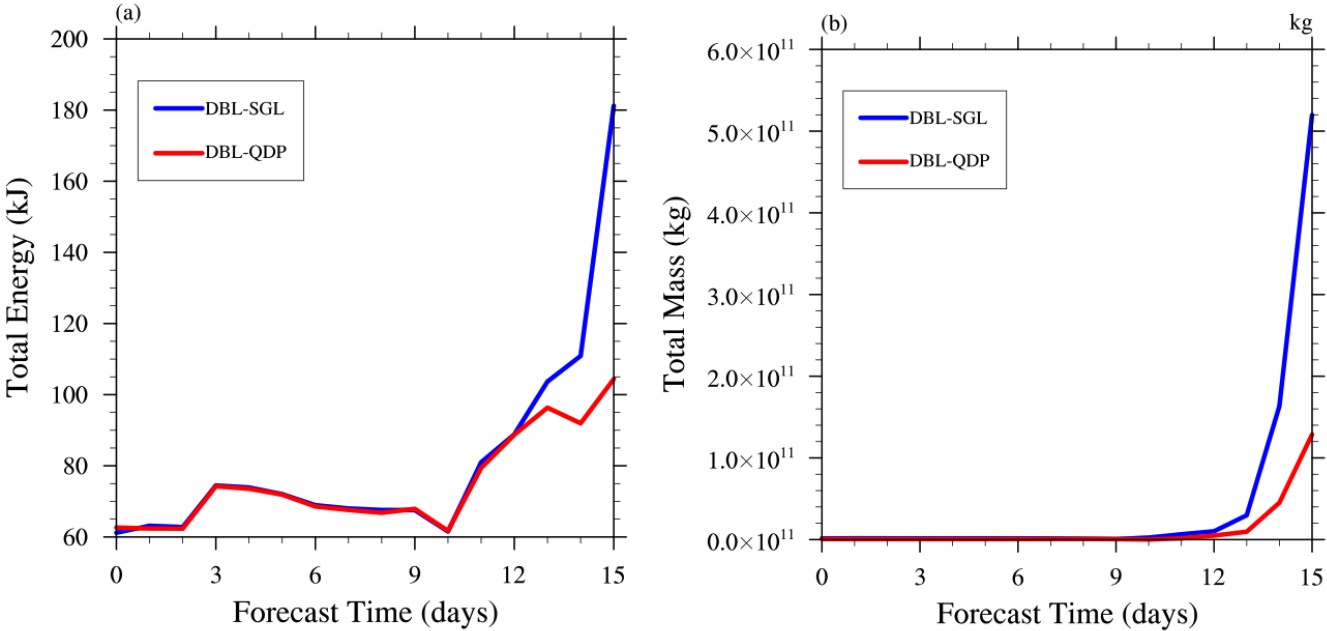

**Figure 5.** Time evolution of differences in (a) total energy and (b) total mass for the supercell case: DBL vs. SGL and DBL vs. QDP. The results highlight QDP algorithm's effective error compensation, with benefits becoming more pronounced over time.

integration. These errors arise from factors such as round-off errors due to reduced numerical precision and energy loss during the propagation process. The QDP algorithm can reduce the impacts of these errors.

While we acknowledge other potential sources of uncertainty, such as initial condition errors, we have not conducted an in-depth study on them in this research. Our primary focus remains on evaluating the improvements provided by the compensation algorithm in addressing round-off errors.

## 3.3 Super-cell case

The test case ( Klemp et al., 2015) is on a reduced-radius sphere, can evaluate the behavior of nonhydrostatic processes in nonhydrostatic global atmospheric dynamical cores provided the simulated cases exhibit good agreement with corresponding flows in a Cartesian geometry, and for which there are known solutions. The settings include a time step of 3 seconds, 40 vertical levels, the total domain size is 84 km × 84 km, and an integration period of 2 hours.

In this case, the reduction of Total energy in error is not significant in QDP (Fig. 5a), except for the initial time, all others showed larger errors than SGL. But the errors of both are negligible. For total mass (Fig. 5b), the error caused by SGL can obtain effective improvement in QDP. This improvement exists throughout the entire integration period.

Figure 6 shows the spatial distribution of perturbation theta, an important variable in numerical models, when reducing the numerical precision from double (Fig. 6a) to single (Fig. 6b), it displays differences, it indicates a significant increase in

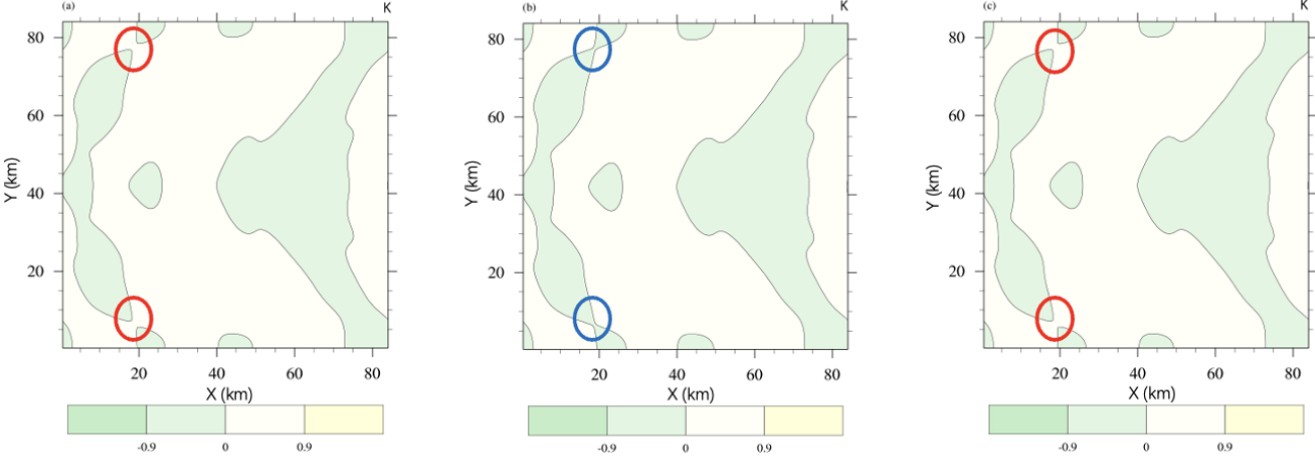

**Figure 6.** Perturbation theta at 5400 s in supercell development: (a) DBL, (b) SGL, and (c) QDP (unit: K). The circle highlights pattern biases (consistent values in the same color). Error regions (blue) appear in (b) with single precision, while QDP reduces these errors in (c).

round-off error. In QDP, this difference can be compensated(Fig. 6c). The spatial RMSE of surface pressure between DBL and SGL is $8.95 \times 10^{-4}$ Pa, as well as $2.19 \times 10^{-4}$ Pa between DBL and QDP, the error reduced by 75%.

### 3.4   Real data cases

In this section, we will show the results from two cases using different resolutions. The settings include a time step of 720 seconds, 55 vertical levels, the total domain size are 240 km × 240 km and 120 km × 120 km, and an integration period of 15
275    days. (Except for the resolution, all other configurations are exactly the same)

    Consistent with the analysis presented in Section 3.2, errors are relatively small in the early stages and begin to emerge after 140 hours. This increase is attributed to the accumulation of round-off errors and energy loss over time. The effects become more pronounced beyond 140 hours. Overall, the QDP algorithm demonstrates a certain level of improvement in addressing these errors. The case with the total domain of 240 km × 240 km (Fig. 7a) show the larger error than 120 km × 120 km (Fig.
7b), and the error can be reduced in QDP caused by SGL.

    Figure 8 and 10 show spatial distributions of surface pressure with different resolution. The error has reduced throughout the all region, and the improvement effect is very obvious. From a spatial perspective, the case of SGL with the total domain size of 240 km × 240 km (Fig. 8a) show the larger error than 120 km × 120 km (Fig. 10a), and the errors both can be reduced by QDP (Figs. 8a and 10a). The spatial RMSE of surface pressure with 240 km × 240 km between DBL and SGL is $6.68 \times 10^{-2}$
285    Pa, as well as $2.25 \times 10^{-3}$ Pa between DBL and QDP, the error reduced by 97%. The spatial RMSE of surface pressuse with 120 km × 120 km between DBL and SGL is $6.33 \times 10^{-2}$ Pa, as well as $2.25 \times 10^{-3}$ Pa between DBL and QDP, the error reduced by 96%.

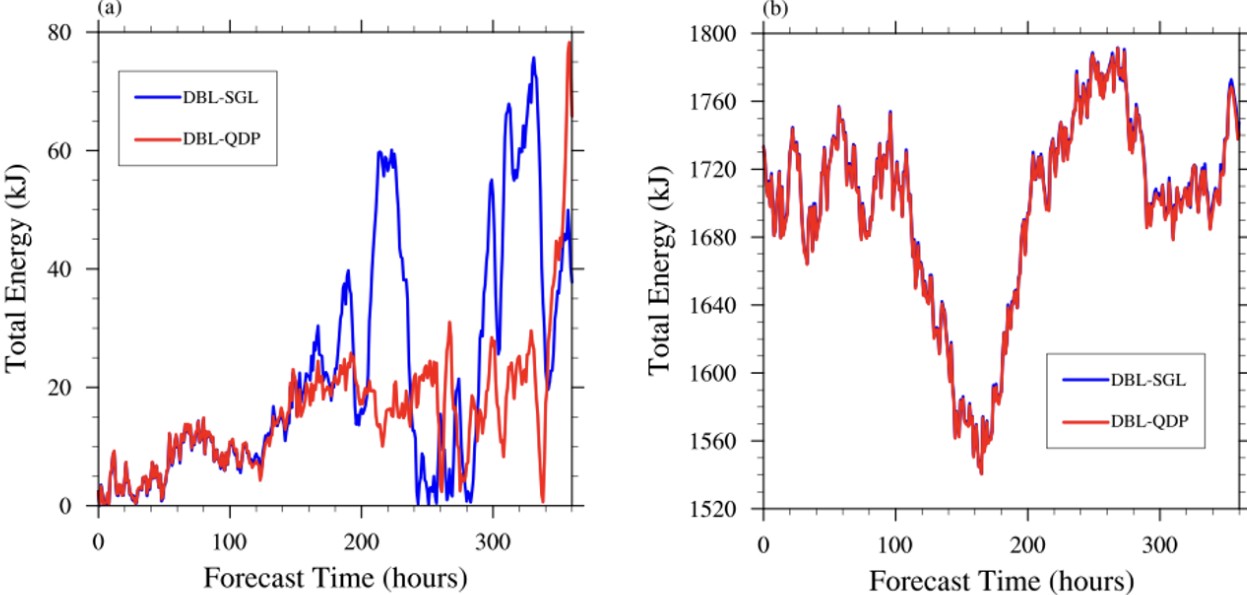

**Figure 7.** Temporal evolution of total energy differences for real-data simulations: DBL vs. SGL (blue) and DBL vs. QDP (red) at resolutions of (a) 240 km × 240 km and (b) 120 km × 120 km. QDP consistently reduces errors from numerical precision loss, with resolution-dependent improvements.

Figure 9 and 11 show spatial distributions of 500 hPa height with different total domain size, 240 km × 240 km (Fig. 9) and 120 km × 120 km (Fig. 11), The error improvement effect is consistent with surface pressure. The spatial RMSE of 500 hPa height with 240 km × 240 km between DBL and SGL is $2.80 \times 10^{-1}$ m, as well as $1.40 \times 10^{-1}$ m between DBL and QDP, the error reduced by 50%. The spatial RMSE of with 120 km × 120 km between DBL and SGL is $4.35 \times 10^{-3}$ Pa, as well as $1.90 \times 10^{-3}$ Pa between DBL and QDP, the error reduced by 56%.

In this research, we focus on the processes of summing the basic field and trends. When the resolution is increased, the basic field remains relatively unchanged; however, the trends become smaller. This characteristic aligns with the nature of adding large and small numbers, making the advantages of the QDP algorithm more pronounced. Thus, it is evident from Figure 8 that as the resolution increases, the improvement achieved by QDP algorithm also enhances.

On the other hand, it is important to note that the propagation of round-off errors is not immediately apparent over short time scales. However, as the number of iterations increases, these errors can become more significant. The QDP algorithm employs compensation mechanisms that help mitigate the propagation of these errors.

Due to the current limitations on the MPAS-A website, which only provides a single set of terrain and initial condition fields for different experiments, our future plan is to request assistance from the MPAS-A website to construct different terrain and initial condition fields for a specific experiment. We aim to conduct sensitivity analysis, particularly for real data experiments.

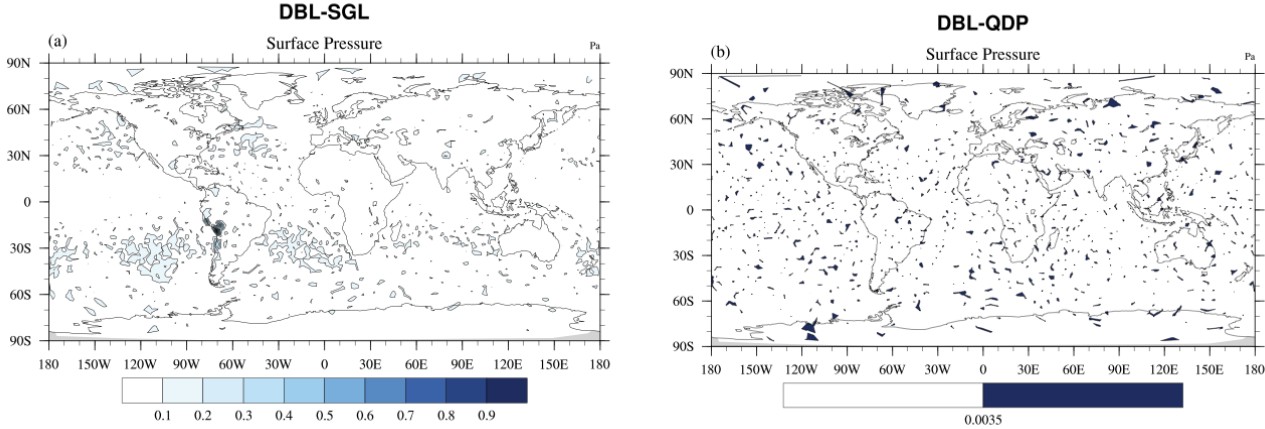

**Figure 8.** Spatial distributions of averaged (1–15 days) surface pressure differences (Pa) between DBL and (a) SGL, (b) QDP (domain size: 240 km × 240 km). The RMSE is $6.68 \times 10^{-2}$ Pa for (a) and $2.25 \times 10^{-3}$ Pa for (b), highlighting QDP's significant error reduction across regions and its effectiveness in correcting errors by several orders of magnitude.

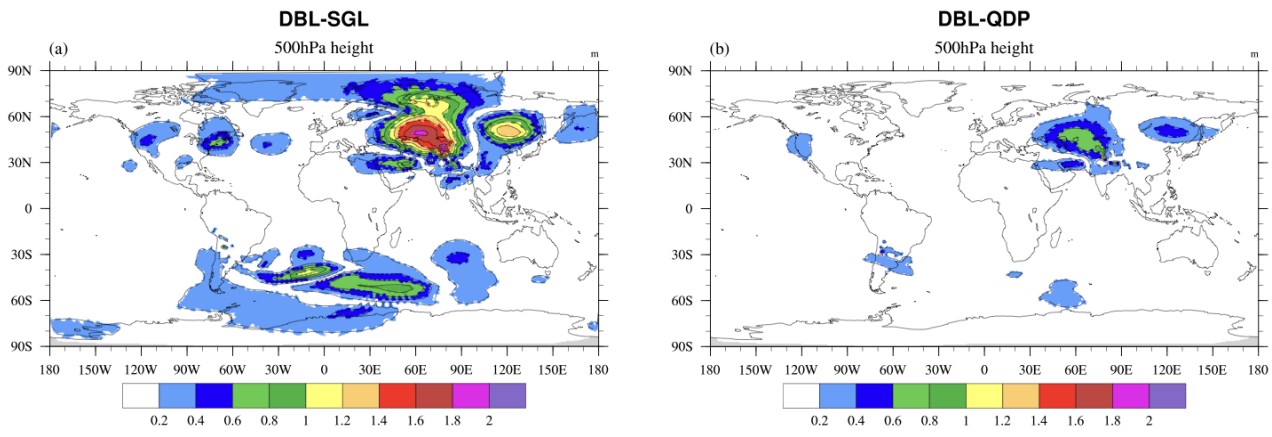

**Figure 9.** Spatial distributions of averaged (1–15 days) 500 hPa height differences (m) between DBL and (a) SGL, (b) QDP (domain size: 240 km × 240 km). The RMSE decreases from $2.80 \times 10^{-1}$ m in (a) to $1.40 \times 10^{-1}$ in (b), indicating notable spatial error reduction with QDP.

Provided that computational resources allow, we plan to carry out simulations with different resolutions and initial conditions. This will help lay the data foundation for future uncertainty analysis.

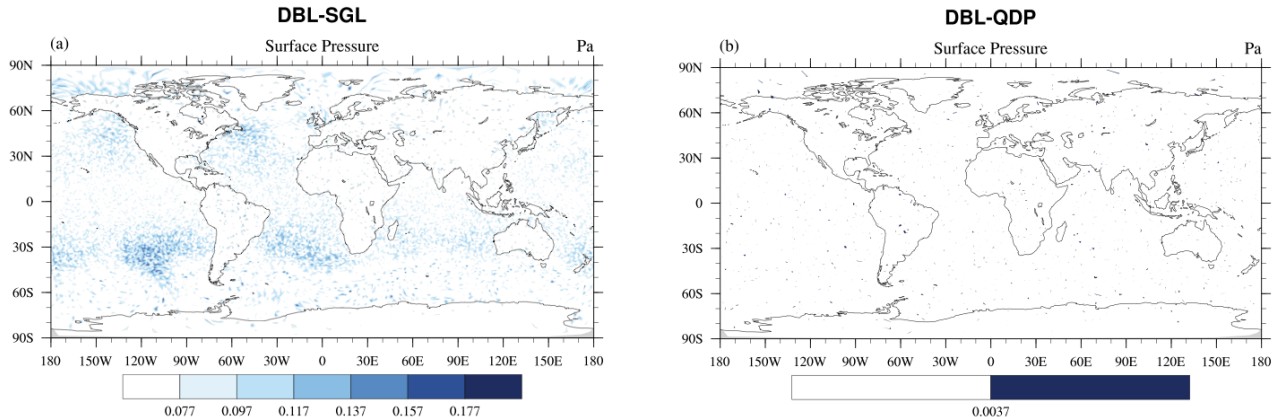

**Figure 10.** Distributions of averaged (1–15 days) surface pressure differences (Pa) between DBL and (a) SGL, (b) QDP (domain size: 120 km $\times$ 120 km). The RMSE decreases from $6.33 \times 10^{-2}$ Pa in (a) to $2.25 \times 10^{-3}$ Pa in (b). Note: color bars differ between (a) and (b). QDP significantly reduces errors across all regions.

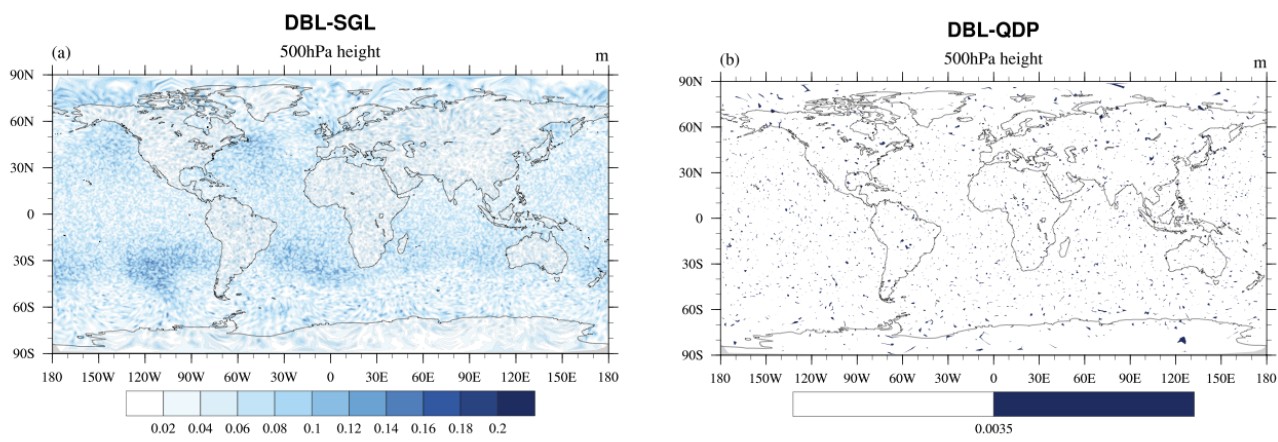

**Figure 11.** Spatial distributions of averaged (1–15 days) 500 hPa height differences (m) between DBL and (a) SGL, (b) QDP (domain size: 120 km $\times$ 120 km). The RMSE decreases from $4.35 \times 10^{-3}$ m in (a) to $1.90 \times 10^{-3}$ m in (b), consistent with the overall improvement shown in Fig. 10.

**3.5 Computational performance**

In comparison with the SGL, although there is a slight increase in runtime, it is minimal, at only 6.0% (JW wave), 0.3% (SC), 2.2% (RD-120) and 17.8% (RD-240) (Table 3). This slight increase is attributed to the addition of a small number of global

variable arrays when using QDP algorithm. And compared to DBL, QDP demonstrated relatively better performance across different cases, reducing the runtime by 28.6% (JW wave), 28.5% (SC), 21.1% (RD-120) and 5.7% (RD-240) (Table 5).

**Table 5.** Comparative Analysis of Computational Efficiency: DBL vs SGL vs **QDP**

| Case name | DBL | SGL | QDP (Proposed) | | |
|---|---|---|---|---|---|
| | Runtime | Runtime | Runtime | vs DBL | vs SGL |
| JW wave | 1768 s | 1191 s | **1263 s** | -28.6% | +6.0% |
| SC | 1507 s | 1073 s | **1077 s** | -28.5% | +0.3% |
| RD-120 | 19126 s | 14765 s | **15092 s** | -21.1% | +2.2% |
| RD-240 | 1397 s | 1118 s | **1317 s** | -5.7% | +17.8% |

Note: JW wave = Jablonowski & Williamson baroclinic wave;

SC = Super-cell; RD-120/240 = Real data with total domain size of 120/240 km

## 4   Conclusions and discussion

This study aims to demonstrate the potential application of QDP algorithm in numerical models. Although QDP algorithm have been widely used for time integration of ordinary differential equations, their application in real-world numerical models has yet to be explored. This research bridges this gap by introducing a novel implementation of the QDP algorithm, thus extending its scope and potential impact in the field of numerical forecasting. As QDP algorithm manages round-off errors through corrections for both large and small numbers, it aligns with numerical models where basic fields significantly exceed tendency fields. Therefore, we have developed a strategy to apply the QDP algorithm in the MPAS-A model. In this study, the application of QDP algorithm in the single-precision version reduced surface pressure errors by 68%, 75%, 97%, and 96% in four cases, respectively, while the runtime decreased by 28.6%, 28.5%, 21.1%, and 5.7% compared to the double-precision experiments (see Section 3). Overall, the QDP algorithm achieves a balance between maintaining accuracy and reducing computational cost.

However, the application of the QDP algorithm has certain limitations. First, the algorithm relies on the iterative process of time integration, and its effectiveness depends on the number of iterations; generally, increasing the number of iterations enhances the error compensation. Second, although the application of the QDP mitigates errors to some extent, it still exhibits errors when compared to DBL, making it less suitable for situations that require high precision. Furthermore, the application of the QDP algorithm necessitates the introduction of additional variables, which increases the complexity to some degree.

Furthermore, whether the QDP algorithm is applicable to the spatial discretization process requires further investigation. Although floating-point operations such as addition, multiplication, and division are often performed multiple times during spatial discretization, which can introduce round-off errors, these errors do not accumulate and amplify as they do in time integration. This is primarily because spatial computations generally do not involve the repetitive time accumulation process. Additionally, the errors in spatial discretization mainly stem from discretization errors (such as grid resolution) and the choice

of discretization methods (e.g., central difference, forward difference), which differ from round-off errors and are primarily related to the discretization method and grid design. Therefore, the effectiveness of QDP algorithm in this context may not be as pronounced as it is in time integration. As a result, this study does not apply QDP algorithm to the spatial discretization process.

In large-scale numerical simulations, the impact of round-off errors cannot be ignored, which is why double precision is commonly employed to maintain the accuracy of results. However, double-precision computations significantly increase computation time and resource consumption, which is often impractical for large-scale simulations. By using QDP algorithm, we not only reduced memory and communication overhead but also enhanced scalability, particularly in ultra-large parallel simulations where inter-node communication can become a major performance bottleneck. Moreover, while our experiments

did not include vectorization strategies, there is considerable potential for further performance improvements.

    In future work, we plan to explore vectorization strategies for QDP algorithm, building on successful implementations of vectorized compensated summation algorithms. Dmitruk et al. (2023) have efficiently vectorized using Intel AVX-512 intrinsics, with parallelization handled through OpenMP constructs. Numerical experiments have shown that the vectorized summation algorithm achieves performance comparable to traditional summation algorithms, especially for large problem sizes, while

maintaining high accuracy. So we intend to apply similar vectorization techniques to QDP algorithm in numerical models, utilizing Single Instruction Multiple Data(SIMD) extensions in modern multicore processors to accelerate the computation of compensated summation and other time-stepping algorithms. Future implementations will also include parallelization via easy-to-use constructs like OpenMP's "declare reduction," which can further speed up execution, especially for large-scale problems. However, for smaller problem sizes or when summation is part of a more complex computation, we may find paral-

lelization to be beneficial even at smaller scales. By incorporating these vectorization and parallelization strategies, we aim to significantly enhance the efficiency and accuracy of QDP algorithm in HPC environments.

    Looking ahead, we plan to extend the application of the QDP algorithm to additional components of the MPAS-A model. Currently, the QDP algorithm is implemented only within the time integration scheme of the dynamic core, with no consideration for its application to tracer transport. Tracer transport processes involve numerous operations where both large and small

values are added together, making them particularly sensitive to precision requirements. In addition, the QDP algorithm has been applied to ideal and real data tests at low and medium resolutions, while its performance at high resolutions has not yet been studied. In future work, we will focus to these fileds.

*Code and data availability.* The provided repository includes all relevant code necessary for the study, categorized into four main components:

**1. Download Model Source Code:** (This includes source code of MPAS-v8.2.1 for different simulation modes)

    – DBL and SGL:

      – GitHub: https://github.com/MPAS-Dev/MPAS-Model/releases/tag/v8.2.1 (last access: 26 December 2024)

- *Note:* If the source code is obtained via the official GitHub repository of the MPAS model, to build a dycore-only MPAS-A model, users need to comment-out or delete the definition of PHYSICS in the Makefile located in the `src/core_atmosphere/`, e.g., `# PHYSICS=-DDO_PHYSICS`.

- *Note:* The source code for SGL is identical to DBL. The difference lies in the compilation process, where a specific compilation option is used to enable single-precision execution. To compile the model in single precision, simply add the `PRECISION=single` flag during the build process.

    – Zenodo: https://doi.org/10.5281/zenodo.14576893 (located in `code_and_data/model/DBL/` and `code_and_data/model/SGL/`)

    – QDP (proposed):

        – Zenodo: https://doi.org/10.5281/zenodo.14576893 (located in `code_and_data/model/QDP/`)

        – *Note:* Add the `PRECISION=single` flag during the build process.

**2. Compile Model Source Code:** (This step includes generating the executable files for `init_atmosphere` and `atmosphere`)

    – Compile `init_atmosphere`: Use the following command: make ifort CORE=init_atmosphere

    – Clean previous builds for `atmosphere`: (if necessary) Use the following command: make clean CORE=atmosphere

    – Compile `atmosphere`: Use the following command: make ifort CORE=atmosphere

    – Compile in Single Precision: To compile the model in single precision, add the `PRECISION=single` flag: make ifort CORE=atmosphere PRECISION=single

**3. Case Setup and Run:** Specific case setups and configurations used in the experiments are provided, including input files, namelist configurations, and scripts for running idealized and real-world scenarios. These allow users to replicate the exact experiments conducted in the study. The steps are as follows:

    – Download the archive file for the test cases, which includes mesh files, decomposition files, and the namelist file, from the official MPAS website at http://mpas-dev.github.io/ (The idealized test cases currently available on the official website include the Supercell, Mountain-Wave, and Jablonowski and Williamson Baroclinic Wave). These can also be downloaded directly at https://doi.org/10.5281/zenodo.14576893 (located in `code_and_data/test/`).

    – Link the `init_atmosphere` and `atmosphere` executable files, compiled in the first part, to the case folder.

    – If the code is downloaded directly from Zenodo, users can run the cases by following the instructions in the README file or directly executing the `run.sh` script. Before running the simulations, ensure to adjust the number of nodes in the script according to the available computational resources to optimize performance.

**4. Visualization and Post-Processing Code:** In order to reproduce the figures in this paper, follow the instructions below:

    – Figure 3 and 4: Run the NCL script by `ncl time.ncl` and `ncl spatical.ncl`. Navigate to the directory: `code_and_data/test/c2_DBL/`.

- Figure 5 and 6: Run the NCL script by `ncl time.ncl` and `ncl spatical.ncl`. Navigate to the directory: `code_and_data/test/c5_DBL/`.

- Figure 7a, 8, and 9: Run the NCL script by `ncl time.ncl` and `ncl spatical.ncl`. Navigate to the directory: `code_and_data/test/c7_240km_DBL/`.

- Figure 7b, 10, and 11: Run the NCL script by `ncl time.ncl` and `ncl spatical.ncl`. Navigate to the directory: `code_and_data/test/c7_120km_DBL/`.

*Author contributions.* JYL and LNW developed the code of applying quasi double-precision algorithm to MPAS-A and design structure of
400 the manuscript. JYL carried out the simulations and analyzed the results with help from LNW, YZY, FW, QZW and HQC. All authors gave comments and contributed to the development of the paper.

*Competing interests.* The authors declare that they have no conflict of interest.

ther geographical representation in this paper. While Copernicus Publications makes every effort to include
appropriate place names, the final responsibility lies with the authors.

*Acknowledgements.* The authors would like to thank the administrator of Beijing Normal University High Performance Computing for providing the high-performance computing (HPC) environment and technical support. The research work presented in this paper was supported by the National Key Research and Development Program of China (grant no. 2023YFB3002405) and the National Natural Science Foundation of China (grant no. 42375162).

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
