# Peer review of "Enhancing Single-Precision with Quasi Double-Precision: Achieving Double-Precision Accuracy in the Model for Prediction Across Scales-Atmosphere (MPAS-A) version 8.2.1"

_EGUsphere, 2024_

## Author Comment (AC1)

**Response to Reviewer 1**

We are incredibly grateful for your efficient review process, providing your feedback in just a few days. Your insightful comments have provided valuable guidance for revising the manuscript. We have revised the manuscript according to your suggestions and will respond to your comments paragraph by paragraph. The comments are given below in black, our responses are in blue, and proposed changes to the manuscript are in red. Additional references are provided at the end of this document. The final revisions and specific locations corresponding to the manuscript will be marked uniformly after receiving feedback from other reviewers.

**Major comments:**

Based on Figure 1 and Figure 2, this computational method appears somewhat simple. There is relatively little research on the keyword "quasi double-precision." The Authors could explain why their work is novel compared to the existed methods already published nowadays.

Thank you for your comments. We have added the significance and novelty of our work in section: introduction and section: Conclusions and discussion and have been summarized below.

Introduction

Most works involving numerical models that reduce numerical precision adopt a mixed-precision scheme, where some variables use single precision while others remain in double precision to ensure integration stability, as demonstrated in the work of Chen et al. (2024). Currently, there are very few studies that almost entirely employ low precision (32-bit) in numerical models, only applied in IFS by Vánˇa et al. (2016). However, they only utilize single precision without considering error compensation for it. In this study, all variables in the numerical model were implemented using single precision, and error compensation was applied to key variables. By using error compensation methods (quasi double-precision), we can maintain integration stability comparable to that applying double precision scheme while significantly reducing memory requirements by lowering the numerical precision of all variables and improved the accuracy comparable to that applying the single precision. This approach not only reduces communication pressure but also allows for substantial increases in computational speed through vectorization optimization.

Conclusion and discussion

We have replaced the first and second paragraphs of the Conclusion section with the following revised paragraph:

The quasi double-precision algorithm can compensate for round-off errors by keeping corrections in addition of large and small numbers. And in numerical models, the basic field is generally much larger than the tends, which aligns with the principles of quasi double-precision, as well as the time integration process. Based on the it, we have established a strategy for applying the quasi double-precision algorithm within the MPAS-A. Through the implementation of quasi double-precision methods, we maintain accuracy similarly to the tests using double precision and achieve comparable integration stability to the tests comparing to single precision tests. The error of surface pressure of

4 cases are reduced by 68%, 75%, 97%, 96% (see Section 3). Overall, QDP using quasi double-precision algorithm demonstrates higher accuracy than the SGL, suggesting the potential for applying quasi double-precision algorithm in numerical models.

We don't apply it to spatial discretization process, because spatial discretization primarily involves subtraction, specifically the subtraction of a small number from a large number or the subtraction of two close values. Whether this algorithm is applicable in spatial discretization remains uncertain, therefore, we don't consider it in this context.

While mixed-precision approaches, where certain variables retain double precision for stability (e.g., Chen et al., 2024), are common for reducing numerical precision in models, and they don't consider the error compensation. This study distinguishes itself by implementing single precision for all model variables and applying error compensation for critical variables.

When applied the quasi double-precision algorithm in MPAS-A, we achieved to reduce all double precision to single precision, although increased few local variables and arrays in every time-integration variable, these have little impact on the overall memory reduction. In general, memory has been reduced by almost half, with a corresponding computational increase of just 6%, 0.3%, 2%, and 18% in the respective cases (see section 3.4), demonstrating a substantial improvement in computational efficiency.

The main point of applying signal precision computing methods is ensuring predictive, and reducing computational costs. The iterative precision compensation increases the computation load. Has this study considered the issue of computational efficiency? For example, runtime, reduced computational cost, they were mentioned in introduction literature review, but not studied in this study.

We apologize for the oversight. The additional computational cost incurred from using error compensation is minimal; it only adds six global arrays to the entire model code. Therefore, we neglected to elaborate on this aspect. We appreciate your suggestion, following your insightful comment, we found the previously reported figures were indeed rough estimations, so we have re-evaluated the exact computational performance using a measurement tool to determine the runtimes. So, we will:

1. Add a dedicated section 3.4 in the paper to describe the computational performance. The size of the computational performance will be represented in terms of runtime, and we will discuss the runtime for each case in tabular form.
2. Revise the description of computational performance in the abstract to reflect these updated and more accurate measurements.

3.4 Computational performance

In comparison with the SGL, although there is a slight increase in runtime, it is minimal, at only 6% (Jablonowski and Williamson baroclinic wave), 0.3% (Super-cell), 2% (Real data with resolution of 120km) and 18% (Real data with resolution of 240km) (Table 1). This slight increase is attributed to the addition of a small number of global variable arrays when using quasi double-precision. And compared to DBL, QDP demonstrated relatively better performance across different cases, reducing

the runtime by 29% (Jablonowski and Williamson baroclinic wave), 29% (Super-cell), 21% (Real data with resolution of 120km) and 6% (Real data with resolution of 240km) (Table 1).

**Table 1**. Elapsed time of DBL, SGL and QDP test (unit:s).

| Case name | DBL | SGL | QDP |
|---|---|---|---|
| Jablonowski and Williamson baroclinic wave | 1768 | 1191 | 1263 |
| Super-cell | 1507 | 1073 | 1077 |
| Real data with resolution of 120km | 19126 | 14765 | 15092 |
| Real data with resolution of 240km | 1397 | 1118 | 1317 |

Abstract

The content 'The round-off error of surface pressure is reduced by 68%, 75%, 97%, 96% in cases, the memory has been reduced by almost half, while the computation increases only 2%, significantly reducing computational cost.' will be revised to 'The bias of surface pressure are reduced respectively by 68%, 75%, 97% and 96% in cases, the memory has been reduced by almost half, while the computation increases only 6%, 0.3%, 2%, and 18% in cases, significantly reducing computational cost.'

The model is described poorly. The solution method for the equations is not even mentioned. For example, the finite difference scheme is mainly used to calculate the primary equations for variables studied in this work. And at which step of the equation is the quasi double-precision algorithm specifically applied? The strategy used to compute cell edge, dry air density, potential temperature with quasi double-precision algorithm is also difficult to understand from Figure 3.

Thank you for your valuable feedback regarding the clarity of our model description and the implementation of the quasi double-precision algorithm. We recognize that the solution method for the equations was not sufficiently detailed in our original manuscript. To address this, we will:

1. Add the clear description of the solution method for the equations including temporal integration scheme and spatial discretization scheme in section 2.2. Moreover, the algorithm applied here primarily addresses the rounding error compensation between large and small numbers in addition. Currently, it is only applicable to the time integration process and has not been implemented in the spatial discretization process. I apologize for not mentioning that part in the manuscript. I will revise and provide the supplement. See 2.2 additional content below.
2. Specify at which step the quasi double-precision algorithm is applied within the computation process in section 2.3 and replace the figure (corresponding to Figure 3 in the manuscript) and explain this process using formulas and explanations. See 2.3 additional content below.

2.2 Additional content

The MPAS-A solves the fully compressible, nonhydrostatic equations of motion (Skamarock et al. 2012). The spatial discretization uses a horizontal (spherical) centroidal Voronoi mesh with a terrain-following geometric-height vertical coordinate and C-grid staggering for momentum. The temporal discretization uses the explicit time-split Runge–Kutta technique from Wicker and Skamarock (2002) and Klemp et al. (2007).

The algorithm applied here primarily addresses the rounding error compensation between large and small numbers in addition. Currently, it is only applicable to the time integration process and has not been implemented in the spatial discretization process. Therefore, this section will provide a detailed introduction to the time integration scheme. For the spatial discretization scheme, please refer to Skamarock et al. (2012), and it will not be introduced upon here.

The formulation of the scheme can be considered in on dimension as equation Wicker and Skamarock (2002):

$$\frac{\partial \emptyset}{\partial t} = RHS_\emptyset \tag{1}$$

The variable $\emptyset$ represents any prognostic variable in the prognostic equations, while RHS represents the right-hand side of the prognostic equations (i.e., the spatial discretization equation). In MPAS-A, a forward-in-time finite difference is used, and it can be written as Eq. (2):

$$\frac{{\emptyset_i}^{n+1} - {\emptyset_i}^n}{\Delta t} = RHS_\emptyset \tag{2}$$

Where superscript represent the time step, and subscript represent the position of grid zone.

The two-order Runge-Kutta time scheme is used in MPAS-A as described in Gear et al. (1971):

$$\emptyset^* = \emptyset^t + \frac{\Delta t}{2} * RHS(\emptyset^t) \tag{3}$$

$$\emptyset^{**} = \emptyset^t + \frac{\Delta t}{2} * RHS(\emptyset^*) \tag{4}$$

$$\emptyset^{t+\Delta t} = \emptyset^t + \Delta t * RHS(\emptyset^{**}) \tag{5}$$

**2.3 Additional content**

According to Equation Eq. (3), (4) and (5), it can be observed that in the time integration scheme, each step involves the process of adding tends on the basic field $\emptyset^t$. In numerical models, the basic field is generally much larger than the tends, which aligns with the principles of numerical computation regarding the addition of large and small numbers, as well as the time integration process. It is important to note that the quasi double-precision algorithm currently only addresses time integration and has not been validated during the spatial discretization process. The spatial discretization primarily involves subtraction, specifically the subtraction of a small number from a large number or the subtraction of two close values. Whether this algorithm is applicable in spatial discretization remains uncertain, therefore, we will not apply it in this context.

Based on the application principles of the algorithm, which involve the processes of adding large and small numbers as well as the time integration process, we have established a strategy for applying the quasi double-precision algorithm within the MPAS-A. Specific improvements are provided based on the predictive equations:

$$\frac{\partial \boldsymbol{V}_H}{\partial t} = -\frac{\rho_d}{\rho_m}\left[\nabla_\zeta\left(\frac{p}{\zeta_z}\right) - \frac{\partial z_H p}{\partial \zeta}\right] - \eta \boldsymbol{k} \times \boldsymbol{V}_H - \boldsymbol{v}_H \nabla_\zeta \cdot \boldsymbol{V} - \frac{\partial \Omega \boldsymbol{v}_H}{\partial \zeta} - \rho_d \nabla_\zeta K - eW cos\alpha_r - $$

$$\frac{\boldsymbol{v}_H W}{r_e} + \boldsymbol{F}_{\boldsymbol{V}_H} \tag{6}$$

$$\frac{\partial W}{\partial t} = -\frac{\rho_d}{\rho_m}\left[\frac{\partial p}{\partial \zeta} + g\tilde{\rho}_m\right] - (\nabla \cdot \boldsymbol{v}W)_\zeta + \frac{uU + vV}{r_e} + e(Ucos\alpha_r - Vsin\alpha_r) + F_W \tag{7}$$

$$\frac{\partial \Theta_m}{\partial t} = -(\nabla \cdot \boldsymbol{V}\theta_m)_\zeta + F_{\Theta_m} \tag{8}$$

$$\frac{\partial \tilde{\rho}_d}{\partial t} = -(\nabla \cdot \boldsymbol{V})_\zeta \tag{9}$$

The meaning of each variable in the equations exactly follows Skamarock et al. (2012), so that we don't repeating explanation. For a numerical model, the most crucial variables are the prognostic variables. Therefore, In the MPAS-A model we applied the quasi double-precision algorithm to the time integration process of these prognostic variables, including horizontal momentum ($\boldsymbol{V}_H$), dry air density ($\tilde{\rho}_d$), potential temperature ($\Theta_m$) and vertical velocity ($W$), that is , the process in red of Eq. (6), (7), (8) and (9). (Only the predictive equations for the dynamic core are presented here, without the scalar transport.)

The color scheme in Figure 5 is very hard to distinguish; the solid purple area is too large, making the gradient difficult to see. Figure 10 has the same issue. It is recommended to refer to the classic NCL color scheme. https://www.ncl.ucar.edu/Applications/era40.shtml

Thank you very much for your suggestions and the website link. I have revised Figures 5 and 10 according to the guidelines provided in the link. Please see below for Figure 1 (corresponding to Figure 5 in the manuscript) and Figure 2 (corresponding to Figure 10 in the manuscript).

[Figure]

**Figure 1.** Spatial distributions of averaged (1-15days) difference of surface pressure (units: Pa) between DBL and (a) SGL simulations, (b) QDP simulations (round-off error has reduced) in case of Jablonowski and Williamson baroclinic wave.

[Figure]

**Figure 2.** Spatial distributions of averaged (1-15days) difference of 500hPa height (units: m) between DBL and (a) SGL simulation, (b) QDP simulation (resolution: 240 km × 240 km). The RMSE of 500hPa height between DBL and (a) SGL simulation is $2.80 \times 10^{-1}$ m, (b) QDP simulation is $1.40 \times 10^{-1}$ m (round-off error has reduced).

Figure 6 (a) shows that DBL-SGL decreases after 1.0, which appears to be caused by a coding error. Please check and confirm the validity of the data.

Thank you for your insightful comment regarding Figure 6(a), we carefully reviewed our code based on your comments and identified that the issue was a problem with the data processing and plotting. We have corrected this issue, and the revised Figure 3 is provided below:

[Figure]

**Figure 3.** The temporal evolution of spatially averaged difference of kinetic energy between DBL and SGL, as well as difference between DBL and QDP in case of super-cell.

I apologize for needing to provide additional information regarding this section. In all cases presented in the manuscript, the time-evolution plots (Figures 4, 6, and 8 in the manuscript) currently utilize kinetic energy and surface pressure, which correspond to the conservation of energy and mass, respectively. However, I realize that directly using total energy and total mass would offer a more accurate representation. Please see the figures 4, 5 and 6 showing the temporal variations of energy and mass for all cases. If you allow it, I would be happy to replace the existing Figures 4, 6, and 8 (in the manuscript) with these updated versions (figures 4, 5 and 6). I want to emphasize that this change would not affect the overall results or conclusions of the study, which remain consistent with those currently presented in the manuscript. I think your suggestions are greatly appreciated and have been very helpful in improving the manuscript.

[Figure]

**Figure 4.** The temporal evolution of spatially averaged difference of (a) total energy, (b) total mass between DBL and SGL, as well as difference between DBL and QDP in case of Jablonowski and Williamson baroclinic wave.

[Figure]

**Figure 5.** The temporal evolution of spatially averaged difference of (a) total energy, (b) total mass between DBL and SGL, as well as difference between DBL and QDP in case of super-cell.

It is important to note that, for enhanced clarity and to facilitate a better understanding of the trend, the x-axis unit for the figure representing Case 7 has been changed to hours (Fig. 6).

[Figure]

**Figure 6.** The temporal evolution of spatially averaged difference of total energy between DBL and SGL, as well as difference between DBL and QDP in case of real data, with resolution of (a) 240 km × 240 km, (b) 120 km × 120 km.

**Minor comments:**

The color bar in Figure 7 seems almost useless.

Thank you very much for your careful review and valuable comments on my manuscript. Regarding the issue you raised about the color bar in the figure 7. After adjusting the color bar, two distinct areas of improvement are now evident, compared to only one in the original figure. I sincerely apologize that the previous color bar setting was not appropriate. I have adjusted the color bar

according to your suggestion, and the revised figure is shown as Figure 7 (corresponding to Figure 7 in the manuscript).

[Figure]

**Figure 7.** Perturbation theta in super-cell development at 5400s in the (a) DBL simulation, (b) SGL simulation and (c) QDP simulation (bias has reduced), unit: K, the circle represents the clearest error.

Please mention "round-off error" in relevant figures' captions.

Thank you very much for your suggestions. I have revised it. You can see it in the manuscript after receiving feedback from other reviewers.

Figure 11 appears to be somewhat blurry.

Thank you very much for your suggestions. I have revised it. Please see below for Figure 8 (corresponding to Figure 11 in the manuscript).

[Figure]

**Figure 8.** distributions of averaged (1-15days) difference of surface pressure (units: Pa) between DBL and (a) SGL simulation, (b) QDP simulation (resolution: 120 km × 120 km) (round-off error has reduced). The RMSE of surface pressure between DBL and (a) SGL simulation is $6.33 \times 10^{-2}$ Pa, (b) QDP simulation is $2.25 \times 10^{-2}$ Pa.

It's important to note that the color bars for Figures (a) and (b) are intentionally set differently due to the significant disparity in their respective threshold ranges. If identical color scales were applied, Figure (b) would appear entirely white.

Wicker, L. J., & Skamarock, W. C.: Time-splitting methods for elastic models using forward time schemes. Monthly Weather Review, 130(8), 2088. https://www.proquest.com/scholarly-journals/time-splitting-methods-elastic-models-using/docview/198148677/se-2, 2002.

Gear, C. W.: Numerical initial value problems in ordinary differential equations[M]. Englewood Cliffs, N.J: Prentice-Hall, 1971.

---

## Author Comment (AC2)

**Response to Reviewer 2**

Thank you very much for your prompt and insightful review. We have carefully considered your suggestions and implemented further revisions to the manuscript. Which has significantly improved the quality of the manuscript. We have revised the manuscript according to your suggestions and will respond to your comments paragraph by paragraph. The comments are given below in black, our responses are in blue, and proposed changes to the manuscript are in red. Additional references are provided at the end of this document. The final revisions and specific locations corresponding to the manuscript will be marked uniformly after receiving feedback from other reviewers.

**Response to Point 1:**

The calculation method used in this paper is based on the method of Møller, O. (1965)., which appears relatively simple. If the authors can find similar studies that employ this "quasi double-precision" method for simulating atmospheric variables, it could demonstrate the advancement of this paper compared to existing research. Chen et al. (2024) and Ván̆a et al. (2016) did not use this method. If no similar studies are found, it could be stated that this paper is the first to do so, which could enhance its citation rate.

Thank you very much for your insightful comment. We greatly appreciate your valuable suggestion, which has significantly improved the manuscript. Following your suggestion, we have added a statement highlighting this novel application in both the "Introduction" and "Conclusion and Discussion" sections of the manuscript. We believe this addition provides important context for our work and clarifies its contribution to the field.

Introduction

The method of Moller (Quasi double-precision) has primarily been applied to the time integration of ordinary differential equations, as demonstrated in studies such as Thompson et al. (1970), Tomonori et al. (1995), and Dmitruk et al. (2023). The application of the Moller method to a realistic numerical model, as presented in this study, represents a novel contribution to the field, with no prior research exploring this specific implementation.

Conclusion and discussion

Although the Moller method (Quasi double-precision) has been extensively employed for the temporal integration of ordinary differential equations, its application within the context of realistic numerical models remains unexplored. This study addresses this gap by presenting a novel implementation of the Moller method, thereby expanding its scope and potential impact within the field.

**Response to Point 3:**

Regarding equations (6)-(9), it is suggested to select only one equation and present it in the form of Figure 2. This could be included as an appendix to specifically demonstrate the iterative calculation process. Because the pseudo-code in Fig. 3 is not sufficient to clearly explain the calculation process.

Thank you very much for your valuable suggestion. We appreciate your feedback and agree that providing an algorithmic description of the iterative process for one of the equations would enhance the clarity of our work. As your recommendation, we will select a representative equation and include a detailed algorithmic description of its iterative calculation process in the Supplement (see Figure 1). We believe this addition will be beneficial for readers who wish to gain a deeper understanding of the computational aspects of our method.

Supplement:

[Figure]

**Figure 1.** The pseudo-code for variable of U.

**Response to Point 5:**

If the authors intend to replace the existing Figures 4, 6, and 8 (in the manuscript) with these updated versions (Figures 4, 5, and 6), feedback from other editors and reviewers regarding the revised structure of the paper should be considered.

Thank you very much for your valuable suggestion. I appreciate your careful attention to detail. I will ensure that I obtain the consent of each reviewer and the editor before proceeding with the replacement of the figures, as you recommended.

**The new figures:**

There are several cases where figure legends and lines overlap, such as in Figure 3 in the review response.

Thank you very much for your valuable suggestion. We have completed the revisions as you recommended. The updated figures are as follows: Figure 2 (corresponding to Figure 3 in RC1) and Figure 3 (corresponding to Figure 5 in RC1).

[Figure]

**Figure 2.** The temporal evolution of spatially averaged difference of kinetic energy between DBL and SGL, as well as difference between DBL and QDP in case of super-cell.

[Figure]

**Figure 3.** The temporal evolution of spatially averaged difference of (a) total energy, (b) total mass between DBL and SGL, as well as difference between DBL and QDP in case of super-cell.

Thompson., Robert, J.:Improving round-off in Runge-Kutta computations with Gill's method, Communications of the Acm, 13(12):739- 740.DOI:10.1145/362814.362823, 1970.

Tomonori, Kouya., Hideko, Nagasaka.: On the correction method of round-off errors in the Yang's Runge-Kutta method, The Japan Society for Industrial and Applied Methematics, 1995.

Dmitruk, B., Przemysaw, Stpiczyn´ski.:Improving accuracy of summation using parallel vectorized Kahan's and Gill-Mller algorithms, Con- currency and Computation: Practice and Experience, doi:10.1002/cpe.7763, 2023.

---

## Author Comment (AC3)

**Response to Reviewer 3**

We greatly appreciate your timely and insightful review. Your comments have been instrumental in guiding our revisions. We provide a detailed point-by-point response below, using black for your comments, blue for our responses, and red for proposed changes to the manuscript. We will finalize the manuscript and indicate specific locations of changes after receiving all reviews. Additional references are included at the end of this document.

The manuscript titled "Enhancing Single-Precision with Quasi Double-Precision: Achieving Double-Precision Accuracy in the Model for Prediction Across Scales-Atmosphere (MPAS-A) version 8.2.1" explores the use of the quasi-double precision algorithm by Møller et al. (1965) within a numerical modelling framework. By implementing the algorithm in MPAS-A across various case studies and idealized scenarios, the study highlights its potential to lower computational costs compared to double-precision approaches while enhancing accuracy over single-precision approaches.

Overall, the manuscript is well-organized and relatively clear, but there are several areas that could benefit from further clarification and elaboration. In particular, I have concerns regarding 1) the practical significance of the reported improvements, 2) the lack of detail and rationale in the model setup, and 3) the insufficient explanation or justification for the choice of metrics. Based on these issues, I recommend major revisions before the manuscript can be reconsidered for publication.

Thank you for your thorough review and valuable feedback on our manuscript, we appreciate your positive comments on the organization and clarity of the manuscript.

We understand your concerns regarding the practical significance of the reported improvements, we have carefully considered your feedback and addressed each of these points in the detailed responses below. We believe the revisions made in response to your comments significantly strengthen the manuscript and clarify the contributions of our work.

We are grateful for your insights and believe that addressing your concerns has resulted in a substantially improved manuscript. We look forward to your further assessment.

**Below, you find a list of my main concerns:**

1. The practical impact of the improvements achieved with the quasi-double precision algorithm compared to the single-precision approach remains unclear to me. While the authors demonstrate large relative accuracy gains compared to single-precision, it is not evident that these improvements are meaningful in absolute terms. For example, in the idealized scenario of Section 3.1, the RMSE of surface pressure improves by only $10^{-2}$ Pa at longer lead times (12+ days), with negligible improvements at shorter times. Similar patterns emerge in Section 3.3. Given that these improvements are an order of magnitude smaller than the World Meteorological Organization's (WMO) recommended measurement uncertainty for atmospheric pressure (e.g., https://gcos.wmo.int/en/essential-climate-variables/pressure), it might be difficult to justify the additional computational costs. I encourage the authors to elaborate on the significance of their results in absolute terms from both a practical and forecasting perspective.

A related point: Are current operational implementations of MPAS-A defaulting to double- or single-precision for the variables discussed in this manuscript? Given the results, if double precision is currently the standard, the authors might want to emphasise the potential of quasi-double precision as a substitute for double precision, rather than focusing primarily on its benefits over single-precision.

Thank you for your insightful comments and suggestions. We appreciate your careful review and have revised the manuscript accordingly. We address your points as follows:

(1) This research is based on double precision testing as the benchmark experiment, we agree that highlighting the computational advantages of quasi-double precision compared to double precision is crucial. We apologize for the omission of this discussion in the original manuscript. As you suggested, we have now added a detailed analysis of computational costs in the revised manuscript, comparing SGL, DBL and QDP. This analysis demonstrates that QDP achieves accuracy very close to DBL while significantly reducing computational costs. We provide further details on this analysis in our response to your second point below.

(2) You correctly pointed out that the SGL errors already meet the WMO's target values, and our primary focus should be the comparison between QDP and DBL. We included the comparison with SGL for the following reasons: within the MPAS-A model, the quasi double-precision algorithm is applied only to the basic and tend processes, while other parts remain in their original single-precision implementation. In essence, the quasi double-precision algorithm is applied to a limited portion of the dynamical core. Since MPAS-A supports single-precision calculations, we aimed to investigate the impact of the quasi double-precision algorithm on both computational cost and accuracy relative to the existing single-precision implementation. A modest increase in computational cost coupled with a substantial improvement in accuracy would suggest the potential of quasi double-precision as a replacement for single precision in scenarios where storage limitations are a major concern. We have retained the comparison to SGL to provide this context but have strengthened the emphasis on the comparison with DBL throughout the manuscript as you recommended.

(3) You correctly pointed out that while the errors observed within the current 12+day simulations appear relatively small, these could accumulate over longer time scales. Therefore, we plan to conduct extended simulations (monthly to yearly) in future research to investigate the potential for error accumulation and to determine the feasibility of using quasi-double precision as a replacement for double precision in such long-term applications. As you suggested, our future work will prioritize direct comparisons between the QDP and DBL in these extended simulations to fully characterize the differences and evaluate the long-term viability of our proposed method.

2. The experimental setup lacks sufficient detail and justification. It is unclear why the authors chose these particular case studies and idealised scenarios, especially in the applications to real data. This makes it difficult to assess the generalisability of the results to other atmospheric conditions or variables. Additionally, a more detailed comparison of computational costs among the three algorithms (single-precision, quasi-double precision, and double precision) is necessary, as

improvements in accuracy may not be feasible in practice if they come with substantial increases in computational expense.

Thank you for your constructive feedback. We address your concerns regarding the case selection and computational cost as follows:

(1)Case Selection and Justification: We apologize that this rationale was not clearly articulated in the original manuscript. We will explain it as follows in Section 2.4. Thank you for your suggestions. We would like to ask whether you believe the descriptions and tables mentioned should be added to the main text of the manuscript. We look forward to your further input on this matter.

(2)Computational cost: We will add a dedicated section 3.4 in the paper to describe the computational performance. The size of the computational performance will be represented in terms of runtime, and we will discuss the runtime for each case in tabular form.

**2.4 Additional content**

The type of initial conditions in MPAS-A cases (Table 1) can be found in "MPAS-Atmosphere Model User's Guide Version 8.1.0" (can be download on MPAS website). Only configurations for the idealized cases 2, 5, and 6 are provided on the official website. To avoid redundancy, the results for Case 6 are not shown, as they were similar to those obtained for Cases 2 and 5. For real-data simulations, case 7 as shown in Table. We explored different resolutions for real-data simulations to assess the impact of grid refinement on the performance of the quasi-double precision algorithm. Our focus is on numerical precision in global models, where boundary condition effects are absent. Thus, we do not consider regional models like case 8 and case 9 in this study.

**Table 1**. Cases in MPAS-A.

| type of initial conditions to create: |
| --- |
| 1.  Jablonowski and Williamson baroclinic wave (no initial perturbation) |
| 2.  Jablonowski and Williamson baroclinic wave (with initial perturbation) |
| 3.  Jablonowski and Williamson baroclinic wave (with normal-mode perturbation) |
| 4.  squall line |
| 5.  super-cell |
| 6.  mountain wave |
| 7.  real-data initial conditions from, e.g., GFS |
| 8.  surface field (SST, sea-ice) update file for use with real-data simulations |
| 9.  lateral boundary conditions update file for use with real-data simulations |

3.4 Computational performance

In comparison with the SGL, although there is a slight increase in runtime, it is minimal, at only 6% (Jablonowski and Williamson baroclinic wave), 0.3% (Super-cell), 2% (Real data with resolution of 120km) and 18% (Real data with resolution of 240km) (Table 2). This slight increase is attributed to the addition of a small number of global variable arrays when using quasi double-precision. And compared to DBL, QDP demonstrated relatively better performance across different cases, reducing the runtime by 29% (Jablonowski and Williamson baroclinic wave), 29% (Super-cell), 21% (Real data with resolution of 120km) and 6% (Real data with resolution of 240km) (Table 2).

**Table 2**. Elapsed time of DBL, SGL and QDP test (unit:s).

| Case name | DBL | SGL | QDP |
|---|---|---|---|
| Jablonowski and Williamson baroclinic wave | 1768 | 1191 | 1263 |
| Super-cell | 1507 | 1073 | 1077 |
| Real data with resolution of 120km | 19126 | 14765 | 15092 |
| Real data with resolution of 240km | 1397 | 1118 | 1317 |

Abstract

The content 'The round-off error of surface pressure is reduced by 68%, 75%, 97%, 96% in cases, the memory has been reduced by almost half, while the computation increases only 2%, significantly reducing computational cost.' will be revised to 'The bias of surface pressure are reduced respectively by 68%, 75%, 97% and 96% in cases, the memory has been reduced by almost half, while the computation increases only 6%, 0.3%, 2%, and 18% in cases, significantly reducing computational cost.'

3. The manuscript currently employs only grid-point level and "spatial RMSE" as comparison metrics, but the term "spatial RMSE" is not clearly defined. I recommend the following:

1) Clearly define "spatial RMSE," particularly when used as a summary statistic over a larger domain. For example, are the authors applying any latitude-based weighting?

2) Justify the use of this metric and consider comparing the algorithms using additional relevant metrics. If alternative metrics are not applicable, please explain why they were not used.

Thank you very much for your insightful comments regarding the metrics used in our study. Your feedback has been incredibly helpful in improving the clarity of our analysis. We address your concerns in three parts:

(1) We apologize for the omission of a clear definition of spatial RMSE in the original manuscript. As your suggestion, we have now included a detailed explanation of the spatial RMSE calculation in Section 3.1. This explanation includes the formula and a description of its application in our study, as detailed below.

(2) We selected spatial RMSE as a primary metric because it is widely used in the evaluation of

atmospheric numerical models and effectively reflects model performance improvements. Studies such as Vánˇa et al. (2016), demonstrate its utility in assessing the accuracy of model simulations. Therefore, we believe spatial RMSE is an appropriate and informative metric for evaluating the impact of the quasi-double precision algorithm in our study.

(3) We appreciate your suggestion to include additional metrics. We decide to add a new metrics of Mean Absolute Error (MAE) to improve our analysis in Section 3.1. To facilitate comparison across different cases, we have also included a table summarizing the spatial RMSE and MAE values for each case in the revised manuscript. The specific changes are detailed below.

3.1 Additional content

To quantify the difference between the simulations using SGL, QDP, and DBL, (used as the benchmark), we calculate the spatial root-mean-square error (RMSE). First, for each grid point, the temporal averages of the variables (e.g., surface pressure, 500hPa height) are computed across the entire simulation period for each experiment (SGL, QDP, and DBL). Then, the spatial RMSE is calculated as the root-mean-square difference between the temporally averaged fields of the control experiment (SGL or QDP) and the benchmark double-precision experiment (DBL), following (1):

$$\text{Spatial RMSE} = \sqrt{\frac{\sum_{i=1}^{N}(M_i - C_i)^2}{N}} \tag{1}$$

Where, N is the total number of grid points, $M_i$ is the temporally averaged value at grid point i for the benchmark double-precision experiment, $C_i$ is the temporally averaged value at grid point i for the control experiment (SGL or QDP).

In addition to the spatial RMSE, we also calculate the Mean Absolute Error (MAE) to assess the magnitude of the difference between the control experiments (SGL and QDP) and the benchmark double-precision experiment (DBL), irrespective of the direction of the difference. Like the spatial RMSE calculation, we first compute the temporal average for each grid point across the entire simulation period for each experiment. The MAE is then calculated as the average absolute difference between the temporally averaged fields of the control experiment and the benchmark experiment, following (2):

$$\text{MAE} = \frac{1}{N} * \sum_{i=1}^{N}|M_i - C_i| \tag{2}$$

where N is the total number of grid points, $M_i$ represents the temporally averaged value at grid point i for the benchmark double-precision experiment, and $C_i$ represents the temporally averaged value at grid point i for the control experiment (either SGL or QDP).

Additional table

As shown in Table 4 (MAE) and Table 3 (spatial RMSE), the addition of the quasi-double precision algorithm consistently improves accuracy (compared to single precision) across all cases.

**Table 3**. The spatial RMSE values of surface pressure compared to DBL for cases,

unit: Pa.

| Case name | SGL | QDP |
|---|---|---|
| Jablonowski and Williamson baroclinic wave | $3.42*10^{-2}$ | $1.09*10^{-2}$ |
| Super-cell | $8.80*10^{-4}$ | $2.27*10^{-4}$ |
| Real data with resolution of 120km | $6.33*10^{-2}$ | $2.25*10^{-3}$ |
| Real data with resolution of 240km | $6.68*10^{-2}$ | $2.25*10^{-3}$ |

**Table 4**. The MAE values of surface pressure compared to DBL for cases,

unit: Pa.

| Case name | SGL | QDP |
|---|---|---|
| Jablonowski and Williamson baroclinic wave | $1.29*10^{-2}$ | $3.81*10^{-3}$ |
| Super-cell | $8.79*10^{-4}$ | $2.26*10^{-4}$ |
| Real data with resolution of 120km | $5.38*10^{-2}$ | $1.95*10^{-3}$ |
| Real data with resolution of 240km | $5.52*10^{-2}$ | $1.94*10^{-3}$ |

**Minor concerns/technical corrections:**

- Figure 6a: The higher error with the quasi-double precision algorithm compared to the single-precision in this subpanel is confusing. If the error with the single-precision algorithm is anyway so small that it does not matter in practice, why was this case study selected, and why include this figure?

Thank you for bringing this to our attention. We sincerely apologize for the error in our data processing, which was also noted by Reviewer 1. We have carefully re-examined our code and identified a mistake. We have corrected the error in our data processing code and have now generated the revised results. The corrected figure is shown below (fig. 1), Figure 6(a) in manuscript:

[Figure]

**Figure 1.** The temporal evolution of spatially averaged difference of kinetic energy between DBL and SGL, as well as difference between DBL and QDP in case of super-cell.

I apologize for needing to provide additional information regarding this section. In all cases presented in the manuscript, the time-evolution plots (Figures 4, 6, and 8 in the manuscript) currently utilize kinetic energy and surface pressure, which correspond to the conservation of energy and mass, respectively. However, I realize that directly using total energy and total mass would offer a more accurate representation. Please see the figures 2, 3 and 4 showing the temporal variations of energy and mass for all cases. If you allow it, I would be happy to replace the existing Figures 4, 6, and 8 (in the manuscript) with these updated versions (figures 2, 3 and 4). I want to emphasize that this change would not affect the overall results or conclusions of the study, which remain consistent with those currently presented in the manuscript. We have received feedback from two reviewers so far, and Reviewer 1 has approved the replacement. We are eagerly awaiting your response. If all four reviewers agree, we will update the figure in the final version of the manuscript.

[Figure]

**Figure 2.** The temporal evolution of spatially averaged difference of (a) total energy, (b) total mass between DBL and SGL, as well as difference between DBL and QDP in case of Jablonowski and Williamson baroclinic wave.

[Figure]

**Figure 3.** The temporal evolution of spatially averaged difference of (a) total energy, (b) total mass between DBL and SGL, as well as difference between DBL and QDP in case of super-cell.

It is important to note that, for enhanced clarity and to facilitate a better understanding of the trend, the x-axis unit for the figure representing Case 7 has been changed to hours (Fig. 4).

[Figure]

**Figure 4.** The temporal evolution of spatially averaged difference of total energy between DBL and SGL, as well as difference between DBL and QDP in case of real data, with resolution of (a) 240 km × 240 km, (b) 120 km × 120 km.

- Figure 7: The phrase "the circle represents the most clear error" lacks objectivity, especially since other areas in the figure show similar errors. Consider rephrasing or finding a more precise way to characterize the comparison between errors in this case.

Thank you very much for your careful review and valuable comments on my manuscript. Regarding the issue you raised in the figure 7. We decide to adjust the color bar, after adjusting, the two distinct areas of improvement are now evident, compared to only one in the original figure. I sincerely

apologize that the previous color bar setting was not appropriate. and the revised figure is shown as Figure 5 (corresponding to Figure 7 in the manuscript).

[Figure]

**Figure 5.** Perturbation theta in super-cell development at 5400s in the (a) DBL simulation, (b) SGL simulation and (c) QDP simulation (bias has reduced), unit: K, the circle represents the pattern bias.

- Please avoid using the term "significant" or "significantly" when not referring to statistically significant results to avoid potential confusion.

Thank you for this valuable feedback. We have corrected this oversight, and the revised text will be included in the final manuscript submission after all reviews have been addressed.

- Unclear phrase (line 95): "achieves basically consistent results comparable to those of double precision" – Do you mean that the accuracy is similar in a specific case, or that the quasi-double precision consistently performs similarly to double precision in most cases?

Thank you for raising this point. The phrase 'the single-precision version improved with the quasi double-precision algorithm achieves basically consistent results comparable to those of double precision. However, we acknowledge that the phrasing was ambiguous. We have revised the sentence to more accurately reflect this finding.   The revised sentence now reads:

The quasi-double precision algorithm effectively mitigates the accuracy limitations of single precision, producing results largely consistent with double-precision calculations.

- Reproducibility note: Although the manuscript adheres to GMD's guidelines by providing a DOI to a permanent repository, no instructions are provided on how to use the code or what the various files and subfolders contain. I suggest including at least a brief README file that describes the content and organization of the repository, providing users with clearer guidance.

Thank you for your valuable suggestion regarding the code documentation. We have updated the supplemental material with a comprehensive description of the code structure and included a README file for each case, as you recommended. We apologize that this information was not initially referenced within the main manuscript. To ensure clarity and accessibility, we have also

added explicit mentions of the code description and README files within the Code and data availability section of the manuscript, directing readers to the supplemental material for these details. Figure 6 shows a screenshot of the updated supplemental material, which now clearly includes these additions. Each case directory within the supplement contains a corresponding README file, an example of which is shown in Figure 7.

We believe these revisions significantly improve the transparency and reproducibility of our research. Thank you again for your helpful feedback.

Code and data availability. Model code and plotting data related to this manuscript is available at: https://doi.org/10.5281/zenodo.13765422. Details regarding the code structure and instructions for running the code are provided in the supplementary material, which can be downloaded and viewed in Fig. S1. This figure provides a visual overview of the code organization. The information of steps how to execute the simulations can be found in README file in each test case folder.

**Supplementary Information**

```
model
├──── DBL  --benchmark using double precision
├──── SGL  --control test using single precision
└──── QDP  --control test using single precision and quasi double-precision algorithm

test
├──── c2_DBL--benchmark using double precision. (The Jablonowski and Williamson baroclinic wave test case)
│     ├──── atmosphere_model        -- run the model
│     ├──── bwave_surface_p.ncl      -- script to produce plots of surface pressure and kinetic energy
│     ├──── init_atmosphere_model  --run to create initial conditions
│     ├──── namelist.atmosphere       --namelist options available when running the MPAS
│     ├──── namelist.init_atmosphere --namelist options available when running the MPAS initialization
│     ├──── README
│     ├──── stream_list.atmosphere.output --the output of MPAS
│     ├──── streams.atmosphere --the XML stream configuration file for an MPAS
│     └──── streams.init_atmosphere --The XML stream configuration file for an MPAS initialization
├──── c2_SGL--benchmark using double precision.
├──── c2_QDP--benchmark using double precision.
├──── c5_DBL--benchmark using double precision. (The super cell)
├──── c5_SGL--benchmark using double precision.
├──── c5_QDP--benchmark using double precision.
├──── c7_240km_DBL--benchmark using double precision. (The real data)
├──── c7_240km_SGL--benchmark using double precision.
├──── c7_240km_QDP--benchmark using double precision.
├──── c7_120km_DBL--benchmark using double precision.
├──── c7_120km_SGL--benchmark using double precision.
└──── c7_120km_QDP--benchmark using double precision.
```

**Figure S1.** The code layout of the research. The model part represent the model code including benchmark using double precision(DBL), control test using precision and control test using single precision(SGL) and quasi double-precision algorithm(QDP). The three models are run separately in 4 tests includes the Jablonowski and Williamson baroclinic wave test case, super cell, real data with 240km and real data with 120km. All configurations can be found in the test file. Only the case 7 use the GFS data, it can also be found under folder case7. Model code and plotting data related to this manuscript is available at: https://doi.org/10.5281/zenodo.13765421.

**Figure 6**. The code layout in supplement.

```
The supercell thunderstorm test case.

Steps to run this test case:

1. Link the init_atmosphere_model and atmosphere_model executables from the top-level MPAS directory.

2. Run init_atmosphere_model to create initial conditions.

3. Run atmosphere_model.

4. Run the supercell.ncl script to produce plots theta, vertical velocity, etc.
* * *
```

**Figure 7**. The screenshot of README file in case suppercell.

Vánˇa, F., Düben, P., Lang, S., Palmer, T., Leutbecher, M., Salmond, D., and Carver, G.: Single Precision in Weather Forecasting Models: An Evaluation with the IFS, Monthly Weather Review, 145, 495-502, doi:10.1175/MWR-D- 16-0228.1, 2016.

---

## Author Comment (AC4)

**Response to Reviewer 4**

We are incredibly grateful for your efficient review process. Your insightful comments have provided valuable guidance for revising the manuscript. We have revised the manuscript according to your suggestions and will respond to your comments paragraph by paragraph. The comments are given below in black, our responses are in blue, and proposed changes to the manuscript are in red. Additional references are provided at the end of this document. The final revisions and specific locations corresponding to the manuscript will be marked uniformly after receiving feedback from other reviewers.

**Major Comments**

The abstract states: "Low precision computations can significantly reduce computational costs, but inevitably introduce rounding errors, which affect computational accuracy." However, it is not clearly defined what is meant by "low precision computations," and the statement that such computations inevitably affect accuracy may not always hold true.

We thank the reviewer for the valuable feedback. In the revised manuscript, we have added a clear definition of "low precision computations".

Regarding the assertion that "low precision computations inevitably affect accuracy," we appreciate the reviewer's insight, and we will revise the description in the abstract. The specific modifications are detailed below.

1 Introduction

Specifically, we define low precision computations as those that utilize a limited number of significant digits (less than 64 bits) during numerical operations, which can significantly reduce the computational resources required while potentially introducing rounding errors.

Abstract

While low precision computations can significantly reduce computational costs, they may introduce rounding errors that can affect computational accuracy under certain conditions.

Both the abstract and conclusions mention the computational impact of the proposed methods; however, these effects are not discussed further within the main body of the manuscript.

We appreciate your suggestion, following your insightful comment, we will:

1. Add a dedicated section 3.4 in the paper to describe the computational performance. The size of the computational performance will be represented in terms of runtime, and we will discuss the runtime for each case in tabular form.

2. Revise the description of computational performance in the abstract to reflect these updated and more accurate measurements.

3.4 Computational performance

In comparison with the SGL, although there is a slight increase in runtime, it is minimal, at only 6% (Jablonowski and Williamson baroclinic wave), 0.3% (Super-cell), 2% (Real data with resolution of 120km) and 18% (Real data with resolution of 240km) (Table 1). This slight increase is attributed to the addition of a small number of global variable arrays when using quasi double-precision. And compared to DBL, QDP demonstrated relatively better performance across different cases, reducing the runtime by 29% (Jablonowski and Williamson baroclinic wave), 29% (Super-cell), 21% (Real data with resolution of 120km) and 6% (Real data with resolution of 240km) (Table 1).

**Table 1**. Elapsed time of DBL, SGL and QDP test (unit:s).

| Case name | DBL | SGL | QDP |
|---|---|---|---|
| Jablonowski and Williamson baroclinic wave | 1768 | 1191 | 1263 |
| Super-cell | 1507 | 1073 | 1077 |
| Real data with resolution of 120km | 19126 | 14765 | 15092 |
| Real data with resolution of 240km | 1397 | 1118 | 1317 |

Abstract

The content 'The round-off error of surface pressure is reduced by 68%, 75%, 97%, 96% in cases, the memory has been reduced by almost half, while the computation increases only 2%, significantly reducing computational cost.' will be revised to 'The bias of surface pressure are reduced respectively by 68%, 75%, 97% and 96% in cases, the memory has been reduced by almost half, while the computation increases only 6%, 0.3%, 2%, and 18% in cases, significantly reducing computational cost.'

Additional context would be beneficial in distinguishing which differences are relevant. Including an uncertainty analysis would help place the magnitude of the errors into perspective. For example, Figure 8 shows significant differences in errors between low- and high-resolution grids, with these discrepancies appearing more impactful than those arising from precision changes alone.

Thank you for your insightful comments. As you mentioned, there are multiple factors contributing to the errors observed in our study. In addition to round-off errors associated with floating-point arithmetic, the choice of grid resolution also has a significant impact on bias. We appreciate your suggestion regarding the inclusion of an uncertainty analysis, and we will consider incorporating this into the revised manuscript to provide further context. We will add a description in Section 3.3, you can see as follows:

3.3 Additional content

In this research, we focus on the processes of summing the basic field and trends. When the resolution is increased, the basic field remains relatively unchanged; however, the trends become smaller. This characteristic aligns with the nature of adding large and small numbers, making the advantages of the quasi double-precision algorithm more pronounced. Thus, it is evident from Figure 8 that as the resolution increases, the improvement achieved by quasi double-precision

algorithm also enhances.

On the other hand, it is important to note that the propagation of rounding errors is not immediately apparent over short time scales. However, as the number of iterations increases, these errors can become more significant. The quasi double-precision algorithm employs compensation mechanisms that help mitigate the propagation of these errors.

Section 2.3 would benefit from further detail on which parts of the code were modified and how these sections were selected for modification.

Thank you for your valuable feedback. To address this, we will:

1. Add the clear description of the solution method for the equations including temporal integration scheme and spatial discretization scheme in section 2.2. See 2.2 additional content below.

2. Specify at which step the quasi double-precision algorithm is applied within the computation process in section 2.3 and replace the figure (corresponding to Figure 3 in the manuscript) and explain this process using formulas and explanations. See 2.3 additional content below.

2.2  Additional content

The MPAS-A solves the fully compressible, nonhydrostatic equations of motion (Skamarock et al. 2012). The spatial discretization uses a horizontal (spherical) centroidal Voronoi mesh with a terrain-following geometric-height vertical coordinate and C-grid staggering for momentum. The temporal discretization uses the explicit time-split Runge–Kutta technique from Wicker and Skamarock (2002) and Klemp et al. (2007).

The algorithm applied here primarily addresses the rounding error compensation between large and small numbers in addition. Currently, it is only applicable to the time integration process and has not been implemented in the spatial discretization process. Therefore, this section will provide a detailed introduction to the time integration scheme. For the spatial discretization scheme, please refer to Skamarock et al. (2012), and it will not be introduced upon here.

The formulation of the scheme can be considered in on dimension as equation Wicker and Skamarock (2002):

$$\frac{\partial \emptyset}{\partial t} = RHS_\emptyset \qquad (1)$$

The variable $\emptyset$ represents any prognostic variable in the prognostic equations, while RHS represents the right-hand side of the prognostic equations (i.e., the spatial discretization equation). In MPAS-A, a forward-in-time finite difference is used, and it can be written as Eq. (2):

$$\frac{\emptyset_i^{n+1} - \emptyset_i^n}{\Delta t} = RHS_\emptyset \qquad (2)$$

Where superscript represent the time step, and subscript represent the position of grid zone.

The two-order Runge-Kutta time scheme is used in MPAS-A as described in Gear et al. (1971):

$$\emptyset^* = \emptyset^t + \frac{\Delta t}{2} * RHS(\emptyset^t) \tag{3}$$

$$\emptyset^{**} = \emptyset^t + \frac{\Delta t}{2} * RHS(\emptyset^*) \tag{4}$$

$$\emptyset^{t+\Delta t} = \emptyset^t + \Delta t * RHS(\emptyset^{**}) \tag{5}$$

**2.3 Additional content**

According to Equation Eq. (3), (4) and (5), it can be observed that in the time integration scheme, each step involves the process of adding tends on the basic field $\emptyset^t$. In numerical models, the basic field is generally much larger than the tends, which aligns with the principles of numerical computation regarding the addition of large and small numbers, as well as the time integration process. It is important to note that the quasi double-precision algorithm currently only addresses time integration and has not been validated during the spatial discretization process. The spatial discretization primarily involves subtraction, specifically the subtraction of a small number from a large number or the subtraction of two close values. Whether this algorithm is applicable in spatial discretization remains uncertain, therefore, we will not apply it in this context.

Based on the application principles of the algorithm, which involve the processes of adding large and small numbers as well as the time integration process, we have established a strategy for applying the quasi double-precision algorithm within the MPAS-A. Specific improvements are provided based on the predictive equations:

$$\frac{\partial V_H}{\partial t} = -\frac{\rho_d}{\rho_m}\left[\nabla_\zeta\left(\frac{p}{\zeta_z}\right) - \frac{\partial z_H p}{\partial \zeta}\right] - \eta k \times V_H - v_H\nabla_\zeta \cdot V - \frac{\partial \Omega v_H}{\partial \zeta} - \rho_d\nabla_\zeta K - eW cos\alpha_r -$$

$$\frac{v_H W}{r_e} + F_{V_H} \tag{6}$$

$$\frac{\partial W}{\partial t} = -\frac{\rho_d}{\rho_m}\left[\frac{\partial p}{\partial \zeta} + g\tilde{\rho}_m\right] - (\nabla \cdot vW)_\zeta + \frac{uU+vV}{r_e} + e(Ucos\alpha_r - Vsin\alpha_r) + F_W \tag{7}$$

$$\frac{\partial \Theta_m}{\partial t} = -(\nabla \cdot V\theta_m)_\zeta + F_{\Theta_m} \tag{8}$$

$$\frac{\partial \tilde{\rho}_d}{\partial t} = -(\nabla \cdot V)_\zeta \tag{9}$$

The meaning of each variable in the equations exactly follows Skamarock et al. (2012), so that we don't repeating explanation. For a numerical model, the most crucial variables are the prognostic variables. Therefore, In the MPAS-A model we applied the quasi double-precision algorithm to the time integration process of these prognostic variables, including horizontal momentum ($V_H$), dry air density ($\tilde{\rho}_d$), potential temperature ($\Theta_m$) and vertical velocity ($W$), that is , the process in red of Eq. (6), (7), (8) and (9). (Only the predictive equations for the dynamic core are presented here, without the scalar transport.)

In Section 3.1, the differences between cases only emerge after 10 days of integration. It would be valuable to contextualize these differences with the error growth from other potential sources of uncertainty.

Thank you for your valuable suggestion. In response, we will add an analysis of other sources of error in Section 3.1. The content will include the following:

3.1 Additional content

The sources of unpredictability, as noted by Bauer et al. (2015), include instabilities that inject chaotic 'noise' at small scales and the upscale propagation of their energy. For the cases examined, both SGL and QDP begin to exhibit errors after 10 days of integration. These errors arise from factors such as rounding errors due to reduced numerical precision and energy loss during the propagation process. The quasi double-precision algorithm can reduce the impacts of these errors.

While we acknowledge other potential sources of uncertainty, such as initial condition errors, we have not conducted an in-depth study on them in this research. Our primary focus remains on evaluating the improvements provided by the compensation algorithm in addressing rounding errors.

Section 3.2 states that "the errors are very small and can be ignored." More context is needed here to help determine which differences are meaningful.

Thank you for your insightful comment regarding Figure 6(a) of section 3.1. According to RC1, we carefully reviewed our code based on your comments and identified that the issue was a problem with the data processing and plotting. We have corrected this issue, and the revised Figure 1 is provided below, it can be seen, the average bias between DBL and QDP is smaller than DBL and SGL.

[Figure]

**Figure 1.** The temporal evolution of spatially averaged difference of kinetic energy between DBL and SGL, as well as difference between DBL and QDP in case of super-cell.

In Section 3.3, the authors note that "Differences in error begin to emerge after 500 steps." This could be strengthened by comparing this error growth to that of other sources of uncertainty, some of which may become relevant earlier in the integration process.

Thank you for your insightful suggestion. We will enhance our discussion in Section 3.3 by adding a comparison of error growth with other sources of uncertainty. The added content will be as follows:

3.3 Additional Contents

Consistent with the analysis presented in Section 3.1, errors are relatively small in the early stages and begin to emerge after 500 steps. This increase is attributed to the accumulation of round-off errors and energy loss over time. The effects become more pronounced beyond 500 steps. Overall, the quasi double-precision algorithm demonstrates a certain level of improvement in addressing these errors.

**Minor Comments**

Line 19 – The authors reference a 2015 source to indicate that systems are expected to grow. While this is still valid, the "current systems" referenced in 2015 are no longer today's current systems.

Thank you for your suggestions, we have revised it by modifying the "current systems" to "2015's systems".

Line 47 – The mixed-precision reference for NEMO notes that "95.8% of the 962 variables could be computed using half precision," though the publication itself refers to single precision.

Thank you for your suggestions, we have revised it by modifying the "half precision" to "single precision".

Line 143 – It is unclear if this version of MPAS-A is indeed the only one capable of utilizing single-precision.

We appreciate your comment. Upon reviewing the official documentation and user guide, we found that MPAS-A can be complied and run in single-precision since version 2.0. We apologize for the oversight and will revise the description accordingly.

Figures 5, 7, 9, 10, 11, and 12 – Alternative colormaps are suggested for all figures containing maps.

Thank you for your suggestion regarding the colormaps. We have made the requested modifications to the colormaps in all relevant figures. The updated figures are provided as Figure 1(corresponding to Figure 5 in the manuscript), Figure 2(corresponding to Figure 7 in the manuscript), Figure 3(corresponding to Figure 9 in the manuscript), Figure 4(corresponding to Figure 10 in the manuscript), Figure 5(corresponding to Figure 11 in the manuscript), Figure 6(corresponding to Figure 12 in the manuscript).

[Figure]

**Figure 1.** Spatial distributions of averaged (1-15days) difference of surface pressure (units: Pa) between DBL and (a) SGL simulations, (b) QDP simulations (round-off error has reduced) in case of Jablonowski and Williamson baroclinic wave.

[Figure]

**Figure 2.** Perturbation theta in super-cell development at 5400s in the (a) DBL simulation, (b) SGL simulation and (c) QDP simulation (bias has reduced), unit: K, the circle represents the pattern bias.

[Figure]

**Figure 3.** Spatial distributions of averaged (1-15days) difference of surface pressure (units: Pa) between DBL and (a) SGL simulation, (b) QDP simulation (resolution: 240 km × 240 km). The RMSE of surface pressure between DBL and (a) SGL simulation is $6.68 \times 10^{-2}$ Pa, (b) QDP simulation is $2.25 \times 10^{-3}$ Pa. (The color bars of (a) and (b) are different)

[Figure]

**Figure 4.** Spatial distributions of averaged (1-15days) difference of 500hPa height (units: m) between DBL and (a) SGL simulation, (b) QDP simulation (resolution: 240 km × 240 km). The RMSE of 500hPa height between DBL and (a) SGL simulation is $2.80 \times 10^{-1}$ m, (b) QDP simulation is $1.40 \times 10^{-1}$ m (round-off error has reduced).

[Figure]

**Figure 5.** distributions of averaged (1-15days) difference of surface pressure (units: Pa) between DBL and (a) SGL simulation, (b) QDP simulation (resolution: 120 km × 120 km) (round-off error has reduced). The RMSE of surface pressure between DBL and (a) SGL simulation is $6.33 \times 10^{-2}$ Pa, (b) QDP simulation is $2.25 \times 10^{-2}$ Pa. (The color bars of (a) and (b) are different)

[Figure]

**Figure 6.** Spatial distributions of averaged (1-15days) difference of 500 hPa height (units: m) between DBL and (a) SGL simulation, (b) QDP simulation (resolution: 120 km × 120 km). The RMSE of 500 hPa height between DBL and (a) SGL simulation is $4.35 \times 10^{-3}$ m, (b) QDP simulation is $1.90 \times 10^{-3}$ m. (The color bars of (a) and (b) are different)

Line 241–242 – The phrase "but the process is more sensitive for the precision" lacks clarity and would benefit from rephrasing.

Thank you for your valuable suggestion. I will rephrase this statement for improved clarity in the revised manuscript. See bellows:

Currently, the quasi double-precision algorithm is only implemented in the time integration scheme of the dynamic core of the MPAS-A model, without factoring in tracer transport. However, the tracer transport process involves numerous operations where large and small numbers are added together, making it more sensitive to precision requirements.

Bauer, P., Thorpe, A., and Brunet, G.: The quiet revolution of numerical weather prediction, Nature, 525(7567):47-55, doi:10.1038/nature14956, 2015.

Skamarock, W. C., Klemp, J. B., Duda, M. G., Fowler, L. D., Park, S. H., and Ringler, T. D.: A Multiscale Nonhydrostatic Atmospheric Model Using Centroidal Voronoi Tesselations and C-Grid Staggering, Monthly Weather Review, 240(9):3090-3105, doi:10.1175/MWR- D-11-00215.1, 2011.

Wicker, L. J., & Skamarock, W. C.: Time-splitting methods for elastic models using forward time schemes. Monthly Weather Review, 130(8), 2088. https://www.proquest.com/scholarly-journals/time-splitting-methods-elastic-models-using/docview/198148677/se-2, 2002.

Gear, C. W.: Numerical initial value problems in ordinary differential equations[M]. Englewood Cliffs, N.J: Prentice-Hall, 1971.

---

## Author Comment (AC5)

**Response to Reviewer 5**

We are incredibly grateful for your efficient review process. Your insightful comments have provided valuable guidance for revising the manuscript. We have revised the manuscript according to your suggestions and will respond to your comments paragraph by paragraph. The comments are given below in black, our responses are in blue, and proposed changes to the manuscript are in red. Additional references are provided at the end of this document. The final revisions and specific locations corresponding to the manuscript will be marked uniformly after receiving feedback from other reviewers.

**Specific comments:**

- The algorithm description needs more attention and description in the text. By comparison, the description of MPAS seems much better detailed and it could be argued that much of it is less important. I recommend finding ways to enhance the content of section 2.1.

Thank you for your valuable feedback. The algorithm itself is proposed by Møller (1965), and it only apply in the step-by-step integration of ordinary differential equations, how to apply it in numerical models is core content, so to address this, we will:

1. Add the clear description of the solution method for the equations including temporal integration scheme and spatial discretization scheme in section 2.2. Moreover, the algorithm applied here primarily addresses the rounding error compensation between large and small numbers in addition. Currently, it is only applicable to the time integration process and has not been implemented in the spatial discretization process. I apologize for not mentioning that part in the manuscript. I will revise and provide the supplement. See 2.2 additional content below.
2. Enhance the algorithm description in Section 2.3. Specify at which step the quasi double-precision algorithm is applied within the computation process in section 2.3 and replace the figure (corresponding to Figure 3 in the manuscript) and explain this process using formulas and explanations. See 2.3 additional content below.
3. we will select a representative equation and include a detailed algorithmic description of its iterative calculation process in the Supplement (see Figure 1). We believe this addition will be beneficial for readers who wish to gain a deeper understanding of the computational aspects of our method.

**Figure 1.** The pseudo-code for variable of U.

3.2  Additional content

The MPAS-A solves the fully compressible, nonhydrostatic equations of motion (Skamarock et al. 2012). The spatial discretization uses a horizontal (spherical) centroidal Voronoi mesh with a terrain-following geometric-height vertical coordinate and C-grid staggering for momentum. The temporal discretization uses the explicit time-split Runge–Kutta technique from Wicker and Skamarock (2002) and Klemp et al. (2007).

The algorithm applied here primarily addresses the rounding error compensation between large and small numbers in addition. Currently, it is only applicable to the time integration process and has not been implemented in the spatial discretization process. Therefore, this section will provide a detailed introduction to the time integration scheme. For the spatial discretization scheme, please refer to Skamarock et al. (2012), and it will not be introduced upon here.

The formulation of the scheme can be considered in on dimension as equation Wicker and Skamarock (2002):

$$\frac{\partial \emptyset}{\partial t} = RHS_\emptyset \tag{1}$$

The variable $\emptyset$ represents any prognostic variable in the prognostic equations, while RHS represents the right-hand side of the prognostic equations (i.e., the spatial discretization equation). In MPAS-A, a forward-in-time finite difference is used, and it can be written as Eq. (2):

$$\frac{\emptyset_i^{n+1} - \emptyset_i^n}{\Delta t} = RHS_\emptyset \tag{2}$$

Where superscript represent the time step, and subscript represent the position of grid zone.

The two-order Runge-Kutta time scheme is used in MPAS-A as described in Gear et al. (1971):

$$\emptyset^* = \emptyset^t + \frac{\Delta t}{2} * RHS(\emptyset^t) \tag{3}$$

$$\emptyset^{**} = \emptyset^t + \frac{\Delta t}{2} * RHS(\emptyset^*) \qquad (4)$$

$$\emptyset^{t+\Delta t} = \emptyset^t + \Delta t * RHS(\emptyset^{**}) \qquad (5)$$

**3.3 Additional content**

According to Equation Eq. (3), (4) and (5), it can be observed that in the time integration scheme, each step involves the process of adding tends on the basic field $\emptyset^t$. In numerical models, the basic field is generally much larger than the tends, which aligns with the principles of numerical computation regarding the addition of large and small numbers, as well as the time integration process. It is important to note that the quasi double-precision algorithm currently only addresses time integration and has not been validated during the spatial discretization process. The spatial discretization primarily involves subtraction, specifically the subtraction of a small number from a large number or the subtraction of two close values. Whether this algorithm is applicable in spatial discretization remains uncertain, therefore, we will not apply it in this context.

Based on the application principles of the algorithm, which involve the processes of adding large and small numbers as well as the time integration process, we have established a strategy for applying the quasi double-precision algorithm within the MPAS-A. Specific improvements are provided based on the predictive equations:

$$\frac{\partial \boldsymbol{V}_H}{\partial t} = -\frac{\rho_d}{\rho_m}\left[\nabla_\zeta\left(\frac{p}{\zeta_z}\right) - \frac{\partial z_H p}{\partial \zeta}\right] - \eta \boldsymbol{k} \times \boldsymbol{V}_H - \boldsymbol{v}_H \nabla_\zeta \cdot \boldsymbol{V} - \frac{\partial \Omega \boldsymbol{v}_H}{\partial \zeta} - \rho_d \nabla_\zeta K - eW\cos\alpha_r -$$

$$\frac{\boldsymbol{v}_H W}{r_e} + \boldsymbol{F}_{\boldsymbol{V}_H} \qquad (6)$$

$$\frac{\partial W}{\partial t} = -\frac{\rho_d}{\rho_m}\left[\frac{\partial p}{\partial \zeta} + g\tilde{\rho}_m\right] - (\nabla \cdot \boldsymbol{v}W)_\zeta + \frac{uU+vV}{r_e} + e(U\cos\alpha_r - V\sin\alpha_r) + F_W \qquad (7)$$

$$\frac{\partial \Theta_m}{\partial t} = -(\nabla \cdot \boldsymbol{V}\theta_m)_\zeta + F_{\Theta_m} \qquad (8)$$

$$\frac{\partial \tilde{\rho}_d}{\partial t} = -(\nabla \cdot \boldsymbol{V})_\zeta \qquad (9)$$

The meaning of each variable in the equations exactly follows Skamarock et al. (2012), so that we don't repeating explanation. For a numerical model, the most crucial variables are the prognostic variables. Therefore, In the MPAS-A model we applied the quasi double-precision algorithm to the time integration process of these prognostic variables, including horizontal momentum ($\boldsymbol{V}_H$), dry air density ($\tilde{\rho}_d$), potential temperature ($\Theta_m$) and vertical velocity ($W$), that is, the process in red of Eq. (6), (7), (8) and (9). (Only the predictive equations for the dynamic core are presented here, without the scalar transport.)

- For clarification, is the QDP runtime is 2% more expensive compared to the DBL runtime or the SGL runtime? Also how much reduction in runtime was there for the SGL version compared to the DBL version? This can be used to give context to what benefits are achieved for the more noticeable errors in SGL.

Thanks for your suggestion, following your insightful comment, we found the previously reported figures were indeed rough estimations, so we have re-evaluated the exact computational performance using a measurement tool to determine the runtimes. So, we will:

1. Add a dedicated section 3.4 in the paper to describe the computational performance. The size of the computational performance will be represented in terms of runtime, and we will discuss the runtime for each case in tabular form.
2. Revise the description of computational performance in the abstract to reflect these updated and more accurate measurements.

3.4 Computational performance

In comparison with the SGL, although there is a slight increase in runtime, it is minimal, at only 6% (Jablonowski and Williamson baroclinic wave), 0.3% (Super-cell), 2% (Real data with resolution of 120km) and 18% (Real data with resolution of 240km) (Table 1). This slight increase is attributed to the addition of a small number of global variable arrays when using quasi double-precision. And compared to DBL, QDP demonstrated relatively better performance across different cases, reducing the runtime by 29% (Jablonowski and Williamson baroclinic wave), 29% (Super-cell), 21% (Real data with resolution of 120km) and 6% (Real data with resolution of 240km) (Table 1).

**Table 1**. Elapsed time of DBL, SGL and QDP test (unit:s).

| Case name | DBL | SGL | QDP |
| --- | --- | --- | --- |
| Jablonowski and Williamson baroclinic wave | 1768 | 1191 | 1263 |
| Super-cell | 1507 | 1073 | 1077 |
| Real data with resolution of 120km | 19126 | 14765 | 15092 |
| Real data with resolution of 240km | 1397 | 1118 | 1317 |

Abstract

The content 'The round-off error of surface pressure is reduced by 68%, 75%, 97%, 96% in cases, the memory has been reduced by almost half, while the computation increases only 2%, significantly reducing computational cost.' will be revised to 'The bias of surface pressure are reduced respectively by 68%, 75%, 97% and 96% in cases, the memory has been reduced by almost half, while the computation increases only 6%, 0.3%, 2%, and 18% in cases, significantly reducing computational cost.'

- Given errors appear to become noticeably larger beyond 10+ days, how are these figures dependent

with different spatial time average lengths? Do they all only look drastically different because of errors after round off error becomes significant around day 10+? Is this error tied to simulation length or is this more determined by the number of time steps taken?

Thank you for your insightful questions. As you noted, errors become significantly larger after approximately 10 days due to factors such as round-off errors arising from reduced numerical precision and energy loss during the propagation process. Around day 10, the accumulation of rounding errors becomes pronounced, continuing to accumulate over time, which can ultimately lead to model instability.

In this study, we did not conduct a comparative analysis involving different time step lengths; therefore, we cannot definitively conclude whether the observed errors are related to the chosen time step sizes. However, round-off errors do accumulate with the simulation length.

We appreciate your suggestion, and in future research, we will focus on whether integration step lengths have an impact on error growth.

-   In some experiments, the reduction of error is a large percentage but feels like a small error. In general to the reader, it's not very clear that reducing these errors will matter that much. For example, while the error is reduced using QDP in a lot of cases, reduction of the surface pressure (hPa) error by a RSME of 2.8x10-1 to 1.4x10-1 seems like a small amount. Likewise for other sources of error and that this all needs context. It is important to consider these errors relative to other sources of model errors.

Thank for your suggestion. The core of this study is the introduction of error compensation methods (quasi double-precision). By using error compensation methods (quasi double-precision), we can maintain integration stability comparable to that applying double precision scheme while significantly reducing memory requirements by lowering the numerical precision of all variables and improved the accuracy comparable to that applying the single precision. This approach not only reduces communication pressure but also allows for substantial increases in computational speed through vectorization optimization. So, we will add a section to describe it (see 3.4 Computational performance)

And we will add an analysis of other sources of error in Section 3.1. The content will include the following:

3.1 Additional content

The sources of unpredictability, as noted by Bauer et al. (2015), include instabilities that inject chaotic 'noise' at small scales and the upscale propagation of their energy. For the cases examined, both SGL and QDP begin to exhibit errors after 10 days of integration. These errors arise from factors such as rounding errors due to reduced numerical precision and energy loss during the propagation process. The quasi double-precision algorithm can reduce the impacts of these errors.

While we acknowledge other potential sources of uncertainty, such as initial condition errors, we have not conducted an in-depth study on them in this research. Our primary focus remains on evaluating the improvements provided by the compensation algorithm in addressing rounding errors.

- Additionally, I was wondering if the authors could leverage the test case where grid resolution was also incorporated as the model discretization is a source of model error compared to the model error due to precision differences.

Thank you for your insightful comments. As you mentioned, there are multiple factors contributing to the errors observed in our study. In addition to round-off errors associated with floating-point arithmetic, the choice of grid resolution also has a significant impact on bias. We will add a description in Section 3.3, you can see as follows:

3.3 Additional content

In this research, we focus on the processes of summing the basic field and trends. When the resolution is increased, the basic field remains relatively unchanged; however, the trends become smaller. This characteristic aligns with the nature of adding large and small numbers, making the advantages of the quasi double-precision algorithm more pronounced. Thus, it is evident from Figure 8 that as the resolution increases, the improvement achieved by quasi double-precision algorithm also enhances.

On the other hand, it is important to note that the propagation of rounding errors is not immediately apparent over short time scales. However, as the number of iterations increases, these errors can become more significant. The quasi double-precision algorithm employs compensation mechanisms that help mitigate the propagation of these errors.

- For the cases in 3.3, it is not clear why the decision to have the higher resolution simulation of 120 km x 120 km have the same time step of 720 seconds as the 240 km x 240 km case. This seems rather unusual as typically a reduction in grid cell size is accompanied by a proportionate reduction in time step size to maintain a consistent Courant number.

Thank you for your question. As you mentioned, it is common for time step sizes to decrease proportionally with grid resolution. However, in our experiments, we found that the simulation could run stably at the finer resolution (i.e., 120 km) without requiring a reduction in the time step size. To minimize errors arising from differences in resolution during data processing, we chose to maintain the same time step across both simulations.

**Minor comments:**

- Figure 1, Figure 2 and a lesser extent Figure 3: The algorithm are written in a way that is difficult to read. I suggest that it is better distinguished the text (i.e. "evaluation") from variables (i.e. s,u,v). In Figure 2, everything is appears quite mathematical while Figure 3 is quite the opposite. It would be beneficial to have these all looking the same.

Thank you for your valuable feedback. We have made the necessary modifications. For details on these changes, please refer to the responses provided in the first specific comments.

- Line 150: "so we close the scalar transport in all cases". Does this mean it's turned off/not solved?

Unless "closed" is a common term in the MPAS community, I suggest the authors consider a term closer to "disable/disabled" instead of "close/closed" when talking about model processes that are not absent in the simulation.

Thank you for your insightful comment. We will revise it from "closed" to "turned off".

- Line 179: Has this set of simulation parameters been utilized in a previous published work as a test case? This would be preferred to stating it was on a website that is subject to change.

Thank you for your suggestion. We will revise this section to clarify the source of our simulation parameters. The modified content will state:

The experimental configuration is consistent with the test case presented by Jablonowski and Williamson (2006).

- Line 195 states the resolution is 84 km x 84 km where I believe this is actually the total domain size and not resolution. Assuming my understanding is correct, how many grid cells are within this domain and/or what is the average grid cell size in this simulation? Is it the ~500 m in Klemp 2015?

Thank you for your question. As you pointed out, 84 km × 84 km refers to the total domain size. We will revise this in the manuscript. Within this domain, there are a total of 28,080 grid cells, which is not the 500 m that you mentioned.

- Line 197: While the errors in Figure 6(a) are indeed quite small, it may be beneficial to say how small relative to the actual values of kinetic energy? And I think a more clear statement explaining this figure's behavior would help clarify the plot as it's not the expected behavior throughout the manuscript that QDP does better in terms of error.

Thank you for your insightful comment regarding Figure 6(a) of section 3.1. According to RC1, we carefully reviewed our code based on your comments and identified that the issue was a problem with the data processing and plotting. We have corrected this issue, and the revised Figure 2 is provided below, it can be seen, the average bias between DBL and QDP is smaller than DBL and SGL.

[Figure]

**Figure 2.** The temporal evolution of spatially averaged difference of kinetic energy between DBL and SGL, as well as difference between DBL and QDP in case of super-cell.

- Figure 7:    The location of the Y (km) should be moved up near the axes (and should probably be labeled X?) and not under the color bar and Z is used which is typically thought of as a vertical direction is on the y axis. Color bar should probably have different scaling with a smaller range. Additionally why are the circles different colors between the different panels?

Thank you for your valuable suggestions. We have made the necessary modifications as follows, as seen in Figure 3 (which corresponds to Figure 7 in the manuscript):

1.   The axis labels have been updated.
2.   The position of the color bar has been adjusted.
3.   The range of the color bar has been modified for better scaling.

Furthermore, we have added a note in the caption to clarify the differences in circle colors between the various panels.

[Figure]

**Figure 3.** Perturbation theta in super-cell development at 5400s in the (a) DBL simulation, (b) SGL simulation and (c) QDP simulation (bias has reduced), unit: K, the circle represents the pattern bias (the same color means the consistent value).

- Figure 9 is one of the only figures that has color bar limits change between the SGL and QDP panels. Given that quite often the color bar limit have been fixed ranges in most figures, a note in the caption may be beneficial for this.

Thank you for your suggestion. We will include a note in the caption of Figure 9 indicating that the color bars used in panels (a) and (b) are different. This will help clarify the distinction for readers.

- Figure 10(b) uses the same color bar as Figure 10(a) but reveals no detail because the same color bar limits are too large. I suggest that a new color bar range is selected (and it is then noted in the caption as being different in an and b as suggested for Figure 9).

Thank you for your suggestion. I believe you may be referring to Figure 11 rather than Figure 10. I will revise the color bar for Figure 11 to improve clarity and detail, as well as update the caption accordingly. Please see below for Figure 4 (corresponding to Figure 11 in the manuscript).

[Figure]

**Figure 4.** distributions of averaged (1-15days) difference of surface pressure (units: Pa) between DBL and (a) SGL simulation, (b) QDP simulation (resolution: 120 km × 120 km) (round-off error has reduced). The RMSE of surface pressure between DBL and (a) SGL simulation is $6.33 \times 10^{-2}$ Pa, (b) QDP simulation is $2.25 \times 10^{-2}$ Pa. (The color bars in (a) and (b) are different)

- For code and data availability section, it is mentioned that "model code and plotting data" is available While I was able to find the versions of the model code, the simulation inputs and plotting scripts, I was unable to locate actual data. It's unclear if this is intended as the wording indicates "plotting data" and we should rather anticipate running the model to produce the data.

Thank you for your valuable feedback. Due to the large size of the output data generated by the model, we have not uploaded the actual output files. If you would like to obtain the processed output data, you can compile and run the model following the instructions in the README file, which will allow you to generate the output data.

To enhance clarity, I will revise the description in the "Code and Data Availability" section accordingly. The updated wording is provided below.

Code and data availability. Model code and plotting data related to this manuscript is available at:

https://doi.org/10.5281/zenodo.13765422. Details regarding the code structure and instructions for running the code are provided in the supplementary material, which can be downloaded and viewed in Fig. S1. This figure provides a visual overview of the code organization. The information of steps how to execute the simulations can be found in README file in each test case folder.

**Technical comments:**

- Negative exponents have been routinely miswritten throughout the manuscript such as 10-2 rather than 10-2.

Thank you for your suggestion. We have corrected these errors throughout the manuscript.

- Line 218: "The Spatial RMSE of with 120 km x 120km.." is missing the variable name, which appears to be surface pressure.

Thank you for your observation. I appreciate your attention to detail. I have corrected this in the revised manuscript by including the variable name, which is indeed surface pressure.

JABLONOWSKI C, WILLIAMSON D L.: A baroclinic instability test case for atmospheric model dynamical cores[J/OL]. Quarterly journal of the Royal Meteorological Society, 132(621C): 2943-2975. DOI:10.1256/qj.06.12, 2006.

---

## Author Comment (AC7)

**Response to reviewers of the manuscript**

We thank the Dr. Huang and the four anonymous authors for their careful review and detailed comments. We have revised the manuscript to respond to these comments, and we also provide a point-by-point response below to each review comment. The comments are given below in black, our responses are in blue, and proposed changes to the manuscript are in red. Additional references are provided at the end of this document.

**Response to Anonymous Referee #1**

**Reply on RC1:**

**Major comments:**

Based on Figure 1 and Figure 2, this computational method appears somewhat simple. There is relatively little research on the keyword "quasi double-precision." The Authors could explain why their work is novel compared to the existed methods already published nowadays.

Thank you for your comments. We have added the significance and novelty of our work in section: introduction and section: Conclusions and discussion and have been summarized below.

**Line 98~107**: Most works involving numerical models that reduce numerical precision adopt a mixed-precision scheme, where some variables use single precision while others remain in double precision to ensure integration stability, as demonstrated in the work of Chen et al. (2024). Currently, there are very few studies that almost entirely employ low precision (32-bit) in numerical models, only applied in IFS by Vánˇa et al. (2016). However, they only utilize single precision without considering error compensation for it. In this study, all variables in the numerical model were implemented using single precision, and error compensation was applied to key variables. By using error compensation methods (quasi double-precision), we can maintain integration stability comparable to that applying double precision scheme while significantly reducing memory requirements by lowering the numerical precision of all variables and improved the accuracy comparable to that applying the single precision. This approach not only reduces communication pressure but also allows for substantial increases in computational speed through vectorization optimization.

**Line 297~315:** The algorithm can compensate for round-off errors by keeping corrections in addition of large and small numbers. And in numerical models, the basic field is generally much larger than the tends, which aligns with the principles of quasi double-precision, as well as the time integration process. Based on the it, we have established a strategy for applying the quasi double-precision algorithm within the MPAS-A. Through the implementation of quasi double-precision methods, we maintain accuracy similarly to the tests using double precision and achieve comparable integration stability to the tests comparing to single precision tests. The error of surface pressure of 4 cases are reduced by 68%, 75%, 97%, 96% (see Section 3). Overall, QDP using quasi double-precision algorithm demonstrates higher accuracy than the SGL, suggesting the potential for applying quasi double-precision algorithm in numerical models.

We don't apply it to spatial discretization process, because spatial discretization primarily involves subtraction, specifically the subtraction of a small number from a large number or the subtraction of two close values. Whether this algorithm is applicable in spatial discretization remains uncertain, therefore, we don't consider it in this context.

While mixed-precision approaches, where certain variables retain double precision for stability (e.g., Chen et al., 2024), are 305 common for reducing numerical precision in models, and they don't consider the error compensation. This study distinguishes itself by implementing single precision for all model variables and applying error compensation for critical variables.

When applied the quasi double-precision algorithm in MPAS-A, we achieved to reduce all double precision to single pre- cision, although increased few local variables and arrays in every time- integration variable, these have little impact on the overall memory reduction. In general, memory has been reduced by almost half, with a corresponding computational increase 310 of just 6.0%, 0.3%, 2.2%, and 17.8% in the respective cases (see section 3.4), demonstrating a substantial improvement in computational efficiency.

The main point of applying signal precision computing methods is ensuring predictive and reducing computational costs. The iterative precision compensation increases the computation load. Has this study considered the issue of computational efficiency? For example, runtime, reduced computational cost, they were mentioned in introduction literature review, but not studied in this study.

We apologize for the oversight. The additional computational cost incurred from using error compensation is minimal; it only adds six global arrays to the entire model code. Therefore, we neglected to elaborate on this aspect. We appreciate your suggestion, following your insightful comment, we found the previously reported figures were indeed rough estimations, so we have re-evaluated the exact computational performance using a measurement tool to determine the runtimes. So, we will:

1. Add a dedicated section 3.4 in the paper to describe the computational performance. The size of the computational performance will be represented in terms of runtime, and we will discuss the runtime for each case in tabular form.
2. Revise the description of computational performance in the abstract to reflect these updated and more accurate measurements.

**Line 286~292:** In comparison with the SGL, although there is a slight increase in runtime, it is minimal, at only 6.0% (Jablonowski and Williamson baroclinic wave), 0.3% (Super-cell), 2.2% (Real data with resolution of 120km) and 17.8% (Real data with resolution of 240km) (Table 3). This slight increase is attributed to the addition of a small number of global variable arrays when using quasi double-precision. And compared to DBL, QDP demonstrated relatively better performance across different cases, reducing the runtime by 28.6% (Jablonowski and Williamson baroclinic wave), 28.5% (Super-cell), 21.1% (Real data with resolution of 120km) and 5.7% (Real data with resolution of 240km) (Table 3).

**Table 3.** Comparative Analysis of Computational Efficiency: DBL vs SGL vs **QDP**

| Case name | DBL | SGL | QDP (Proposed) | | |
|---|---|---|---|---|---|
| | Runtime | Runtime | Runtime | vs DBL | vs SGL |
| JW wave | 1768 s | 1191 s | **1263 s** | -28.6% | +6.0% |
| SC | 1507 s | 1073 s | **1077 s** | -28.5% | +0.3% |
| RD-120 | 19126 s | 14765 s | **15092 s** | -21.1% | +2.2% |
| RD-240 | 1397 s | 1118 s | **1317 s** | -5.7% | +17.8% |

Note: JW wave = Jablonowski & Williamson baroclinic wave;

SC = Super-cell; RD-120/240 = Real data with total domain size of 120/240 km

**Line 7 ~ 9:** The bias of surface pressure are reduced respectively by 68%, 75%, 97% and 96% in cases, the memory has been reduced by almost half, while the computation increases only 6.0%, 0.3%, 2.2%, and 17.8% respectively, significantly reducing computational cost.

The model is described poorly. The solution method for the equations is not even mentioned. For example, the finite difference scheme is mainly used to calculate the primary equations for variables studied in this work. And at which step of the equation is the quasi double-precision algorithm specifically applied? The strategy used to compute cell edge, dry air density, potential temperature with quasi double-precision algorithm is also difficult to understand from Figure 3.

Thank you for your valuable feedback regarding the clarity of our model description and the implementation of the quasi double-precision algorithm. We recognize that the solution method for the equations was not sufficiently detailed in our original manuscript. To address this, we will:

1. Add the clear description of the solution method for the equations including temporal integration scheme and spatial discretization scheme in section 2.2. Moreover, the algorithm applied here primarily addresses the rounding error compensation between large and small numbers in addition. Currently, it is only applicable to the time integration process and has not been implemented in the spatial discretization process. I apologize for not mentioning that part in the manuscript. I will revise and provide the supplement. See below.
2. Specify at which step the quasi double-precision algorithm is applied within the computation process in section 2.3 and replace the figure (corresponding to Figure 3 in the manuscript) and explain this process using formulas and explanations. See below.

**Line 135 ~ 153:** The MPAS-A solves the fully compressible, nonhydrostatic equations of motion (Skamarock et al. 2011). The spatial dis- cretization uses a horizontal (spherical) centroidal Voronoi mesh with a terrain-following geometric-height vertical coordinate and C-grid staggering for momentum. The temporal discretization uses the explicit time-split Runge–Kutta technique from Wicker and Skamarock (2002) and Klemp et al. (2007).

The algorithm applied here primarily addresses the rounding error compensation between large and small numbers in addi- tion. Currently, it is only applicable to the time integration process and has not been implemented in the spatial discretization process. Therefore, this section will provide a detailed introduction to the time integration scheme. For the spatial discretization scheme, please refer to Skamarock et al. (2011), and it will not be introduced upon here.

The formulation of the scheme can be considered in on dimension as equation Wicker and Skamarock (2002):

$$\frac{\partial \emptyset}{\partial t} = RHS_\emptyset \tag{1}$$

The variable $\emptyset$ represents any prognostic variable in the prognostic equations, while RHS represents the right-hand side of the prognostic equations (i.e., the spatial discretization equation). In MPAS-A, a forward-in-time finite difference is used, and it can be written as Eq. (2):

$$\frac{{\emptyset_i}^{n+1} - {\emptyset_i}^{n}}{\Delta t} = RHS_\emptyset \tag{2}$$

Where superscript represent the time step, and subscript represent the position of grid zone.

The two-order Runge-Kutta time scheme is used in MPAS-A as described in Gear et al. (1971):

$$\emptyset^* = \emptyset^t + \frac{\Delta t}{2} * RHS(\emptyset^t) \tag{3}$$

$$\emptyset^{**} = \emptyset^t + \frac{\Delta t}{2} * RHS(\emptyset^*) \tag{4}$$

$$\emptyset^{t+\Delta t} = \emptyset^t + \Delta t * RHS(\emptyset^{**}) \tag{5}$$

**Line 161~179:** According to Equation Eq.(3), (4) and (5), it can be observed that in the time integration scheme, each step involves the process of adding tends on the basic field $\varphi t$. In numerical models, the basic field is generally much larger than the tends, which aligns with the principles of numerical computation regarding the addition of large and small numbers, as well as the time integration process. It is important to note that the quasi double-precision algorithm currently only addresses time integration and has not been validated during the spatial discretization process. The spatial discretization primarily involves subtraction, specifically the subtraction of a small number from a large number or the subtraction of two close values. Whether this algorithm is applicable in spatial discretization remains uncertain, therefore, we will not apply it in this context.

Based on the application principles of the algorithm, which involve the processes of adding large and small numbers as well as the time integration process, we have established a strategy for applying the quasi double-precision algorithm within the MPAS-A. Specific improvements are provided based on the predictive equations:

$$\frac{\partial \boldsymbol{V}_H}{\partial t} = -\frac{\rho_d}{\rho_m}\left[\nabla_\zeta\left(\frac{p}{\zeta_z}\right) - \frac{\partial z_H p}{\partial \zeta}\right] - \eta \boldsymbol{k} \times \boldsymbol{V}_H - \boldsymbol{v}_H \nabla_\zeta \cdot \boldsymbol{V} - \frac{\partial \Omega v_H}{\partial \zeta} - \rho_d \nabla_\zeta K - eW\cos\alpha_r -$$

$$\frac{v_H W}{r_e} + \boldsymbol{F}_{\boldsymbol{V}_H} \tag{6}$$

$$\frac{\partial W}{\partial t} = -\frac{\rho_d}{\rho_m}\left[\frac{\partial p}{\partial \zeta} + g\tilde{\rho}_m\right] - (\nabla \cdot \boldsymbol{v}W)_\zeta + \frac{uU+vV}{r_e} + e(U\cos\alpha_r - V\sin\alpha_r) + F_W \tag{7}$$

$$\frac{\partial \Theta_m}{\partial t} = -(\nabla \cdot \boldsymbol{V}\theta_m)_\zeta + F_{\Theta_m} \tag{8}$$

$$\frac{\partial \tilde{\rho}_d}{\partial t} = -(\nabla \cdot \boldsymbol{V})_\zeta \tag{9}$$

The meaning of each variable in the equations exactly follows Skamarock et al. (2012), so that we don't repeating explanation. For a numerical model, the most crucial variables are the prognostic variables. Therefore, In the MPAS-A model we applied the quasi double-precision algorithm to the time integration process of these prognostic variables, including horizontal momentum ($\boldsymbol{V}_H$) , dry air density ($\tilde{\rho}_d$), potential temperature ($\Theta_m$) and vertical velocity ($W$) , that is , the process in red of Eq. (6), (7), (8) and (9). (Only the predictive equations for the dynamic core are presented here, without the scalar transport.)

The color scheme in Figure 5 is very hard to distinguish; the solid purple area is too large, making the gradient difficult to see. Figure 10 has the same issue. It is recommended to refer to the classic NCL color scheme. https://www.ncl.ucar.edu/Applications/era40.shtml

Thank you very much for your suggestions and the website link. I have revised Figures 5 and 10 according to the guidelines provided in the link. Please see below for Figure 1 (corresponding to Figure 5 in the manuscript) and Figure 2 (corresponding to Figure 10 in the manuscript).

[Figure]

**Figure 1.** Spatial distributions of averaged (1-15days) difference of surface pressure (units: Pa) between DBL and (a) SGL simulations, (b) QDP simulations (round-off error has reduced) in case of Jablonowski and Williamson baroclinic wave.

[Figure]

**Figure 2.** Spatial distributions of averaged (1-15days) difference of 500hPa height (units: m) between DBL and (a) SGL simulation, (b) QDP simulation (resolution: 240 km × 240 km). The RMSE of 500hPa height between DBL and (a) SGL simulation is $2.80 \times 10^{-1}$ m, (b) QDP simulation is $1.40 \times 10^{-1}$ m (round-off error has reduced).

Figure 6 (a) shows that DBL-SGL decreases after 1.0, which appears to be caused by a coding error. Please check and confirm the validity of the data.

Thank you for your insightful comment regarding Figure 6(a), we carefully reviewed our code based on your comments and identified that the issue was a problem with the data processing and plotting. We have corrected this issue, and the revised Figure 3 is provided below:

[Figure]

**Figure 3.** The temporal evolution of spatially averaged difference of kinetic energy between DBL and SGL, as well as difference between DBL and QDP in case of super-cell.

I apologize for needing to provide additional information regarding this section. In all cases presented in the manuscript, the time-evolution plots (Figures 4, 6, and 8 in the manuscript) currently utilize kinetic energy and surface pressure, which correspond to the conservation of energy and mass, respectively. However, I realize that directly using total energy and total mass would offer a more accurate representation. Please see the figures 4, 5 and 6 showing the temporal

variations of energy and mass for all cases. If you allow it, I would be happy to replace the existing Figures 4, 6, and 8 (in the manuscript) with these updated versions (figures 4, 5 and 6). I want to emphasize that this change would not affect the overall results or conclusions of the study, which remain consistent with those currently presented in the manuscript. I think your suggestions are greatly appreciated and have been very helpful in improving the manuscript.

[Figure]

**Figure 4.** The temporal evolution of spatially averaged difference of (a) total energy, (b) total mass between DBL and SGL, as well as difference between DBL and QDP in case of Jablonowski and Williamson baroclinic wave.

[Figure]

**Figure 5.** The temporal evolution of spatially averaged difference of (a) total energy, (b) total mass between DBL and SGL, as well as difference between DBL and QDP in case of super-cell.

It is important to note that, for enhanced clarity and to facilitate a better understanding of the trend, the x-axis unit for the figure representing Case 7 has been changed to hours (Fig. 6).

[Figure]

**Figure 6.** The temporal evolution of spatially averaged difference of total energy between DBL and SGL, as well as difference between DBL and QDP in case of real data, with resolution of (a) 240 km × 240 km, (b) 120 km × 120 km.

**Minor comments:**

The color bar in Figure 7 seems almost useless.

Thank you very much for your careful review and valuable comments on my manuscript. Regarding the issue you raised about the color bar in the figure 7. After adjusting the color bar, two distinct areas of improvement are now evident, compared to only one in the original figure. I sincerely apologize that the previous color bar setting was not appropriate. I have adjusted the color bar according to your suggestion, and the revised figure is shown as Figure 7 (corresponding to Figure 7 in the manuscript).

[Figure]

**Figure 7.** Perturbation theta in super-cell development at 5400s in the (a) DBL simulation, (b) SGL simulation and (c) QDP simulation (bias has reduced), unit: K, the circle represents the pattern bias (the same color means the consistent value).

Please mention "round-off error" in relevant figures' captions.

Thank you very much for your suggestions. I have revised it. You can see it in Figure 6 and Figure 9 in the manuscript.

Figure 11 appears to be somewhat blurry.

Thank you very much for your suggestions. I have revised it. Please see below for Figure 8 (corresponding to Figure 11 in the manuscript).

[Figure]

**Figure 8.** distributions of averaged (1-15days) difference of surface pressure (units: Pa) between DBL and (a) SGL simulation, (b) QDP simulation (resolution: 120 km × 120 km) (round-off error has reduced). The RMSE of surface pressure between DBL and (a) SGL simulation is $6.33 \times 10^{-2}$ Pa, (b) QDP simulation is $2.25 \times 10^{-2}$ Pa. It's important to note that the color bars for Figures (a) and (b) are intentionally set differently due to the significant disparity in their respective threshold ranges. If identical color scales were applied, Figure (b) would appear entirely white.

Wicker, L. J., & Skamarock, W. C.: Time-splitting methods for elastic models using forward time schemes. Monthly Weather Review, 130(8), 2088. https://www.proquest.com/scholarly-journals/time-splitting-methods-elastic-models-using/docview/198148677/se-2, 2002.

Gear, C. W.: Numerical initial value problems in ordinary differential equations[M]. Englewood Cliffs, N.J: Prentice-Hall, 1971.

**Reply on RC2:**

**Response to Point 1:**

The calculation method used in this paper is based on the method of Møller, O. (1965)., which appears relatively simple. If the authors can find similar studies that employ this "quasi double-precision" method for simulating atmospheric variables, it could demonstrate the advancement of this paper compared to existing research. Chen et al. (2024) and Vánˇa et al. (2016) did not use this method. If no similar studies are found, it could be stated that this paper is the first to do so, which could enhance its citation rate.

Thank you very much for your insightful comment. We greatly appreciate your valuable suggestion, which has significantly improved the manuscript. Following your suggestion, we have added a statement highlighting this novel application in both the "Introduction" and "Conclusion and Discussion" sections of the manuscript. We believe this addition provides important context for our work and clarifies its contribution to the field.

**Line 95~97:** The application of the Moller method to a realistic numerical model, as presented in this study, represents a novel contribution to the field, with no prior research exploring this specific implementation.

**Line 294~297:** Although the Moller method (Quasi double-precision) has been extensively employed for the temporal integration of ordinary differential equations, its application within the context of realistic numerical models remains unexplored. This study addresses this gap by presenting a novel implementation of the Moller method, thereby expanding its scope and potential impact within the field.

**Response to Point 3:**

Regarding equations (6)-(9), it is suggested to select only one equation and present it in the form of Figure 2. This could be included as an appendix to specifically demonstrate the iterative calculation process. Because the pseudo-code in Fig. 3 is not sufficient to clearly explain the calculation process.

Thank you very much for your valuable suggestion. We appreciate your feedback and agree that providing an algorithmic description of the iterative process for one of the equations would enhance the clarity of our work. As your recommendation, we will select a representative equation and include a detailed algorithmic description of its iterative calculation process in the Supplement (see Figure 1). We believe this addition will be beneficial for readers who wish to gain a deeper understanding of the computational aspects of our method.

Supplement:

```
Basic field: u_basic, Tend: u_tend
Define array: u_compensation
Initialize u_c to 0.0
Define reals: u_sum, u_big, u_small
Do rk_step from 1 to 3:
    If abs(u_basic) > abs(u_tend):
            u_big = u_basic
            u_small = u_tend
    Else:
            u_big = u_tend
            u_small = u_basic
    end if
    u_small += u_compensation
    u_sum = u_big + u_small
    u_ compensation = (u_small - (u_compensation - u_big))
                        + (u_big - (u_compensation - u_big))
    u_basic = u_sum
    ...
End do rk_step
```

```
Basic field: u_basic, Tend: u_tend
For rk_step from 1 to 3:
    u_basic = u_basic + u_tend
```

**Figure 1.** The pseudo-code for variable of U.

**Response to Point 5:**

If the authors intend to replace the existing Figures 4, 6, and 8 (in the manuscript) with these updated versions (Figures 4, 5, and 6), feedback from other editors and reviewers regarding the revised structure of the paper should be considered.

Thank you very much for your valuable suggestion. I appreciate your careful attention to detail. I will ensure that I obtain the consent of each reviewer and the editor before proceeding with the replacement of the figures, as you recommended.

**The new figures:**

There are several cases where figure legends and lines overlap, such as in Figure 3 in the review response.

Thank you very much for your valuable suggestion. We have completed the revisions as you recommended. The updated figures are as follows: Figure 2 (corresponding to Figure 3 in RC1) and Figure 3 (corresponding to Figure 5 in RC1).

[Figure]

**Figure 2.** The temporal evolution of spatially averaged difference of kinetic energy between DBL and SGL, as well as difference between DBL and QDP in case of super-cell.

[Figure]

**Figure 3.** The temporal evolution of spatially averaged difference of (a) total energy, (b) total mass between DBL and SGL, as well as difference between DBL and QDP in case of super-cell.

Thompson., Robert, J.:Improving round-off in Runge-Kutta computations with Gill's method, Communications of the Acm, 13(12):739- 740.DOI:10.1145/362814.362823, 1970.

Tomonori, Kouya., Hideko, Nagasaka.: On the correction method of round-off errors in the Yang's Runge-Kutta method, The Japan Society for Industrial and Applied Methematics, 1995.

Dmitruk, B., Przemysaw, Stpiczyn ́ski.:Improving accuracy of summation using parallel vectorized Kahan's and Gill-Mller algorithms, Con- currency and Computation: Practice and Experience, doi:10.1002/cpe.7763, 2023.

**Response to Reviewer #2**

**Reply on RC3:**

The manuscript titled "Enhancing Single-Precision with Quasi Double-Precision: Achieving Double-Precision Accuracy in the Model for Prediction Across Scales-Atmosphere (MPAS-A) version 8.2.1" explores the use of the quasi-double precision algorithm by Møller et al. (1965) within a numerical modelling framework. By implementing the algorithm in MPAS-A across various case studies and idealized scenarios, the study highlights its potential to lower computational costs compared to double-precision approaches while enhancing accuracy over single-precision approaches.

Overall, the manuscript is well-organized and relatively clear, but there are several areas that could benefit from further clarification and elaboration. In particular, I have concerns regarding 1) the practical significance of the reported improvements, 2) the lack of detail and rationale in the model setup, and 3) the insufficient explanation or justification for the choice of metrics. Based on these issues, I recommend major revisions before the manuscript can be reconsidered for publication.

Thank you for your thorough review and valuable feedback on our manuscript, we appreciate your positive comments on the organization and clarity of the manuscript.

We understand your concerns regarding the practical significance of the reported improvements, we have carefully considered your feedback and addressed each of these points in the detailed responses below. We believe the revisions made in response to your comments significantly strengthen the manuscript and clarify the contributions of our work.

We are grateful for your insights and believe that addressing your concerns has resulted in a substantially improved manuscript. We look forward to your further assessment.

**Below, you find a list of my main concerns:**

1. The practical impact of the improvements achieved with the quasi-double precision algorithm compared to the single-precision approach remains unclear to me. While the authors demonstrate large relative accuracy gains compared to single-precision, it is not evident that these improvements are meaningful in absolute terms. For example, in the idealized scenario of Section 3.1, the RMSE of surface pressure improves by only 10^-2 Pa at longer lead times (12+ days), with negligible improvements at shorter times. Similar patterns emerge in Section 3.3. Given that these improvements are an order of magnitude smaller than the World Meteorological Organization's (WMO) recommended measurement uncertainty for atmospheric pressure (e.g., https://gcos.wmo.int/en/essential-climate-variables/pressure), it might be difficult to justify the additional computational costs. I encourage the authors to elaborate on the significance of their results in absolute terms from both a practical and forecasting perspective.

A related point: Are current operational implementations of MPAS-A defaulting to double- or single-precision for the variables discussed in this manuscript? Given the results, if double precision is currently the standard, the authors might want to emphasise the potential of quasi-double precision

as a substitute for double precision, rather than focusing primarily on its benefits over single-precision.

Thank you for your insightful comments and suggestions. We appreciate your careful review and have revised the manuscript accordingly. We address your points as follows:

(1) This research is based on double precision testing as the benchmark experiment, we agree that highlighting the computational advantages of quasi-double precision compared to double precision is crucial. We apologize for the omission of this discussion in the original manuscript. As you suggested, we have now added a detailed analysis of computational costs in the revised manuscript, comparing SGL, DBL and QDP. This analysis demonstrates that QDP achieves accuracy very close to DBL while significantly reducing computational costs. We provide further details on this analysis in our response to your second point below.

(2) You correctly pointed out that the SGL errors already meet the WMO's target values, and our primary focus should be the comparison between QDP and DBL. We included the comparison with SGL for the following reasons: within the MPAS-A model, the quasi double-precision algorithm is applied only to the basic and tend processes, while other parts remain in their original single-precision implementation. In essence, the quasi double-precision algorithm is applied to a limited portion of the dynamical core. Since MPAS-A supports single-precision calculations, we aimed to investigate the impact of the quasi double-precision algorithm on both computational cost and accuracy relative to the existing single-precision implementation. A modest increase in computational cost coupled with a substantial improvement in accuracy would suggest the potential of quasi double-precision as a replacement for single precision in scenarios where storage limitations are a major concern. We have retained the comparison to SGL to provide this context but have strengthened the emphasis on the comparison with DBL throughout the manuscript as you recommended.

(3) You correctly pointed out that while the errors observed within the current 12+day simulations appear relatively small, these could accumulate over longer time scales. Therefore, we plan to conduct extended simulations (monthly to yearly) in future research to investigate the potential for error accumulation and to determine the feasibility of using quasi-double precision as a replacement for double precision in such long-term applications. As you suggested, our future work will prioritize direct comparisons between the QDP and DBL in these extended simulations to fully characterize the differences and evaluate the long-term viability of our proposed method.

2. The experimental setup lacks sufficient detail and justification. It is unclear why the authors chose these particular case studies and idealised scenarios, especially in the applications to real data. This makes it difficult to assess the generalisability of the results to other atmospheric conditions or variables. Additionally, a more detailed comparison of computational costs among the three algorithms (single-precision, quasi-double precision, and double precision) is necessary, as improvements in accuracy may not be feasible in practice if they come with substantial increases in computational expense.

Thank you for your constructive feedback. We address your concerns regarding the case selection and computational cost as follows:

(1) Case Selection and Justification: We apologize that this rationale was not clearly articulated in the original manuscript. We will explain it. Thank you for your suggestions. We would like to ask whether you believe the descriptions and tables mentioned should be added to the main text of the manuscript. We look forward to your further input on this matter.

(2) Computational cost: We will add it in the paper to describe the computational performance. The size of the computational performance will be represented in terms of runtime, and we will discuss the runtime for each case in tabular form.

The type of initial conditions in MPAS-A cases (Table 1) can be found in "MPAS-Atmosphere Model User's Guide Version 8.1.0" (can be download on MPAS website). Only configurations for the idealized cases 2, 5, and 6 are provided on the official website. To avoid redundancy, the results for Case 6 are not shown, as they were similar to those obtained for Cases 2 and 5. For real-data simulations, case 7 as shown in Table. We explored different resolutions for real-data simulations to assess the impact of grid refinement on the performance of the quasi-double precision algorithm. Our focus is on numerical precision in global models, where boundary condition effects are absent. Thus, we do not consider regional models like case 8 and case 9 in this study.

**Table 1**. Cases in MPAS-A.

| type of initial conditions to create: |
| --- |
| 1.  Jablonowski and Williamson baroclinic wave (no initial perturbation) |
| 2.  Jablonowski and Williamson baroclinic wave (with initial perturbation) |
| 3.  Jablonowski and Williamson baroclinic wave (with normal-mode perturbation) |
| 4.  squall line |
| 5.  super-cell |
| 6.  mountain wave |
| 7.  real-data initial conditions from, e.g., GFS |
| 8.  surface field (SST, sea-ice) update file for use with real-data simulations |
| 9.  lateral boundary conditions update file for use with real-data simulations |

**Line 286~292:** In comparison with the SGL, although there is a slight increase in runtime, it is minimal, at only 6.0% (Jablonowski and Williamson baroclinic wave), 0.3% (Super-cell), 2.2% (Real data with resolution of 120km) and 17.8% (Real data with resolution of 240km) (Table 3). This slight increase is attributed to the addition of a small number of global variable arrays when using quasi double-precision. And compared to DBL, QDP demonstrated relatively better performance across different cases, reducing the runtime by 28.6% (Jablonowski and Williamson baroclinic wave), 28.5% (Super-cell), 21.1% (Real data with resolution of 120km) and 5.7% (Real data with resolution of 240km) (Table 3).

**Table 3.** Comparative Analysis of Computational Efficiency: DBL vs SGL vs **QDP**

| Case name | DBL | SGL | QDP (Proposed) | | |
|---|---|---|---|---|---|
| | Runtime | Runtime | Runtime | vs DBL | vs SGL |
| JW wave | 1768 s | 1191 s | **1263 s** | -28.6% | +6.0% |
| SC | 1507 s | 1073 s | **1077 s** | -28.5% | +0.3% |
| RD-120 | 19126 s | 14765 s | **15092 s** | -21.1% | +2.2% |
| RD-240 | 1397 s | 1118 s | **1317 s** | -5.7% | +17.8% |

Note: JW wave = Jablonowski & Williamson baroclinic wave;

SC = Super-cell; RD-120/240 = Real data with total domain size of 120/240 km

**Line 7 ~ 9:** The bias of surface pressure are reduced respectively by 68%, 75%, 97% and 96% in cases, the memory has been reduced by almost half, while the computation increases only 6.0%, 0.3%, 2.2%, and 17.8% respectively, significantly reducing computational cost.

3. The manuscript currently employs only grid-point level and "spatial RMSE" as comparison metrics, but the term "spatial RMSE" is not clearly defined. I recommend the following:

1) Clearly define "spatial RMSE," particularly when used as a summary statistic over a larger domain. For example, are the authors applying any latitude-based weighting?

2) Justify the use of this metric and consider comparing the algorithms using additional relevant metrics. If alternative metrics are not applicable, please explain why they were not used.

Thank you very much for your insightful comments regarding the metrics used in our study. Your feedback has been incredibly helpful in improving the clarity of our analysis. We address your concerns in three parts:

(1) We apologize for the omission of a clear definition of spatial RMSE in the original manuscript. As your suggestion, we have now included a detailed explanation of the spatial RMSE calculation in Section 3.1. This explanation includes the formula and a description of its application in our study, as detailed below.

(2) We selected spatial RMSE as a primary metric because it is widely used in the evaluation of atmospheric numerical models and effectively reflects model performance improvements. Studies such as Váňa et al. (2016), demonstrate its utility in assessing the accuracy of model simulations. Therefore, we believe spatial RMSE is an appropriate and informative metric for evaluating the impact of the quasi-double precision algorithm in our study.

(3) We appreciate your suggestion to include additional metrics. We decide to add a new metrics of Mean Absolute Error (MAE) to improve our analysis. To facilitate comparison across different cases, we have also included a table summarizing the spatial RMSE and MAE values for each case in the revised manuscript. The specific changes are detailed below.

**Line 195 ~ 221:** In this section, we introduce the Spatial RMSE and MAE(the accuracy indicators), and show results across four cases, include two ideal scenarios: Jablonowski and Williamson baroclinic wave and super-cell, as well as two real case ( with initial conditions generated using GFS data) using two different resolutions. By using the RMSE and MAE for quantitative

comparison, the differences between the benchmark and control experiments are used to evaluate the effectiveness of the quasi double-precision algorithm in reducing round-off error.

To quantify the difference between the simulations using SGL, QDP, and DBL, (used as the benchmark), we calculate the spatial root-mean-square error (RMSE). First, for each grid point, the temporal averages of the variables (e.g., surface pressure, 500hPa height) are computed across the entire simulation period for each experiment (SGL, QDP, and DBL). Then, the spatial RMSE is calculated as the root-mean-square difference between the temporally averaged fields of the control experiment (SGL or QDP) and the benchmark double-precision experiment (DBL), following (1):

$$\text{Spatial RMSE} = \sqrt{\frac{\sum_{i=1}^{N}(M_i - C_i)^2}{N}} \qquad (1)$$

Where, N is the total number of grid points, $M_i$ is the temporally averaged value at grid point i for the benchmark double-precision experiment, $C_i$ is the temporally averaged value at grid point i for the control experiment (SGL or QDP).

In addition to the spatial RMSE, we also calculate the Mean Absolute Error (MAE) to assess the magnitude of the difference between the control experiments (SGL and QDP) and the benchmark double-precision experiment (DBL), irrespective of the direction of the difference. Like the spatial RMSE calculation, we first compute the temporal average for each grid point across the entire simulation period for each experiment. The MAE is then calculated as the average absolute difference between the temporally averaged fields of the control experiment and the benchmark experiment, following (2):

$$\text{MAE} = \frac{1}{N} * \sum_{i=1}^{N}|M_i - C_i| \qquad (2)$$

where N is the total number of grid points, $M_i$ represents the temporally averaged value at grid point i for the benchmark double-precision experiment, and $C_i$ represents the temporally averaged value at grid point i for the control experiment (either SGL or QDP).

As shown in Table 1 (spatial RMSE) and Table 2 (MAE), the addition of the quasi-double precision algorithm consistently improves accuracy (compared to single precision) across all cases. For specific analysis, please refer to the following contents (The results of RMSE and MAE are consistent, so to avoid duplication, only the results of RMSE are analyzed in the following text).

**Table 1. The spatial RMSE** values of surface pressure compared to DBL for cases, unit: Pa. Note: JW wave = Jablonowski & Williamson baroclinic wave;SC = Super-cell; RD-120/240 = Real data with total domain size of 120/240 km.

| Case name | SGL | QDP |
|---|---|---|
| JW wave | $3.42 * 10^{-2}$ | $1.09 * 10^{-2}$ |
| SC | $8.80 * 10^{-4}$ | $2.27 * 10^{-4}$ |
| RD-120 | $6.33 * 10^{-2}$ | $2.25 * 10^{-3}$ |
| RD-240 | $6.68 * 10^{-2}$ | $2.25 * 10^{-3}$ |

**Table 2. The MAE** values of surface pressure compared to DBL for cases, unit: Pa. Note: JW wave = Jablonowski & Williamson baroclinic wave;SC = Super-cell; RD-120/240 = Real data with total domain size of 120/240 km.

| Case name | SGL | QDP |
|---|---|---|
| JW wave | $1.29 * 10^{-2}$ | $3.81 * 10^{-2}$ |
| SC | $8.79 * 10^{-4}$ | $2.26 * 10^{-4}$ |
| RD-120 | $5.38 * 10^{-2}$ | $1.95 * 10^{-3}$ |
| RD-240 | $5.52 * 10^{-2}$ | $1.94 * 10^{-3}$ |

**Minor concerns/technical corrections:**

- Figure 6a: The higher error with the quasi-double precision algorithm compared to the single-precision in this subpanel is confusing. If the error with the single-precision algorithm is anyway so small that it does not matter in practice, why was this case study selected, and why include this figure?

Thank you for bringing this to our attention. We sincerely apologize for the error in our data processing, which was also noted by Reviewer 1. We have carefully re-examined our code and identified a mistake. We have corrected the error in our data processing code and have now generated the revised results. The corrected figure is shown below (fig. 1), Figure 6(a) in manuscript:

[Figure]

**Figure 1.** The temporal evolution of spatially averaged difference of kinetic energy between DBL and SGL, as well as difference between DBL and QDP in case of super-cell.

I apologize for needing to provide additional information regarding this section. In all cases presented in the manuscript, the time-evolution plots (Figures 4, 6, and 8 in the manuscript) currently utilize kinetic energy and surface pressure, which correspond to the conservation of energy and mass, respectively. However, I realize that directly using total energy and total mass would offer a more accurate representation. Please see the figures 2, 3 and 4 showing the temporal variations of energy and mass for all cases. If you allow it, I would be happy to replace the existing Figures 4, 6, and 8 (in the manuscript) with these updated versions (figures 2, 3 and 4). I want to emphasize that this change would not affect the overall results or conclusions of the study, which remain consistent with those currently presented in the manuscript. We have received feedback from two reviewers so far, and Reviewer 1 has approved the replacement. We are eagerly awaiting your response. If all four reviewers agree, we will update the figure in the final version of the manuscript.

[Figure]

**Figure 2.** The temporal evolution of spatially averaged difference of (a) total energy, (b) total mass between DBL and SGL, as well as difference between DBL and QDP in case of Jablonowski and Williamson baroclinic wave.

[Figure]

**Figure 3.** The temporal evolution of spatially averaged difference of (a) total energy, (b) total mass between DBL and SGL, as well as difference between DBL and QDP in case of super-cell.

It is important to note that, for enhanced clarity and to facilitate a better understanding of the trend, the x-axis unit for the figure representing Case 7 has been changed to hours (Fig. 4).

[Figure]

**Figure 4.** The temporal evolution of spatially averaged difference of total energy between DBL and SGL, as well as difference between DBL and QDP in case of real data, with resolution of (a) 240 km × 240 km, (b) 120 km × 120 km.

- Figure 7: The phrase "the circle represents the most clear error" lacks objectivity, especially since other areas in the figure show similar errors. Consider rephrasing or finding a more precise way to characterize the comparison between errors in this case.

Thank you very much for your careful review and valuable comments on my manuscript. Regarding the issue you raised in the figure 7. We decide to adjust the color bar, after adjusting, the two distinct areas of improvement are now evident, compared to only one in the original figure. I sincerely apologize that the previous color bar setting was not appropriate. and the revised figure is shown as Figure 5 (corresponding to Figure 7 in the manuscript).

[Figure]

**Figure 5.** Perturbation theta in super-cell development at 5400s in the (a) DBL simulation, (b) SGL simulation and (c) QDP simulation (bias has reduced), unit: K, the circle represents the pattern bias (the same color means the consistent value).

- Please avoid using the term "significant" or "significantly" when not referring to statistically significant results to avoid potential confusion.

Thank you for this valuable feedback. We have corrected this oversight.

- Unclear phrase (line 95): "achieves basically consistent results comparable to those of double precision" – Do you mean that the accuracy is similar in a specific case, or that the quasi-double precision consistently performs similarly to double precision in most cases?

Thank you for raising this point. The phrase 'the single-precision version improved with the quasi double-precision algorithm achieves basically consistent results comparable to those of double precision. However, we acknowledge that the phrasing was ambiguous. We have revised the sentence to more accurately reflect this finding.    The revised sentence now reads:
**Line 103～105:** By using error compensation methods (quasi double-precision), we can maintain integration stability comparable to that applying double precision scheme while significantly reducing memory requirements by lowering the numerical precision of all variables and improved the accuracy comparable to that applying the single precision.

- Reproducibility note: Although the manuscript adheres to GMD's guidelines by providing a DOI to a permanent repository, no instructions are provided on how to use the code or what the various files and subfolders contain. I suggest including at least a brief README file that describes the content and organization of the repository, providing users with clearer guidance.

Thank you for your valuable suggestion regarding the code documentation. We have updated the supplemental material with a comprehensive description of the code structure and included a README file for each case, as you recommended. We apologize that this information was not initially referenced within the main manuscript. To ensure clarity and accessibility, we have also added explicit mentions of the code description and README files within the Code and data availability section of the manuscript, directing readers to the supplemental material for these details. Figure 6 shows a screenshot of the updated supplemental material, which now clearly includes these additions. Each case directory within the supplement contains a corresponding README file, an example of which is shown in Figure 7.

We believe these revisions significantly improve the transparency and reproducibility of our research. Thank you again for your helpful feedback.
**Line 329～333:** Code and data availability. Model code and plotting data related to this manuscript is available at: https://doi.org/10.5281/zenodo.13765422. Details regarding the code structure and instructions for running the code are provided in the supplementary material, which can be down-loaded and viewed in Fig. S1. This figure provides a visual overview of the code organization. The information of steps how to execute the simulations can be found in README file in each test case folder.

**Supplementary Information**

```
model
├────── DBL   --benchmark using double precision
├────── SGL   --control test using single precision
└────── QDP   --control test using single precision and quasi double-precision algorithm

test
├────── c2_DBL--benchmark using double precision. (The Jablonowski and Williamson baroclinic wave test case)
│       ├────── atmosphere_model      -- run the model
│       ├────── bwave_surface_p.ncl      -- script to produce plots of surface pressure and kinetic energy
│       ├────── init_atmosphere_model  --run to create initial conditions
│       ├────── namelist.atmosphere      --namelist options available when running the MPAS
│       ├────── namelist.init_atmosphere --namelist options available when running the MPAS initialization
│       ├────── README
│       ├────── stream_list.atmosphere.output --the output of MPAS
│       ├────── streams.atmosphere --the XML stream configuration file for an MPAS
│       └────── streams.init_atmosphere --The XML stream configuration file for an MPAS initialization
├────── c2_SGL--benchmark using double precision.
├────── c2_QDP--benchmark using double precision.
├────── c5_DBL--benchmark using double precision. (The super cell)
├────── c5_SGL--benchmark using double precision.
├────── c5_QDP--benchmark using double precision.
├────── c7_240km_DBL--benchmark using double precision. (The real data)
├────── c7_240km_SGL--benchmark using double precision.
├────── c7_240km_QDP--benchmark using double precision.
├────── c7_120km_DBL--benchmark using double precision.
├────── c7_120km_SGL--benchmark using double precision.
├────── c7_120km_QDP--benchmark using double precision.
```

**Figure S1.** The code layout of the research. The model part represent the model code including benchmark using double precision(DBL), control test using precision and control test using single precision(SGL) and quasi double-precision algorithm(QDP). The three models are run separately in 4 tests includes the Jablonowski and Williamson baroclinic wave test case, super cell, real data with 240km and real data with 120km. All configurations can be found in the test file. Only the case 7 use the GFS data, it can also be found under folder case7. Model code and plotting data related to this manuscript is available at: https://doi.org/10.5281/zenodo.13765421.

**Figure 6**. The code layout in supplement.

```
The supercell thunderstorm test case.

Steps to run this test case:

1. Link the init_atmosphere_model and atmosphere_model executables from the top-level MPAS directory.

2. Run init_atmosphere_model to create initial conditions.

3. Run atmosphere_model.

4. Run the supercell.ncl script to produce plots theta, vertical velocity, etc.
* * *
```

**Figure 7**. The screenshot of README file in case suppercell.

Vánˇa, F., Düben, P., Lang, S., Palmer, T., Leutbecher, M., Salmond, D., and Carver, G.: Single Precision in Weather Forecasting Models: An Evaluation with the IFS, Monthly Weather Review, 145, 495-502, doi:10.1175/MWR-D- 16-0228.1, 2016.

**Response to Reviewer #3**

**Reply on RC4:**

The abstract states: "Low precision computations can significantly reduce computational costs, but inevitably introduce rounding errors, which affect computational accuracy." However, it is not clearly defined what is meant by "low precision computations," and the statement that such computations inevitably affect accuracy may not always hold true.

We thank the reviewer for the valuable feedback. In the revised manuscript, we have added a clear definition of "low precision computations".

Regarding the assertion that "low precision computations inevitably affect accuracy," we appreciate the reviewer's insight, and we will revise the description in the abstract. The specific modifications are detailed below.

**Line 30~33:** Specifically, we define low precision computations as those that utilize a limited number of significant digits (less than 64 bits) during numerical operations, which can significantly reduce the computational resources required while potentially introducing rounding errors.
**Line 2~3:** While low precision computations can significantly reduce computational costs, they may introduce rounding errors that can affect computational accuracy under certain conditions.

Both the abstract and conclusions mention the computational impact of the proposed methods; however, these effects are not discussed further within the main body of the manuscript.

We appreciate your suggestion, following your insightful comment, we will:

1. Add it in the paper to describe the computational performance. The size of the computational performance will be represented in terms of runtime, and we will discuss the runtime for each case in tabular form.

2. Revise the description of computational performance in the abstract to reflect these updated and more accurate measurements.

**Line 286~292:** In comparison with the SGL, although there is a slight increase in runtime, it is minimal, at only 6.0% (Jablonowski and Williamson baroclinic wave), 0.3% (Super-cell), 2.2% (Real data with resolution of 120km) and 17.8% (Real data with resolution of 240km) (Table 3). This slight increase is attributed to the addition of a small number of global variable arrays when using quasi double-precision. And compared to DBL, QDP demonstrated relatively better performance across different cases, reducing the runtime by 28.6% (Jablonowski and Williamson baroclinic wave), 28.5% (Super-cell), 21.1% (Real data with resolution of 120km) and 5.7% (Real data with resolution of 240km) (Table 3).

**Table 3.** Comparative Analysis of Computational Efficiency: DBL vs SGL vs **QDP**

| Case name | DBL | SGL | QDP (Proposed) | | |
|---|---|---|---|---|---|
| | Runtime | Runtime | Runtime | vs DBL | vs SGL |
| JW wave | 1768 s | 1191 s | **1263 s** | -28.6% | +6.0% |
| SC | 1507 s | 1073 s | **1077 s** | -28.5% | +0.3% |
| RD-120 | 19126 s | 14765 s | **15092 s** | -21.1% | +2.2% |
| RD-240 | 1397 s | 1118 s | **1317 s** | -5.7% | +17.8% |

Note: JW wave = Jablonowski & Williamson baroclinic wave;

SC = Super-cell; RD-120/240 = Real data with total domain size of 120/240 km

**Line 7 ~ 9:** The bias of surface pressure are reduced respectively by 68%, 75%, 97% and 96% in cases, the memory has been reduced by almost half, while the computation increases only 6.0%, 0.3%, 2.2%, and 17.8% respectively, significantly reducing computational cost.

Additional context would be beneficial in distinguishing which differences are relevant. Including an uncertainty analysis would help place the magnitude of the errors into perspective. For example, Figure 8 shows significant differences in errors between low- and high-resolution grids, with these discrepancies appearing more impactful than those arising from precision changes alone.

Thank you for your insightful comments. As you mentioned, there are multiple factors contributing to the errors observed in our study. In addition to round-off errors associated with floating-point arithmetic, the choice of grid resolution also has a significant impact on bias. We appreciate your suggestion regarding the inclusion of an uncertainty analysis, and we will consider incorporating this into the revised manuscript to provide further context. We will add a description you can see as follows:

**Line 279 ~ 285:** In this research, we focus on the processes of summing the basic field and trends. When the resolution is increased, the basic field remains relatively unchanged; however, the trends become smaller. This characteristic aligns with the nature of adding large and small numbers, making the advantages of the quasi double-precision algorithm more pronounced. Thus, it is evident from Figure 8 that as the resolution increases, the improvement achieved by quasi double-precision algorithm also enhances.

On the other hand, it is important to note that the propagation of rounding errors is not immediately apparent over short time scales. However, as the number of iterations increases, these errors can become more significant. The quasi double-precision algorithm employs compensation mechanisms that help mitigate the propagation of these errors.

Section 2.3 would benefit from further detail on which parts of the code were modified and how these sections were selected for modification.

Thank you for your valuable feedback. To address this, we will:

1. Add the clear description of the solution method for the equations including temporal integration scheme and spatial discretization scheme. See Line 135 ~ 153 below.

2. Specify at which step the quasi double-precision algorithm is applied within the computation

process in section 2.3 and replace the figure (corresponding to Figure 3 in the manuscript) and explain this process using formulas and explanations. See Line 161 ~ 179 below.

**Line 135 ~ 153:** The MPAS-A solves the fully compressible, nonhydrostatic equations of motion (Skamarock et al. 2011). The spatial dis- cretization uses a horizontal (spherical) centroidal Voronoi mesh with a terrain-following geometric-height vertical coordinate and C-grid staggering for momentum. The temporal discretization uses the explicit time-split Runge–Kutta technique from Wicker and Skamarock (2002) and Klemp et al. (2007).

The algorithm applied here primarily addresses the rounding error compensation between large and small numbers in addi- tion. Currently, it is only applicable to the time integration process and has not been implemented in the spatial discretization process. Therefore, this section will provide a detailed introduction to the time integration scheme. For the spatial discretization scheme, please refer to Skamarock et al. (2011), and it will not be introduced upon here.

The formulation of the scheme can be considered in on dimension as equation Wicker and Skamarock (2002):

$$\frac{\partial \phi}{\partial t} = RHS_\phi \tag{1}$$

The variable $\phi$ represents any prognostic variable in the prognostic equations, while RHS represents the right-hand side of the prognostic equations (i.e., the spatial discretization equation). In MPAS-A, a forward-in-time finite difference is used, and it can be written as Eq. (2):

$$\frac{{\phi_i}^{n+1} - {\phi_i}^n}{\Delta t} = RHS_\phi \tag{2}$$

Where superscript represent the time step, and subscript represent the position of grid zone.

The two-order Runge-Kutta time scheme is used in MPAS-A as described in Gear et al. (1971):

$$\phi^* = \phi^t + \frac{\Delta t}{2} * RHS(\phi^t) \tag{3}$$

$$\phi^{**} = \phi^t + \frac{\Delta t}{2} * RHS(\phi^*) \tag{4}$$

$$\phi^{t+\Delta t} = \phi^t + \Delta t * RHS(\phi^{**}) \tag{5}$$

**Line 161 ~ 179:** According to Equation Eq.(3), (4) and (5), it can be observed that in the time integration scheme, each step involves the process of adding tends on the basic field φt. In numerical models, the basic field is generally much larger than the tends, which aligns with the principles of numerical computation regarding the addition of large and small numbers, as well as the time integration process. It is important to note that the quasi double-precision algorithm currently only addresses time integration and has not been validated during the spatial discretization process. The spatial discretization primarily involves subtraction, specifically the subtraction of a small number from a large number or the subtraction of two close values. Whether this algorithm is applicable in spatial discretization remains uncertain, therefore, we will not apply it in this context.

Based on the application principles of the algorithm, which involve the processes of adding large and small numbers as well as the time integration process, we have established a strategy for applying the quasi double-precision algorithm within the MPAS-A. Specific improvements are provided based on the predictive equations:

$$\frac{\partial \boldsymbol{V}_H}{\partial t} = -\frac{\rho_d}{\rho_m}\left[\nabla_\zeta\left(\frac{p}{\zeta_z}\right) - \frac{\partial z_H p}{\partial \zeta}\right] - \eta \boldsymbol{k} \times \boldsymbol{V}_H - \boldsymbol{v}_H \nabla_\zeta \cdot \boldsymbol{V} - \frac{\partial \Omega v_H}{\partial \zeta} - \rho_d \nabla_\zeta K - eW cos\alpha_r -$$

$$\frac{v_H W}{r_e} + \boldsymbol{F}_{\boldsymbol{V}_H} \tag{6}$$

$$\frac{\partial W}{\partial t} = -\frac{\rho_d}{\rho_m}\left[\frac{\partial p}{\partial \zeta} + g\tilde{\rho}_m\right] - (\nabla \cdot \boldsymbol{v}W)_\zeta + \frac{uU+vV}{r_e} + e(U cos\alpha_r - V sin\alpha_r) + F_W \tag{7}$$

$$\frac{\partial \Theta_m}{\partial t} = -(\nabla \cdot \boldsymbol{V}\theta_m)_\zeta + F_{\Theta_m} \tag{8}$$

$$\frac{\partial \tilde{\rho}_d}{\partial t} = -(\nabla \cdot \boldsymbol{V})_\zeta \tag{9}$$

The meaning of each variable in the equations exactly follows Skamarock et al. (2012), so that we don't repeating explanation. For a numerical model, the most crucial variables are the prognostic variables. Therefore, In the MPAS-A model we applied the quasi double-precision algorithm to the time integration process of these prognostic variables, including horizontal momentum ($\boldsymbol{V}_H$), dry air density ($\tilde{\rho}_d$), potential temperature ($\Theta_m$) and vertical velocity ($W$), that is, the process in red of Eq. (6), (7), (8) and (9). (Only the predictive equations for the dynamic core are presented here, without the scalar transport.)

In Section 3.1, the differences between cases only emerge after 10 days of integration. It would be valuable to contextualize these differences with the error growth from other potential sources of uncertainty.

Thank you for your valuable suggestion. In response, we will add an analysis of other sources of error. The content will include the following:

**Line 239~245:** The sources of unpredictability, as noted by Bauer et al. (2015), include instabilities that inject chaotic 'noise' at small scales 240 and the upscale propagation of their energy. For the cases examined, both SGL and QDP begin to exhibit errors after 10 days of integration. These errors arise from factors such as rounding errors due to reduced numerical precision and energy loss during the propagation process. The quasi double-precision algorithm can reduce the impacts of these errors.

While we acknowledge other potential sources of uncertainty, such as initial condition errors, we have not conducted an in- depth study on them in this research. Our primary focus remains on evaluating the improvements provided by the compensation algorithm in addressing rounding errors.

Section 3.2 states that "the errors are very small and can be ignored." More context is needed here to help determine which differences are meaningful.

Thank you for your insightful comment regarding Figure 6(a) of section 3.1. According to RC1, we carefully reviewed our code based on your comments and identified that the issue was a problem with the data processing and plotting. We have corrected this issue, and the revised Figure 1 is provided below, it can be seen, the average bias between DBL and QDP is smaller than DBL and SGL.

[Figure]

**Figure 1.** The temporal evolution of spatially averaged difference of kinetic energy between DBL and SGL, as well as difference between DBL and QDP in case of super-cell.

In Section 3.3, the authors note that "Differences in error begin to emerge after 500 steps." This could be strengthened by comparing this error growth to that of other sources of uncertainty, some of which may become relevant earlier in the integration process.

Thank you for your insightful suggestion. We will enhance our discussion in Section 3.3 by adding a comparison of error growth with other sources of uncertainty. The added content will be as follows:

**Line 262-266:** Consistent with the analysis presented in Section 3.2, errors are relatively small in the early stages and begin to emerge after 140 hours. This increase is attributed to the accumulation of round-off errors and energy loss over time. The effects become more pronounced beyond 140 hours. Overall, the quasi double-precision algorithm demonstrates a certain level of improvement in addressing these errors. The case with the total domain of 240 km × 240 km (Figs. 7a) show the larger error than 120 km × 120 km (Figs. 7b), and the error can be reduced in QDP caused by SGL.

**Minor Comments**

Line 19 – The authors reference a 2015 source to indicate that systems are expected to grow. While this is still valid, the "current systems" referenced in 2015 are no longer today's current systems.

Thank you for your suggestions, we have revised it by modifying the "current systems" to "2015's systems" in Line 22.

Line 47 – The mixed-precision reference for NEMO notes that "95.8% of the 962 variables could be computed using half precision," though the publication itself refers to single precision.

Thank you for your suggestions, we have revised it by modifying the "half precision" to "single precision" in Line 53.

Line 143 – It is unclear if this version of MPAS-A is indeed the only one capable of utilizing single-precision.

We appreciate your comment. Upon reviewing the official documentation and user guide, we found that MPAS-A can be complied and run in single-precision since version 2.0. We apologize for the oversight and will revise the description accordingly.

**Line 159:** It is not the only version that supports single precision, but the latest version currently released.

Figures 5, 7, 9, 10, 11, and 12 – Alternative colormaps are suggested for all figures containing maps.

Thank you for your suggestion regarding the colormaps. We have made the requested modifications to the colormaps in all relevant figures. The updated figures are provided as Figure 1(corresponding to Figure 5 in the manuscript), Figure 2(corresponding to Figure 7 in the manuscript), Figure 3(corresponding to Figure 9 in the manuscript), Figure 4(corresponding to Figure 10 in the manuscript), Figure 5(corresponding to Figure 11 in the manuscript), Figure 6(corresponding to Figure 12 in the manuscript).

[Figure]

**Figure 1.** Spatial distributions of averaged (1-15days) difference of surface pressure (units: Pa) between DBL and (a) SGL simulations, (b) QDP simulations (round-off error has reduced) in case of Jablonowski and Williamson baroclinic wave.

[Figure]

**Figure 2.** Perturbation theta in super-cell development at 5400s in the (a) DBL simulation, (b) SGL simulation and (c) QDP simulation (bias has reduced), unit: K, the circle represents the pattern bias (the same color means the consistent value).

[Figure]

**Figure 3.** Spatial distributions of averaged (1-15days) difference of surface pressure (units: Pa) between DBL and (a) SGL simulation, (b) QDP simulation (resolution: 240 km × 240 km). The RMSE of surface pressure between DBL and (a) SGL simulation is $6.68 \times 10^{-2}$ Pa, (b) QDP simulation is $2.25 \times 10^{-3}$ Pa. (The color bars of (a) and (b) are different)

[Figure]

**Figure 4.** Spatial distributions of averaged (1-15days) difference of 500hPa height (units: m) between DBL and (a) SGL simulation, (b) QDP simulation (resolution: 240 km × 240 km). The

RMSE of 500hPa height between DBL and (a) SGL simulation is $2.80 \times 10^{-1}$ m, (b) QDP simulation is $1.40 \times 10^{-1}$ m (round-off error has reduced).

[Figure]

**Figure 5.** distributions of averaged (1-15days) difference of surface pressure (units: Pa) between DBL and (a) SGL simulation, (b) QDP simulation (resolution: 120 km × 120 km) (round-off error has reduced). The RMSE of surface pressure between DBL and (a) SGL simulation is $6.33 \times 10^{-2}$ Pa, (b) QDP simulation is $2.25 \times 10^{-2}$ Pa. (The color bars of (a) and (b) are different)

[Figure]

**Figure 6.** Spatial distributions of averaged (1-15days) difference of 500 hPa height (units: m) between DBL and (a) SGL simulation, (b) QDP simulation (resolution: 120 km × 120 km). The RMSE of 500 hPa height between DBL and (a) SGL simulation is $4.35 \times 10^{-3}$ m, (b) QDP simulation is $1.90 \times 10^{-3}$ m. (The color bars of (a) and (b) are different)

Line 241–242 – The phrase "but the process is more sensitive for the precision" lacks clarity and would benefit from rephrasing.

Thank you for your valuable suggestion. I will rephrase this statement for improved clarity in the revised manuscript. See bellows:

**Line 323~325:** Currently, the quasi double-precision algorithm is only implemented in the time integration scheme of the dynamic core of the MPAS-A model, without factoring in tracer transport.

However, the tracer transport process involves numerous operations where large and small numbers are added together, making it more sensitive to precision requirements.

Bauer, P., Thorpe, A., and Brunet, G.: The quiet revolution of numerical weather prediction, Nature, 525(7567):47-55, doi:10.1038/nature14956, 2015.

Skamarock, W. C., Klemp, J. B., Duda, M. G., Fowler, L. D., Park, S. H., and Ringler, T. D.: A Multiscale Nonhydrostatic Atmospheric Model Using Centroidal Voronoi Tesselations and C-Grid Staggering, Monthly Weather Review, 240(9):3090-3105, doi:10.1175/MWR- D-11-00215.1, 2011.

Wicker, L. J., & Skamarock, W. C.: Time-splitting methods for elastic models using forward time schemes. Monthly Weather Review, 130(8), 2088. https://www.proquest.com/scholarly-journals/time-splitting-methods-elastic-models-using/docview/198148677/se-2, 2002.

Gear, C. W.: Numerical initial value problems in ordinary differential equations[M]. Englewood Cliffs, N.J: Prentice-Hall, 1971.

**Response to Reviewer #4**

**Reply on RC5:**

**Specific comments:**

- The algorithm description needs more attention and description in the text. By comparison, the description of MPAS seems much better detailed and it could be argued that much of it is less important. I recommend finding ways to enhance the content of section 2.1.

Thank you for your valuable feedback. The algorithm itself is proposed by Møller (1965), and it only apply in the step-by-step integration of ordinary differential equations, how to apply it in numerical models is core content, so to address this, we will:

1. Add the clear description of the solution method for the equations including temporal integration scheme and spatial discretization scheme in section 2.2. Moreover, the algorithm applied here primarily addresses the rounding error compensation between large and small numbers in addition. Currently, it is only applicable to the time integration process and has not been implemented in the spatial discretization process. I apologize for not mentioning that part in the manuscript. I will revise and provide the supplement. See below.

2. Enhance the algorithm description in Section 2.3. Specify at which step the quasi double-precision algorithm is applied within the computation process in section 2.3 and replace the figure (corresponding to Figure 3 in the manuscript) and explain this process using formulas and explanations. See below.

3. we will select a representative equation and include a detailed algorithmic description of its iterative calculation process in the Supplement (see Figure 1). We believe this addition will be beneficial for readers who wish to gain a deeper understanding of the computational aspects of our method.

```
Basic field: u_basic, Tend: u_tend        Basic field: u_basic, Tend: u_tend
For rk_step from 1 to 3:                   Define array: u_compensation
    u_basic = u_basic + u_tend             Initialize u_c to 0.0
                                           Define reals: u_sum, u_big, u_small
                                           Do rk_step from 1 to 3:
                                               If abs(u_basic) > abs(u_tend):
                                                   u_big = u_basic
                                                   u_small =  u_tend
                                               Else:
                                                   u_big =  u_tend
                                                   u_small =  u_basic
                                               end if
                                               u_small += u_compensation
                                               u_sum = u_big + u_small
                                               u_ compensation = (u_small - (u_compensation - u_big))
                                                                 + (u_big - (u_compensation - u_big))
                                               u_basic = u_sum
                                               ...
                                           End do rk_step
```

**Figure 1.** The pseudo-code for variable of U.

**Line 135～153:** The MPAS-A solves the fully compressible, nonhydrostatic equations of motion

(Skamarock et al. 2011). The spatial dis- cretization uses a horizontal (spherical) centroidal Voronoi mesh with a terrain-following geometric-height vertical coordinate and C-grid staggering for momentum. The temporal discretization uses the explicit time-split Runge–Kutta technique from Wicker and Skamarock (2002) and Klemp et al. (2007).

The algorithm applied here primarily addresses the rounding error compensation between large and small numbers in addi- tion. Currently, it is only applicable to the time integration process and has not been implemented in the spatial discretization process. Therefore, this section will provide a detailed introduction to the time integration scheme. For the spatial discretization scheme, please refer to Skamarock et al. (2011), and it will not be introduced upon here.

The formulation of the scheme can be considered in on dimension as equation Wicker and Skamarock (2002):

$$\frac{\partial \emptyset}{\partial t} = RHS_\emptyset \tag{1}$$

The variable $\emptyset$ represents any prognostic variable in the prognostic equations, while RHS represents the right-hand side of the prognostic equations (i.e., the spatial discretization equation). In MPAS-A, a forward-in-time finite difference is used, and it can be written as Eq. (2):

$$\frac{\emptyset_i^{n+1} - \emptyset_i^n}{\Delta t} = RHS_\emptyset \tag{2}$$

Where superscript represent the time step, and subscript represent the position of grid zone.

The two-order Runge-Kutta time scheme is used in MPAS-A as described in Gear et al. (1971):

$$\emptyset^* = \emptyset^t + \frac{\Delta t}{2} * RHS(\emptyset^t) \tag{3}$$

$$\emptyset^{**} = \emptyset^t + \frac{\Delta t}{2} * RHS(\emptyset^*) \tag{4}$$

$$\emptyset^{t+\Delta t} = \emptyset^t + \Delta t * RHS(\emptyset^{**}) \tag{5}$$

**Line 161～179:** According to Equation Eq.(3), (4) and (5), it can be observed that in the time integration scheme, each step involves the process of adding tends on the basic field φt. In numerical models, the basic field is generally much larger than the tends, which aligns with the principles of numerical computation regarding the addition of large and small numbers, as well as the time integration process. It is important to note that the quasi double-precision algorithm currently only addresses time integration and has not been validated during the spatial discretization process. The spatial discretization primarily involves subtraction, specifically the subtraction of a small number from a large number or the subtraction of two close values. Whether this algorithm is applicable in spatial discretization remains uncertain, therefore, we will not apply it in this context.

Based on the application principles of the algorithm, which involve the processes of adding large and small numbers as well as the time integration process, we have established a strategy for applying the quasi double-precision algorithm within the MPAS-A. Specific improvements are provided based on the predictive equations:

$$\frac{\partial \boldsymbol{V}_H}{\partial t} = -\frac{\rho_d}{\rho_m}\left[\nabla_\zeta\left(\frac{p}{\zeta_z}\right) - \frac{\partial z_H p}{\partial \zeta}\right] - \eta \boldsymbol{k} \times \boldsymbol{V}_H - \boldsymbol{\nu}_H \nabla_\zeta \cdot \boldsymbol{V} - \frac{\partial \Omega \nu_H}{\partial \zeta} - \rho_d \nabla_\zeta K - eW cos\alpha_r -$$

$$\frac{\nu_H W}{r_e} + \boldsymbol{F}_{\boldsymbol{V}_H} \tag{6}$$

$$\frac{\partial W}{\partial t} = -\frac{\rho_d}{\rho_m}\left[\frac{\partial p}{\partial \zeta} + g\tilde{\rho}_m\right] - (\nabla \cdot \boldsymbol{\nu}W)_\zeta + \frac{uU+vV}{r_e} + e(U cos\alpha_r - V sin\alpha_r) + F_W \tag{7}$$

$$\frac{\partial \Theta_m}{\partial t} = -(\nabla \cdot \boldsymbol{V}\theta_m)_\zeta + F_{\Theta_m} \tag{8}$$

$$\frac{\partial \tilde{\rho}_d}{\partial t} = -(\nabla \cdot \boldsymbol{V})_\zeta \tag{9}$$

The meaning of each variable in the equations exactly follows Skamarock et al. (2012), so that we don't repeating explanation. For a numerical model, the most crucial variables are the prognostic variables. Therefore, In the MPAS-A model we applied the quasi double-precision algorithm to the time integration process of these prognostic variables, including horizontal momentum ($\boldsymbol{V}_H$) , dry air density ($\tilde{\rho}_d$), potential temperature ($\Theta_m$) and vertical velocity ($W$) , that is , the process in red of Eq. (6), (7), (8) and (9). (Only the predictive equations for the dynamic core are presented here, without the scalar transport.)

- For clarification, is the QDP runtime is 2% more expensive compared to the DBL runtime or the SGL runtime? Also how much reduction in runtime was there for the SGL version compared to the DBL version? This can be used to give context to what benefits are achieved for the more noticeable errors in SGL.

Thanks for your suggestion, following your insightful comment, we found the previously reported figures were indeed rough estimations, so we have re-evaluated the exact computational performance using a measurement tool to determine the runtimes. So, we will:

3. Add a dedicated section in the paper to describe the computational performance. The size of the computational performance will be represented in terms of runtime, and we will discuss the runtime for each case in tabular form.
4. Revise the description of computational performance in the abstract to reflect these updated and more accurate measurements.

**Line 286~292:** In comparison with the SGL, although there is a slight increase in runtime, it is minimal, at only 6.0% (Jablonowski and Williamson baroclinic wave), 0.3% (Super-cell), 2.2% (Real data with resolution of 120km) and 17.8% (Real data with resolution of 240km) (Table 3). This slight increase is attributed to the addition of a small number of global variable arrays when using quasi double-precision. And compared to DBL, QDP demonstrated relatively better performance across different cases, reducing the runtime by 28.6% (Jablonowski and Williamson baroclinic wave), 28.5% (Super-cell), 21.1% (Real data with resolution of 120km) and 5.7% (Real data with resolution of 240km) (Table 3).

**Table 3.** Comparative Analysis of Computational Efficiency: DBL vs SGL vs **QDP**

| Case name | DBL | SGL | QDP (Proposed) | | |
|---|---|---|---|---|---|
| | Runtime | Runtime | Runtime | vs DBL | vs SGL |
| JW wave | 1768 s | 1191 s | **1263 s** | -28.6% | +6.0% |
| SC | 1507 s | 1073 s | **1077 s** | -28.5% | +0.3% |
| RD-120 | 19126 s | 14765 s | **15092 s** | -21.1% | +2.2% |
| RD-240 | 1397 s | 1118 s | **1317 s** | -5.7% | +17.8% |

Note: JW wave = Jablonowski & Williamson baroclinic wave;

SC = Super-cell; RD-120/240 = Real data with total domain size of 120/240 km

**Line 7 ~ 9:** The bias of surface pressure are reduced respectively by 68%, 75%, 97% and 96% in cases, the memory has been reduced by almost half, while the computation increases only 6.0%, 0.3%, 2.2%, and 17.8% respectively, significantly reducing computational cost.

- Given errors appear to become noticeably larger beyond 10+ days, how are these figures dependent with different spatial time average lengths? Do they all only look drastically different because of errors after round off error becomes significant around day 10+? Is this error tied to simulation length or is this more determined by the number of time steps taken?

Thank you for your insightful questions. As you noted, errors become significantly larger after approximately 10 days due to factors such as round-off errors arising from reduced numerical precision and energy loss during the propagation process. Around day 10, the accumulation of rounding errors becomes pronounced, continuing to accumulate over time, which can ultimately lead to model instability.

In this study, we did not conduct a comparative analysis involving different time step lengths; therefore, we cannot definitively conclude whether the observed errors are related to the chosen time step sizes. However, round-off errors do accumulate with the simulation length.

We appreciate your suggestion, and in future research, we will focus on whether integration step lengths have an impact on error growth.

- In some experiments, the reduction of error is a large percentage but feels like a small error. In general to the reader, it's not very clear that reducing these errors will matter that much. For example, while the error is reduced using QDP in a lot of cases, reduction of the surface pressure (hPa) error by a RSME of 2.8x10-1 to 1.4x10-1 seems like a small amount. Likewise for other sources of error and that this all needs context. It is important to consider these errors relative to other sources of model errors.

Thank for your suggestion. The core of this study is the introduction of error compensation methods (quasi double-precision). By using error compensation methods (quasi double-precision), we can maintain integration stability comparable to that applying double precision scheme while significantly reducing memory requirements by lowering the numerical precision of all variables and improved the accuracy comparable to that applying the single precision. This approach not only reduces communication pressure but also allows for substantial increases in computational speed

through vectorization optimization. So, we will add a section to describe it (see 3.4 Computational performance)

And we will add an analysis of other sources of error. The content will include the following:

**Line 239~245:** The sources of unpredictability, as noted by Bauer et al. (2015), include instabilities that inject chaotic 'noise' at small scales 240 and the upscale propagation of their energy. For the cases examined, both SGL and QDP begin to exhibit errors after 10 days of integration. These errors arise from factors such as rounding errors due to reduced numerical precision and energy loss during the propagation process. The quasi double-precision algorithm can reduce the impacts of these errors.

While we acknowledge other potential sources of uncertainty, such as initial condition errors, we have not conducted an in- depth study on them in this research. Our primary focus remains on evaluating the improvements provided by the compensation algorithm in addressing rounding errors.

- Additionally, I was wondering if the authors could leverage the test case where grid resolution was also incorporated as the model discretization is a source of model error compared to the model error due to precision differences.

Thank you for your insightful comments. As you mentioned, there are multiple factors contributing to the errors observed in our study. In addition to round-off errors associated with floating-point arithmetic, the choice of grid resolution also has a significant impact on bias. We will add a description, you can see as follows:

**Line 279 ~ 285:** In this research, we focus on the processes of summing the basic field and trends. When the resolution is increased, the basic field remains relatively unchanged; however, the trends become smaller. This characteristic aligns with the nature of adding large and small numbers, making the advantages of the quasi double-precision algorithm more pronounced. Thus, it is evident from Figure 8 that as the resolution increases, the improvement achieved by quasi double-precision algorithm also enhances.

On the other hand, it is important to note that the propagation of rounding errors is not immediately apparent over short time scales. However, as the number of iterations increases, these errors can become more significant. The quasi double-precision algorithm employs compensation mechanisms that help mitigate the propagation of these errors.

- For the cases in 3.3, it is not clear why the decision to have the higher resolution simulation of 120 km x 120 km have the same time step of 720 seconds as the 240 km x 240 km case. This seems rather unusual as typically a reduction in grid cell size is accompanied by a proportionate reduction in time step size to maintain a consistent Courant number.

Thank you for your question. As you mentioned, it is common for time step sizes to decrease proportionally with grid resolution. However, in our experiments, we found that the simulation could run stably at the finer resolution (i.e., 120 km) without requiring a reduction in the time step size. To minimize errors arising from differences in resolution during data processing, we chose to maintain the same time step across both simulations.

**Minor comments:**

- Figure 1, Figure 2 and a lesser extent Figure 3: The algorithm are written in a way that is difficult to read. I suggest that it is better distinguished the text (i.e. "evaluation") from variables (i.e. s,u,v). In Figure 2, everything is appears quite mathematical while Figure 3 is quite the opposite. It would be beneficial to have these all looking the same.

Thank you for your valuable feedback. We have made the necessary modifications. For details on these changes, please refer to the responses provided in the first specific comments.

- Line 150: "so we close the scalar transport in all cases". Does this mean it's turned off/not solved? Unless "closed" is a common term in the MPAS community, I suggest the authors consider a term closer to "disable/disabled" instead of "close/closed" when talking about model processes that are not absent in the simulation.

Thank you for your insightful comment. We have revised it from "closed" to "turned off" in Line 180.

- Line 179: Has this set of simulation parameters been utilized in a previous published work as a test case? This would be preferred to stating it was on a website that is subject to change.

Thank you for your suggestion. We will revise this section to clarify the source of our simulation parameters. The modified content will state:

**Line 223~224**: This case is a deterministic initial-value test case for dry dynamical cores of atmospheric general-circulation models (Jablonowski and Williamson 2006).

- Line 195 states the resolution is 84 km x 84 km where I believe this is actually the total domain size and not resolution. Assuming my understanding is correct, how many grid cells are within this domain and/or what is the average grid cell size in this simulation? Is it the ~500 m in Klemp 2015?

Thank you for your question. As you pointed out, 84 km × 84 km refers to the total domain size. We will revise this in the manuscript. Within this domain, there are a total of 28,080 grid cells, which is not the 500 m that you mentioned.

- Line 197: While the errors in Figure 6(a) are indeed quite small, it may be beneficial to say how small relative to the actual values of kinetic energy? And I think a more clear statement explaining this figure's behavior would help clarify the plot as it's not the expected behavior throughout the manuscript that QDP does better in terms of error.

Thank you for your insightful comment regarding Figure 6(a) of section 3.1. According to RC1, we carefully reviewed our code based on your comments and identified that the issue was a problem with the data processing and plotting. We have corrected this issue, and the revised Figure 2 is provided below, it can be seen, the average bias between DBL and QDP is smaller than DBL and SGL.

[Figure]

**Figure 2.** The temporal evolution of spatially averaged difference of kinetic energy between DBL and SGL, as well as difference between DBL and QDP in case of super-cell.

- Figure 7:   The location of the Y (km) should be moved up near the axes (and should probably be labeled X?) and not under the color bar and Z is used which is typically thought of as a vertical direction is on the y axis. Color bar should probably have different scaling with a smaller range. Additionally why are the circles different colors between the different panels?

Thank you for your valuable suggestions. We have made the necessary modifications as follows, as seen in Figure 3 (which corresponds to Figure 7 in the manuscript):

1.  The axis labels have been updated.
2.  The position of the color bar has been adjusted.
3.  The range of the color bar has been modified for better scaling.

Furthermore, we have added a note in the caption to clarify the differences in circle colors between the various panels.

[Figure]

**Figure 3.** Perturbation theta in super-cell development at 5400s in the (a) DBL simulation, (b) SGL simulation and (c) QDP simulation (bias has reduced), unit: K, the circle represents the pattern bias (the same color means the consistent value).

- Figure 9 is one of the only figures that has color bar limits change between the SGL and QDP

panels. Given that quite often the color bar limit have been fixed ranges in most figures, a note in the caption may be beneficial for this.

Thank you for your suggestion. We have included a note in the caption of Figure 9 indicating that the color bars used in panels (a) and (b) are different. This will help clarify the distinction for readers.

- Figure 10(b) uses the same color bar as Figure 10(a) but reveals no detail because the same color bar limits are too large. I suggest that a new color bar range is selected (and it is then noted in the caption as being different in an and b as suggested for Figure 9).

Thank you for your suggestion. I believe you may be referring to Figure 11 rather than Figure 10. I will revise the color bar for Figure 11 to improve clarity and detail, as well as update the caption accordingly. Please see below for Figure 4 (corresponding to Figure 11 in the manuscript).

[Figure]

**Figure 4.** distributions of averaged (1-15days) difference of surface pressure (units: Pa) between DBL and (a) SGL simulation, (b) QDP simulation (resolution: 120 km × 120 km) (round-off error has reduced). The RMSE of surface pressure between DBL and (a) SGL simulation is $6.33 \times 10^{-2}$ Pa, (b) QDP simulation is $2.25 \times 10^{-2}$ Pa. (The color bars in (a) and (b) are different)

For code and data availability section, it is mentioned that "model code and plotting data" is available While I was able to find the versions of the model code, the simulation inputs and plotting scripts, I was unable to locate actual data. It's unclear if this is intended as the wording indicates "plotting data" and we should rather anticipate running the model to produce the data.

Thank you for your valuable feedback. Due to the large size of the output data generated by the model, we have not uploaded the actual output files. If you would like to obtain the processed output data, you can compile and run the model following the instructions in the README file, which will allow you to generate the output data.

To enhance clarity, I will revise the description in the "Code and Data Availability" section accordingly. The updated wording is provided below.

**Line 329-332:** Code and data availability. Model code and plotting data related to this manuscript is available at: https://doi.org/10.5281/zenodo.13765422. Details regarding the code structure and

instructions for running the code are provided in the supplementary material, which can be downloaded and viewed in Fig. S1. This figure provides a visual overview of the code organization. The information of steps how to execute the simulations can be found in README file in each test case folder.

**Technical comments:**

- Negative exponents have been routinely miswritten throughout the manuscript such as 10-2 rather than 10-2.

Thank you for your suggestion. We have corrected these errors throughout the manuscript.

- Line 218: "The Spatial RMSE of with 120 km x 120km.." is missing the variable name, which appears to be surface pressure.

Thank you for your observation. I appreciate your attention to detail. I have corrected this in the revised manuscript (Line 272) by including the variable name, which is indeed surface pressure.

JABLONOWSKI C, WILLIAMSON D L.: A baroclinic instability test case for atmospheric model dynamical cores[J/OL]. Quarterly journal of the Royal Meteorological Society, 132(621C): 2943-2975. DOI:10.1256/qj.06.12, 2006.

---

## Author Response (AR3)

**Enhancing Single-Precision with Quasi Double-Precision: Achieving Double-Precision Accuracy in the Model for Prediction Across Scales-Atmosphere (MPAS-A) version 8.2.1**

Addressed Comments for Publication to

Geoscientific Model Development

by

The Authors

Dear Editor,

Enclosed herewith is the updated edition of our earlier submission titled "Enhancing Single-Precision with Quasi Double-Precision: Achieving Double-Precision Accuracy in the Model for Prediction Across Scales-Atmosphere (MPAS-A) version 8.2.1" assigned the manuscript number egusphere-2024-2986. We extend our sincere appreciation to you for offering invaluable feedback, which has significantly improved the overall quality of our manuscript. In this updated version, we have carefully addressed the suggestions provided by you. Herein, you will discover a summary of the modifications implemented, along with a thorough response to the your comments.

Sincerely,
The Authors

**Note:** To enhance the legibility of this response letter, editor's comments are typeset in boxes. Rephrased or added sentences are typeset in color. The respective parts in the manuscript are highlighted to indicate changes.

**Response to the editor**

**Comment 1.1**

Computational Efficiency Analysis: In the discussion section, further elaborate on the practical significance and potential impacts of the improved computational efficiency. For example, explain how it benefits large - scale climate simulations in terms of faster result acquisition and resource utilization. Also, discuss its stability and scalability across different hardware environments.

**Response 1.1:**

Thank you for your valuable feedback. I have made the necessary revisions to the discussion section (line 334 - 343 in manuscript) in response to your suggestion, further elaborating on the practical significance and potential impacts of the improved computational efficiency. And beneficials on large-scale simulations have been well discussed. Thank you for your guidance and support.

> This substantial memory reduction and relatively modest increase in computational cost can be attributed to the inherent trade-off between precision and performance. In large-scale numerical simulations, the impact of rounding errors cannot be ignored, which is why double precision is commonly employed to maintain the accuracy of results. However, double-precision computations significantly increase computation time and resource consumption, which is often impractical for large-scale simulations. By using QDP, we not only reduced memory and communication overhead but also enhanced scalability, particularly in ultra-large parallel simulations where internode communication can become a major performance bottleneck. Moreover, while our experiments did not include vectorization techniques, there is considerable potential for further performance improvements. In similar computational environments, vectorization optimization combined with QDP has been shown to offer significant computational efficiency advantages over double precision (Dmitruk et al., 2023). Although this study is limited by hardware and scale, vectorization will be a key area for future research, especially in fully exploiting the potential of HPC architectures.

**Comment 1.2**

Model and Algorithm Application: Explore the potential for applying the algorithm in the spatial discretization process in future work and provide examples of its application in different scenarios to aid understanding.

**Response 1.2:**

Thank you for your helpful suggestions. Unfortunately, the quasi double-precision algorithm is currently mainly applied in the process of time-integration, where its core function is to compensate for the larger round-off errors caused by the reduction in significant digits. The primary application addresses the issue of increased round-off errors in numerical computations due to the addition of numbers with vastly different magnitudes. However, it is uncertain whether this method would be beneficial for spatial discretization. In spatial discretization

algorithms, a common source of increased rounding errors occurs when subtracting two nearly equal numbers. We hope to make progress in addressing these types of errors in the future.
* * *
**Comment 1.3**

Experimental Setup and Case Selection: Add a subsection summarizing and comparing the characteristics and applicability of different cases. Mention future plans for expanding the case study range, if any.

**Response 1.3:**

Thank you for your insightful feedback, it's been very helpful. We selected these two ideal cases and the real-data case because they are the only complete datasets available for download on the MPAS website. In Section 2.4, we provide a description of the selected cases (line 192 - 200 in manuscript). We also plan to explore other cases in the future to validate the applicability of the algorithm across different scenarios.

> To assess the application effect of the quasi double-precision algorithm, We selected these two ideal cases and the real-data case because they are the only complete datasets available for download on the MPAS website:
>
> (1) **Jablonowski and Williamson baroclinic wave**: A deterministic initial-value test case (Jablonowski and Williamson 2006) for dry dynamical cores of atmospheric general-circulation models, is presented that assesses the evolution of an idealized baroclinic wave in the northern hemisphere. The primary objective is to assess the model's efficacy in replicating the typical dynamics of moist atmospheric conditions across various precision settings.
>
> (2) **Super-cell**: A reduced-radius sphere (Klemp et al. 2015) can be used to assess the behavior of nonhydrostatic processes in global atmospheric dynamical cores, as long as the simulated cases demonstrate good agreement with the corresponding flows in Cartesian geometry, for which analytical solutions are available.
>
> (3) **Real data**: with initial conditions generated using GFS data at 2014-09-10_00) using two different resolutions (total domain size of 120 km × 120 km and 240 km × 240 km).
* * *
**Comment 1.4**

Metrics Comparison: Analyze the correlation and complementarity between different metrics, such as when spatial RMSE or MAE is more suitable, and how to use them comprehensively for algorithm performance evaluation.

**Response 1.4:**

Thank you for your constructive feedback, it is greatly appreciated. We have added a section (line 227 - 233 in the manuscript) to explain the correlation and complementarity between these two different evaluation metrics. The added content is as follows:

RMSE is primarily used to measure the difference between predicted and actual values and is more sensitive to large errors. MAE calculates the average absolute prediction error, and is less sensitive to outliers than RMSE, making it more suitable for conventional error measurements. Therefore, the combination of RMSE and MAE provides a more comprehensive evaluation. When comparing the performance of different experiments, RMSE may be used to quantify differences in extreme values (such as temperature fluctuations, ocean current speeds, etc.), while MAE is used to assess the accuracy of the model's overall trend. Combining both provides a better reflection of the algorithm's performance advantages.

**Comment 1.5**

Figure Improvement: Optimize the overall layout of figures for better aesthetics and readability. Use a unified style for related figures. Enrich figure captions to guide readers in interpreting the information and its connection to the paper's key points.

**Response 1.5:**

I'm grateful for your thoughtful suggestions and the improvements they've brought. I have enriched and revised the figure captions accordingly.

[revised manuscript text omitted]

**Comment 1.6**

Terminology and Expression: Conduct a consistency check for terminology and expressions throughout the paper. Provide detailed explanations for technical terms upon first use to assist non - specialist readers.

**Response 1.6:**

Thank you very much for your constructive suggestion. I have carefully conducted a consistency check for terminology and expressions throughout the paper. Detailed explanations for technical terms have been provided upon first use to assist non-specialist readers. These changes aim to improve the clarity and accessibility of the paper.

**Comment 1.7**

Code and Data Availability: Include a FAQ section in the README file, addressing common issues like operating system - specific problems and key parameter settings.

**Response 1.7:**

Thank you very much for your valuable suggestion. We have added a FAQ section to the README file, and the specific content to be added is as follows:

> FAQ:
> Q1: How do I select different cases for the simulation, and how do I set the time step?
> A1: The case selection is controlled by modifying the config_init_case parameter in the namelist.init_atmosphere file. The time step is set by modifying the config_dt parameter in the namelist.atmosphere file.
> Q2: How can I modify the variables in the output file?
> A2: To modify the output variables, you need to edit the stream_list.atmosphere.output file. Currently, variables related to physical processes have been removed from the output.
> Q3: How can I optimize parameter settings when facing memory limitations or computational resource constraints?
> A3: You can optimize by setting the precision parameter to single in the streams.atmosphere and streams.init_atmosphere files. This will force real-valued fields to be written as 4-byte floating-point values, rather than the default 8-byte floating-point values.

**Comment 1.8**

Uncertainty Analysis: If possible, present a preliminary framework for future uncertainty analysis, including methods and expected results.

**Response 1.8:**

I sincerely appreciate your suggestion on uncertainty analysis. I will provide a preliminary framework for future uncertainty analysis in the paper as follows (line 301 - 305 in manuscript).

> Due to the current limitations on the MPAS-A website, which only provides a single set of terrain and initial condition fields for different experiments, our future plan is to request assistance from the MPAS-A website to construct different terrain and initial condition fields for a specific experiment. We aim to conduct sensitivity analysis, particularly for real data experiments. Provided that computational resources allow, we plan to carry out simulations with different resolutions and initial conditions. This will help lay the data foundation for future uncertainty analysis.

**Comment 1.9**

Logical Coherence: Check the transitions between sections for smoothness and ensure the paper structure adheres to academic norms with a reasonable distribution of content.

**Response 1.9:**

Thank you for your insightful comments, which have been very helpful. I have carefully reviewed the transitions between sections and made necessary revisions to ensure smooth flow. Additionally, I have adjusted the paper structure to better align with academic norms and ensured a more balanced distribution of content across the sections.

---

## Author Response (AR4)

**Enhancing Single-Precision with Quasi Double-Precision: Achieving Double-Precision Accuracy in the Model for Prediction Across Scales-Atmosphere (MPAS-A) version 8.2.1**

Addressed Comments for Publication to

Geoscientific Model Development

by

The Authors

Dear Editor,

Enclosed herewith is the updated edition of our earlier submission titled "Enhancing Single-Precision with Quasi Double-Precision: Achieving Double-Precision Accuracy in the Model for Prediction Across Scales-Atmosphere (MPAS-A) version 8.2.1" assigned the manuscript number egusphere-2024-2986. We extend our sincere appreciation to you for offering invaluable feedback, which has significantly improved the overall quality of our manuscript. In this updated version, we have carefully addressed the suggestions provided by you. Herein, you will discover a summary of the modifications implemented, along with a thorough response to the your comments.

Sincerely,
The Authors

**Note:** To enhance the legibility of this response letter, editor's comments are typeset in boxes. Rephrased or added sentences are typeset in color. The respective parts in the manuscript are highlighted to indicate changes.

**Response to the editor**

**Comment 1.1**

Clarify Vectorization Benefits: Could you briefly discuss potential vectorization strategies applicable to QDP in a future work section?

**Response 1.1:**

We express our gratitude to you for your review and valuable feedback. In response to your comment regarding the "potential vectorization strategies," we have included a more thorough discussion in the revised version. This includes an exploration of vectorization strategies applied to QDP and provides an outlook for future research in this area. It can be seen as follows (Line 341-351 in manuscript):

> In future work, we plan to explore vectorization strategies for QDP algorithm, building on successful implementations of vectorized compensated summation algorithms. Dmitruk et al. (2023) have efficiently vectorized using Intel AVX-512 intrinsics, with parallelization handled through OpenMP constructs. Numerical experiments have shown that the vectorized summation algorithm achieves performance comparable to traditional summation algorithms, especially for large problem sizes, while maintaining high accuracy. So we intend to apply similar vectorization techniques to QDP algorithm in numerical models, utilizing Single Instruction Multiple Data(SIMD) extensions in modern multicore processors to accelerate the computation of compensated summation and other time-stepping algorithms. Future implementations will also include parallelization via easy-to-use constructs like OpenMP's "declare reduction," which can further speed up execution, especially for large-scale problems. However, for smaller problem sizes or when summation is part of a more complex computation, we may find parallelization to be beneficial even at smaller scales. By incorporating these vectorization and parallelization strategies, we aim to significantly enhance the efficiency and accuracy of QDP algorithm in HPC environments.

**Comment 1.2**

Expand on Algorithm Limitations: Please provide a brief discussion on any known limitations of QDP in spatial discretization.

**Response 1.2:**

Thank you for your insightful suggestion. We have expanded the manuscript to include a brief discussion on the limitations of the QDP method in spatial discretization. This added section can be found as follows(Line 326-334 in manuscript):

> Furthermore, whether the QDP algorithm is applicable to the spatial discretization process requires further investigation. Although floating-point operations such as addition, multiplication, and division are often performed multiple times during spatial discretization, which can introduce round-off errors, these errors do not

accumulate and amplify as they do in time integration. This is primarily because spatial computations generally do not involve the repetitive time accumulation process. Additionally, the errors in spatial discretization mainly stem from discretization errors (such as grid resolution) and the choice of discretization methods (e.g., central difference, forward difference), which differ from round-off errors and are primarily related to the discretization method and grid design. Therefore, the effectiveness of QDP algorithm in this context may not be as pronounced as it is in time integration. As a result, this study does not apply QDP algorithm to the spatial discretization process.

**Comment 1.3**

Enhance Figure Captions: Ensure all figure captions are concise and directly relate to the content of the figures.

**Response 1.3:**

Thank you for the suggestion regarding the figure captions. We have revised the captions to ensure they are more concise and directly aligned with the content of the figures. This improvement aims to enhance the clarity and effectiveness of the visual information presented. The revised version is as follows.

**Figure 1.** Iterative Process of QDP algorithm in Step-by-Step Integration.

**Figure 2.** The QDP algorithm with magnitude preconditioning for Identifying large and small numbers.

**Figure 3.** Time evolution of differences in (a) total energy and (b) total mass between DBL, SGL, and QDP (proposed) in the JW wave. Both figures highlight QDP's superior error compensation, especially over extended integration periods.

**Figure 4.** Spatial distributions of 1–15 day averaged surface pressure differences (Pa) between DBL and (a) SGL, (b) QDP in the JW wave case. QDP reduces errors more effectively, particularly in mid- to high- latitude regions.

**Figure 5.** Time evolution of differences in (a) total energy and (b) total mass for the supercell case: DBL vs. SGL and DBL vs. QDP. The results highlight QDP algorithm's effective error compensation, with benefits becoming more pronounced over time.

**Figure 6.** Perturbation theta at 5400 s in supercell development: (a) DBL, (b) SGL, and (c) QDP (unit: K). The circle highlights pattern biases (consistent values in the same color). Error regions (blue) appear in (b) with single precision, while QDP reduces these errors in (c).

**Figure 7.** Temporal evolution of total energy differences for real-data simulations: DBL vs. SGL (blue) and DBL vs. QDP (red) at resolutions of (a) 240 km × 240 km and (b) 120 km × 120 km. QDP consistently reduces errors from numerical precision loss, with resolution-dependent improvements.

**Figure 8.** Spatial distributions of averaged (1–15 days) surface pressure differences (Pa) between DBL and (a) SGL, (b) QDP (domain size: 240 km × 240 km). The Spatial RMSE is $6.68 \times 10^{-2}$ Pa for (a) and $2.25 \times 10^{-3}$ Pa for (b), highlighting QDP algorithm's significant error reduction across regions and its effectiveness in correcting errors by several orders of magnitude.

**Figure 9.** Spatial distributions of averaged (1–15 days) 500 hPa height differences (m) between DBL and (a) SGL, (b) QDP (domain size: 240 km × 240 km). The Spatial RMSE decreases from $2.80 \times 10^{-1}$ m in (a) to $1.40 \times 10^{-1}$ m in (b), indicating notable spatial error reduction with QDP algorithm.

**Figure 10.** Distributions of averaged (1–15 days) surface pressure differences (Pa) between DBL and (a) SGL, (b) QDP (domain size: 120 km × 120 km). The Spatial RMSE decreases from $6.33 \times 10^{-2}$ Pa in (a) to $2.25 \times 10^{-3}$ Pa in (b). Note: color bars differ between (a) and (b). QDP significantly reduces errors across all regions.

**Figure 11.** Spatial distributions of averaged (1–15 days) 500 hPa height differences (m) between DBL and (a) SGL, (b) QDP (domain size: 120 km × 120 km). The Spatial RMSE decreases from $4.35 \times 10^{-3}$ m in (a) to $1.90 \times 10^{-3}$ m in (b), consistent with the overall improvement shown in Fig. 10.

**Comment 1.4**

Technical Terminology: Include a list of abbreviations or a glossary for better accessibility to a wider audience.

**Response 1.4:**

Thank you for your helpful suggestion. We have added a list of abbreviations and a glossary to the manuscript in the relevant section. The relevant details can be found in Table 1 and Table 2 of the manuscript.

**Comment 1.5**

README File: Consider adding a section on reproducing the study's results, including specific commands used.

**Response 1.5:**

We appreciate your suggestion to include a section on reproducing the study's results. In the revised version of the manuscript, we have added a detailed content in the "code data availability" part. This addition aims to enhance the transparency and reproducibility of our work. It can be seen as follows.

The provided repository includes all relevant code necessary for the study, categorized into four main components:

**1. Download Model Source Code:** (This includes source code of MPAS-v8.2.1 for different simulation modes)

- DBL and SGL:
  - GitHub: `https://github.com/MPAS-Dev/MPAS-Model/releases/tag/v8.2.1` (last access: 26 December 2024)
    * *Note:* If the source code is obtained via the official GitHub repository of the MPAS model, to build a dycore-only MPAS-A model, users need to comment-out or delete the definition of PHYSICS

in the Makefile located in the `src/core_atmosphere/`, e.g., #
PHYSICS=-DDO_PHYSICS.

* *Note:* The source code for SGL is identical to DBL. The difference lies
  in the compilation process, where a specific compilation option is
  used to enable single-precision execution. To compile the model in
  single precision, simply add the `PRECISION=single` flag during the
  build process.

- Zenodo: `https://doi.org/10.5281/zenodo.14576893` (located in
  `code_and_data/model/DBL/` and `code_and_data/model/SGL/`)

- SGL-QDP (proposed):

  - Zenodo: `https://doi.org/10.5281/zenodo.14576893` (located in
    `code_and_data/model/SGL-QDP/`)

  - *Note:* Add the `PRECISION=single` flag during the build process.

**2. Compile Model Source Code:** (This step includes generating the executable files
for `init_atmosphere` and `atmosphere`)

- Compile `init_atmosphere`: Use the following command: make ifort
  CORE=init_atmosphere

- Clean previous builds for `atmosphere`: (if necessary) Use the following com-
  mand: make clean CORE=atmosphere

- Compile `atmosphere`: Use the following command: make ifort
  CORE=atmosphere

- Compile in Single Precision: To compile the model in single precision, add the
  `PRECISION=single` flag: make ifort CORE=atmosphere PRECISION=single

**3. Case Setup and Run:** Specific case setups and configurations used in the experi-
ments are provided, including input files, namelist configurations, and scripts for
running idealized and real-world scenarios. These allow users to replicate the exact
experiments conducted in the study. The steps are as follows:

- Download the archive file for the test cases, which includes mesh files, de-
  composition files, and the namelist file, from the official MPAS website
  at `http://mpas-dev.github.io/` (The idealized test cases currently avail-
  able on the official website include the Supercell, Mountain-Wave, and
  Jablonowski and Williamson Baroclinic Wave). These can also be down-
  loaded directly at `https://doi.org/10.5281/zenodo.14576893` (located
  in `code_and_data/test/`).

- Link the `init_atmosphere` and `atmosphere` executable files, compiled in the
  first part, to the case folder.

- If the code is downloaded directly from Zenodo, users can run the cases by
  following the instructions in the README file or directly executing the `run.sh`
  script. Before running the simulations, ensure to adjust the number of nodes

in the script according to the available computational resources to optimize performance.

**4. Visualization and Post-Processing Code:** In order to reproduce the figures in this paper, follow the instructions below:

- Figure 3 and 4: Run the NCL script by `ncl time.ncl` and `ncl spatical.ncl`. Navigate to the directory: `code_and_data/test/c2_DBL/`.

- Figure 5 and 6: Run the NCL script by `ncl time.ncl` and `ncl spatical.ncl`. Navigate to the directory: `code_and_data/test/c5_DBL/`.

- Figure 7a, 8, and 9: Run the NCL script by `ncl time.ncl` and `ncl spatical.ncl`. Navigate to the directory: `code_and_data/test/c7_240km_DBL/`.

- Figure 7b, 10, and 11: Run the NCL script by `ncl time.ncl` and `ncl spatical.ncl`. Navigate to the directory: `code_and_data/test/c7_120km_DBL/`.